# Schrödinger Bridge Flow
# for Unpaired Data Translation

**Valentin De Bortoli** [*]
Google DeepMind

**Iryna Korshunova** [*]
Google DeepMind

**Andriy Mnih**
Google DeepMind

**Arnaud Doucet**
Google DeepMind

## Abstract

Mass transport problems arise in many areas of machine learning whereby one wants to compute a map transporting one distribution to another. Generative modeling techniques like Generative Adversarial Networks (GANs) and Denoising Diffusion Models (DDMs) have been successfully adapted to solve such transport problems, resulting in CycleGAN and Bridge Matching respectively. However, these methods do not approximate Optimal Transport (OT) maps, which are known to have desirable properties. Existing techniques approximating OT maps for high-dimensional data-rich problems, such as DDM-based Rectified Flow and Schrödinger Bridge procedures, require fully training a DDM-type model at each iteration, or use mini-batch techniques which can introduce significant errors. We propose a novel algorithm to compute the Schrödinger Bridge, a dynamic entropy-regularised version of OT, that eliminates the need to train multiple DDM-like models. This algorithm corresponds to a discretisation of a flow of path measures, which we call the Schrödinger Bridge Flow, whose only stationary point is the Schrödinger Bridge. We demonstrate the performance of our algorithm on a variety of unpaired data translation tasks.

## 1 Introduction

The problem of finding a map to transport one probability distribution to another one has numerous applications in machine learning. In particular, it is at the core of generative modeling where the idea is to transform a noise distribution into the data distribution, and is also central to transfer learning tasks such as image-to-image translation. For discrete probability distributions, it is possible to compute the Optimal Transport (OT) map but this is computationally expensive (Peyré et al., 2019). By showing that an entropy-regularised version of OT, the Entropic OT (EOT), could be computed much more efficiently using the Sinkhorn algorithm, Cuturi (2013) has enabled transport ideas to be used in numerous applications (Ge et al., 2021; Zhou et al., 2022). However, the computational complexity of Sinkhorn algorithm is quadratic in the sample size, which makes its application to very large datasets impractical. Mini-batch versions have been proposed, see e.g. (Genevay et al., 2018), but tend to introduce significant errors in high dimensions (Sommerfeld et al., 2019).

In the context of generative modeling, Denoising Diffusion Models (DDMs) (Song et al., 2021a; Ho et al., 2020) have shown impressive performance in a variety of domains. DDMs define a forward process progressively noising the data, and sample generation is achieved by approximating the time-reversal of this diffusion. In order to leverage the iterative refinement properties of DDMs in the OT setting, methods exploiting the equivalence between the static versions of (E)OT and their dynamic counterparts (Benamou and Brenier, 2000; Léonard, 2014) have been developed. A procedure to approximate the dynamic OT is considered by Liu et al. (2023b), while techniques to approximate the dynamic equivalent to EOT, the Schrödinger Bridge (SB), have been proposed in (De Bortoli et al., 2021; Vargas et al., 2021; Chen et al., 2022; Peluchetti, 2023; Shi et al., 2023). These techniques are

---

[*]Equal contribution.

expensive however, as they require training multiple DDM-type models. Mini-batch versions of OT and Sinkhorn (Pooladian et al., 2023; Tong et al., 2024b) combined with bridge or flow matching have also been proposed to approximate the OT path and SB, but they optimise a minibatch OT objective that can introduce significant errors in high dimensions: the error in Wasserstein-1 distance is of order $O(B^{-1/(2d)})$, where $d$ is the dimension of the problem and $B$ the minibatch size, see (Sommerfeld et al., 2019, Corollary 1).

In this paper, we propose a novel approach to computing the SB. Similarly to Iterative Markovian Fitting (IMF) and its practical implementation, Diffusion Schrödinger Bridge Matching (DSBM) (Shi et al., 2023; Peluchetti, 2023), it leverages the fact that the SB is the only Markov process with prescribed marginals at the endpoints which is in the reciprocal class of the Brownian motion, i.e. it has the same bridge as the Brownian motion (Léonard, 2014); see Section 2 for more details on Markov processes and the reciprocal class. Compared to DSBM, our approach is easier to implement as it does not require caching samples, alternating between optimising two different losses, and, optionally, uses one neural network instead of two. In Section 3, we start by introducing a flow of path measures whose time-discretisation yields a family of algorithms called $\alpha$-IMF and presented in Section 4. Notably, we show that $\alpha$-IMF converges to the Schrödinger Bridge for any $\alpha \in (0, 1]$. Additionally, for a special value of the discretisation stepsize $\alpha = 1$, we recover the IMF procedure (Peluchetti, 2023; Shi et al., 2023), while $\alpha < 1$ corresponds to online versions of IMF. We implement a parametric version of the $\alpha$-IMF as an online DSBM procedure, called $\alpha$-DSBM. We illustrate the efficiency of our approach in *unpaired* image-to-image translation settings in Section 6.

**Notation.** We denote the space of *path measures* by $\mathcal{P}(\mathcal{C})$, i.e. $\mathcal{P}(\mathcal{C}) = \mathcal{P}(\mathrm{C}([0, 1], \mathbb{R}^d))$, where $\mathrm{C}([0, 1], \mathbb{R}^d)$ is the space of continuous functions from $[0, 1]$ to $\mathbb{R}^d$. The subset of *Markov* path measures associated with a diffusion of the form $\mathrm{d}\mathbf{X}_t = v_t(\mathbf{X}_t)\mathrm{d}t + \sigma_t\mathrm{d}\mathbf{B}_t$, with $\sigma, v$ locally Lipschitz, is denoted $\mathcal{M}$. For $\mathbb{Q}$ induced by $(\sqrt{\varepsilon}\mathbf{B}_t)_{t \in [0,1]}$, with $\varepsilon > 0$ and $(\mathbf{B}_t)_{t \geq 0}$ a $d$-dimensional Brownian motion, the *reciprocal class* of $\mathbb{Q}$ is denoted $\mathcal{R}(\mathbb{Q})$, see Definition 2.1. For any $\mathbb{P} \in \mathcal{P}(\mathcal{C})$, we denote by $\mathbb{P}_t$ its marginal distribution at time $t$, $\mathbb{P}_{s,t}$ the joint distribution at times $s, t$, $\mathbb{P}_{s|t}$ the conditional distribution at time $s$ given the state at time $t$, and $\mathbb{P}_{|0,1} \in \mathcal{P}(\mathcal{C})$ the distribution of the path on time interval $(0, 1)$ given its endpoints; e.g. $\mathbb{Q}_{|0,1}$ is a scaled Brownian bridge. Unless specified otherwise, all gradient operators $\nabla$ are w.r.t. the variable $x_t$ with time index $t$. Given probability spaces $(\mathsf{X}, \mathcal{X})$ and $(\mathsf{Y}, \mathcal{Y})$, a Markov kernel $\mathrm{K} : \mathsf{X} \times \mathcal{Y} \to [0, 1]$, and a probability measure $\mu$ defined on $\mathcal{X}$, we write $\mu\mathrm{K}$ for the probability measure on $\mathcal{Y}$ such that for any $\mathsf{A} \in \mathcal{Y}$ we have $\mu\mathrm{K}(\mathsf{A}) = \int_{\mathsf{X}} \mathrm{K}(x, \mathsf{A})\mathrm{d}\mu(x)$. In particular, for any joint distribution $\Pi_{0,1}$ over $\mathbb{R}^d \times \mathbb{R}^d$, we denote the *mixture of bridges* measure as $\Pi = \Pi_{0,1}\mathbb{P}_{|0,1} \in \mathcal{P}(\mathcal{C})$, which is short for $\Pi(\cdot) = \int_{\mathbb{R}^d \times \mathbb{R}^d} \mathbb{P}_{|0,1}(\cdot|x_0, x_1)\mathrm{d}\Pi_{0,1}(x_0, x_1)$. Finally, we define the Kullback–Leibler (KL) divergence between two probability measures $\pi_0, \pi_1 \in \mathcal{P}(\mathsf{X})$ as $\mathrm{KL}(\pi_0|\pi_1) = \int_{\mathsf{X}} \log((\mathrm{d}\pi_0/\mathrm{d}\pi_1)(x))\mathrm{d}\pi_0(x)$ if $\pi_0$ is absolutely continuous w.r.t. $\pi_1$ and $\mathrm{KL}(\pi_0|\pi_1) = +\infty$ otherwise.

## 2  Optimal Transport and Schrödinger Bridge

**Unpaired Transfer and Optimal Transport.** Given unpaired data samples from $\pi_0$ and $\pi_1$, where $\pi_0, \pi_1$ are two distributions on $\mathbb{R}^d$, we are interested in designing a transport map from $\pi_0$ to $\pi_1$. This corresponds to an *unpaired data transfer task*. We can formulate this problem as finding a distribution $\Pi$ on $\mathbb{R}^d \times \mathbb{R}^d$ with marginals $\Pi_0 = \pi_0$ and $\Pi_1 = \pi_1$ so that if $\mathbf{X}_0 \sim \pi_0$ then $\mathbf{X}_1|\mathbf{X}_0 \sim \Pi_{1|0}(\cdot|\mathbf{X}_0)$ satisfies $\mathbf{X}_1 \sim \pi_1$. Among an infinite number of such so-called coupling distributions $\Pi$, we are here interested in finding the Entropic Optimal Transport (EOT) coupling $\Pi^\star$ defined as

$$\Pi^\star = \mathrm{argmin}_{\Pi \in \mathcal{P}(\mathbb{R}^d \times \mathbb{R}^d)} \left\{ \int_{\mathbb{R}^d \times \mathbb{R}^d} \frac{1}{2}\|x - y\|^2 \mathrm{d}\Pi(x, y) - \varepsilon\mathrm{H}(\Pi) \ ; \ \Pi_0 = \pi_0, \ \Pi_1 = \pi_1 \right\}, \quad (1)$$

where $\mathrm{H}(\Pi)$ is the differential entropy of $\Pi$ and $\varepsilon > 0$ is a regularisation hyperparameter (Peyré et al., 2019). For $\varepsilon = 0$, we recover the standard OT.

In order to leverage the recent advances in generative modeling, and in particular the concept of *iterative refinement* central to DDMs, we turn to a *dynamic* formulation of EOT known as the *Schrödinger Bridge* problem (Léonard, 2014). It is defined as follows: find $\mathbb{P}^\star \in \mathcal{P}(\mathcal{C})$ such that

$$\mathbb{P}^\star = \mathrm{argmin}_{\mathbb{P} \in \mathcal{P}(\mathcal{C})}\{\mathrm{KL}(\mathbb{P}|\mathbb{Q}) \ ; \ \mathbb{P}_0 = \pi_0, \ \mathbb{P}_1 = \pi_1\}, \quad (2)$$

with $\mathbb{Q} \in \mathcal{P}(\mathcal{C})$ induced by a scaled $d$-dimensional Brownian motion $(\sqrt{\varepsilon}\mathbf{B}_t)_{t \in [0,1]}$. The term *dynamic* here refers to the fact that (2) is defined on path measures, i.e. on (stochastic) processes, in contrast to the *static* problem (1) which is defined on measures on the space $\mathbb{R}^d \times \mathbb{R}^d$. In Section 3, we show that solving (2) is equivalent to optimising the vector field of a stochastic process using objectives similar to the ones of bridge matching (Peluchetti, 2021; Albergo and Vanden-Eijnden, 2023; Lipman et al., 2023; Liu et al., 2023a). Under mild assumptions, it can be shown that $\mathbb{P}^\star_{0,1} = \Pi^\star$, see e.g. (Léonard, 2014; Pavon et al., 2021). Hence solving (1) reduces to solving (2). Once we have found $\mathbb{P}^\star$ associated with $(\mathbf{X}^\star_t)_{t \in [0,1]}$, we can sample from $\mathbb{P}^\star$ by first sampling $\mathbf{X}^\star_0 \sim \pi_0$ and then sampling the trajectory $(\mathbf{X}^\star_t)_{t \in (0,1]}$ which yields $(\mathbf{X}^\star_0, \mathbf{X}^\star_1) \sim \Pi^\star$.

**Reciprocal and Markov projections.**   To introduce our methodology, it is necessary to recall the notions of reciprocal and Markov projections. We refer to Shi et al. (2023) for more details. For practitioners, a more intuitive explanation of these projections is given in Appendix E.

> **Definition 2.1 (Reciprocal projection):** $\mathbb{P} \in \mathcal{P}(\mathcal{C})$ *is in the reciprocal class* $\mathcal{R}(\mathbb{Q})$ *of* $\mathbb{Q}$ *if* $\mathbb{P} = \mathbb{P}_{0,1}\mathbb{Q}_{|0,1}$. *We define the* reciprocal projection *of* $\mathbb{P} \in \mathcal{P}(\mathcal{C})$ *as* $\mathbb{P}^\star = \mathrm{proj}_{\mathcal{R}(\mathbb{Q})}(\mathbb{P}) = \mathbb{P}_{0,1}\mathbb{Q}_{|0,1}$. *We will write* $\mathrm{proj}_\mathcal{R}$ *instead of* $\mathrm{proj}_{\mathcal{R}(\mathbb{Q})}$ *to simplify notation.*

In other words, $\mathbb{P}$ is in the reciprocal class of $\mathbb{Q}$ if the conditional distribution of a path given its endpoints is identical under $\mathbb{P}$ and $\mathbb{Q}$, see (Rœlly, 2013). Sampling from the reciprocal projection of $\mathbb{P}$ can be achieved by sampling a path $(\mathbf{X}_t)_{t \in [0,1]}$ from $\mathbb{P}$, keeping only the values of the endpoints, say $\mathbf{X}_0, \mathbf{X}_1$, and then sampling a new value for the bridge $(\mathbf{X}_t)_{t \in (0,1)}$ from $\mathbb{Q}_{|0,1}$.

> **Definition 2.2 (Markov projection):** *Assume that* $\mathbb{Q}$ *is induced by* $(\sqrt{\varepsilon}\mathbf{B}_t)_{t \in [0,1]}$ *for* $\varepsilon > 0$. *Then, when it is well-defined, for any* $\mathbb{P} \in \mathcal{R}(\mathbb{Q})$, *the* Markovian projection $\mathbb{M} = \mathrm{proj}_\mathcal{M}(\mathbb{P}) \in \mathcal{M}$ *is the path measure induced by the diffusion* $(\mathbf{X}^\star_t)_{t \in [0,1]}$ *with for any* $t \in [0,1]$
>
> $$\mathrm{d}\mathbf{X}^\star_t = v^\star_t(\mathbf{X}^\star_t)\mathrm{d}t + \sqrt{\varepsilon}\mathrm{d}\mathbf{B}_t, \qquad v^\star_t(x_t) = \left(\mathbb{E}_{\mathbb{P}_{1|t}}[\mathbf{X}_1 \mid \mathbf{X}_t = x_t] - x_t\right)/(1-t), \qquad \mathbf{X}^\star_0 \sim \mathbb{P}_0.$$

In practice, implementing a Markovian projection requires solving a regression problem to approximate $\mathbb{E}_{\mathbb{P}_{1|t}}[\mathbf{X}_1 \mid \mathbf{X}_t = x_t]$, similar to the one appearing in bridge matching and flow matching. One key property of the Markovian projection is that $\mathbb{M}_t = \mathbb{P}_t$ for all $t \in [0,1]$, i.e. the Markovian projection preserves the marginals; see (Peluchetti, 2021) for instance.

**Iterative Markovian Fitting.**   Leveraging the reciprocal and Markovian projections, Peluchetti (2023) and Shi et al. (2023) concurrently introduced IMF. Starting from $\hat{\mathbb{P}}^0 = (\pi_0 \otimes \pi_1)\mathbb{Q}_{|0,1}$, a measure where endpoints are sampled independently from $\pi_0$ and $\pi_1$ and then interpolated using a (scaled) Brownian bridge, they define a sequence of path measures $(\mathbb{P}^n, \hat{\mathbb{P}}^n)_{n \in \mathbb{N}}$ where $\mathbb{P}^n = \mathrm{proj}_\mathcal{M}(\hat{\mathbb{P}}^n)$ and $\hat{\mathbb{P}}^{n+1} = \mathrm{proj}_\mathcal{R}(\mathbb{P}^n)$. This ensures that $\mathbb{P}^n_0 = \pi_0$, $\mathbb{P}^n_1 = \pi_1$ for all $n$, and it can be shown that the sequence $(\mathbb{P}^n)_{n \in \mathbb{N}}$ converges to the SB, see (Peluchetti, 2023, Theorem 2). The practical implementation of this algorithm proposed by Shi et al. (2023) is called DSBM. Implementing DSBM poses challenges, as each Markovian projection requires training a neural network to approximate the relevant conditional expectations by minimising a bridge matching loss. Furthermore, in practice, generated model samples are stored in a cache in order to train the next iterations of DSBM. This introduces additional hyperparameters that require tuning. In Section 3 we propose $\alpha$-IMF, an algorithm which can be interpreted as the discretisation of a *flow of path measures*. This leads to $\alpha$-DSBM, an algorithm that is computationally much more efficient than DSBM as it does not rely on a Markovian projection at each step.

## 3   Schrödinger Bridge flow

We will now introduce a flow of path measures $(\mathbb{P}^s)_{s \geq 0}$, and show that the time-discretisation of this flow with an appropriate stepsize $\alpha \in (0,1]$ yields a family of procedures called $\alpha$-IMF, which all converge to the Schrödinger Bridge. While $\alpha = 1$ yields the classical IMF, $\alpha \in (0,1)$ yields an *incremental* version of IMF. In Section 4 we show that $\alpha$-IMF can be implemented as an *online* version of DSBM.

## 3.1 A flow of path measures

Let $(\mathbb{P}^s, \hat{\mathbb{P}}^s)_{s \geq 0}$ be a *flow of path measures* defined for any $s \geq 0$ by

$$\hat{\mathbb{P}}^0 = (\pi_0 \otimes \pi_1)\mathbb{Q}_{|0,1}, \quad \partial_s \hat{\mathbb{P}}^s = \mathrm{proj}_{\mathcal{R}}(\mathrm{proj}_{\mathcal{M}}(\hat{\mathbb{P}}^s)) - \hat{\mathbb{P}}^s, \quad \mathbb{P}^s = \mathrm{proj}_{\mathcal{M}}(\hat{\mathbb{P}}^s), \qquad (3)$$

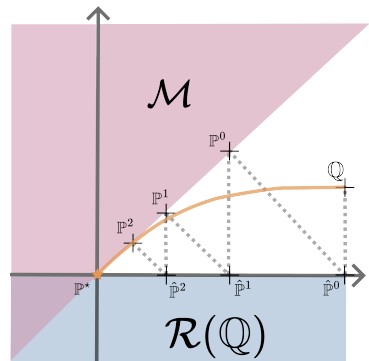

Figure 1: Illustration of the SB Flow and comparison with IMF. $\mathbb{P}^\star$ is the SB, $(\hat{\mathbb{P}}^n)_{n \in \mathbb{N}}$ the IMF sequence and $(\hat{\mathbb{P}}^s)_{s \geq 0}$ the flow we consider. See Appendix B for the analysis of this example.

which we assume is well-defined. Note that for any $s \geq 0$, $\mathbb{P}^s$ is Markov while $\hat{\mathbb{P}}^s$ is in the reciprocal class of $\mathbb{Q}$. Crucially, the only fixed point of (3) is the SB. Indeed, let $\bar{\mathbb{P}}$ be a fixed point of $(\mathbb{P}^s)_{s \geq 0}$ in (3). Then, we have that $\bar{\mathbb{P}} = \mathrm{proj}_{\mathcal{R}}(\mathrm{proj}_{\mathcal{M}}(\bar{\mathbb{P}}))$. Hence, we get $\bar{\mathbb{P}} = \mathrm{proj}_{\mathcal{R}}(\mathrm{proj}_{\mathcal{M}}(\ldots(\mathrm{proj}_{\mathcal{R}}(\mathrm{proj}_{\mathcal{M}}(\bar{\mathbb{P}})))\ldots))$. Hence, under mild assumptions, $\bar{\mathbb{P}}$ is a limit point of IMF and therefore $\bar{\mathbb{P}}$ is the SB $\mathbb{P}^\star$ given by (2), see (Peluchetti, 2023, Theorem 2).

Next, for any $\alpha \in (0, 1]$, we define the following discretisation of (3) called $\alpha$-IMF:

$$\hat{\mathbb{P}}^{n+1} = (1 - \alpha)\hat{\mathbb{P}}^n + \alpha\mathrm{proj}_{\mathcal{R}}(\mathrm{proj}_{\mathcal{M}}(\hat{\mathbb{P}}^n)), \quad (4)$$

and $\mathbb{P}^n = \mathrm{proj}_{\mathcal{M}}(\hat{\mathbb{P}}^n)$. Note that for any $n \in \mathbb{N}$, $\hat{\mathbb{P}}^n \in \mathcal{R}(\mathbb{Q})$. This recovers the IMF procedure (Shi et al., 2023; Peluchetti, 2023) when $\alpha = 1$. Using the definition of the sequence $(\hat{\mathbb{P}}^n)_{n \in \mathbb{N}}$, it is possible to analyse the sequence $(\mathbb{P}^n)_{n \in \mathbb{N}}$ using the properties of the KL divergence as well as the Pythagorean identities derived in (Shi et al., 2023; Peluchetti, 2023). We first introduce some assumptions on the Schrödinger Bridge problem. We recall that the differential entropy of a probability measure $\pi$ is given by

$$\mathrm{H}(\pi) = -\int_{\mathbb{R}^d} \log((\mathrm{d}\pi/\mathrm{dLeb})(x))\mathrm{d}\pi(x),$$

if $\pi$ admits a density with respect to the Lebesgue measure and $+\infty$ otherwise. Recall that $\mathbb{Q}$ is associated with $(\sqrt{\varepsilon}\mathbf{B}_t)_{t \in [0,1]}$ and assume that $\mathbb{Q}_0 = \mathrm{Leb}$. Let $\pi_0, \pi_1 \in \mathcal{P}(\mathbb{R}^d)$ such that

$$\int_{\mathbb{R}^d} \|x\|^2 \mathrm{d}\pi_i(x) < +\infty, \qquad \mathrm{H}(\pi_i) < +\infty,$$

for $i \in \{0, 1\}$. Under these assumptions, we can use the characterisation of the SB as the only path measure that preserves $\pi_0, \pi_1$, and is both Markov and in the reciprocal class of $\mathbb{Q}$ (see e.g. (Léonard, 2014, Theorem 2.12)). We get the following result.

**Theorem 3.1 (Convergence of $\alpha$-IMF):** *Let $\alpha \in (0, 1]$ and $(\mathbb{P}^n, \hat{\mathbb{P}}^n)_{n \in \mathbb{N}}$ defined by (4). Under mild assumptions, we have that $\lim_{n \to +\infty} \mathbb{P}^n = \mathbb{P}^\star$, where $\mathbb{P}^\star$ is the solution of the Schrödinger Bridge problem (2).*

## 3.2 Discretisation and non-parametric loss

We show here that $\alpha$-IMF is associated with an *incremental* version of DSBM for $\alpha \in (0, 1)$.

**Iterative Markovian Fitting.** For any $v : [0, 1] \times \mathbb{R}^d \to \mathbb{R}^d$, we introduce the loss function

$$\mathcal{L}(v, \mathbb{P}) = \int_0^1 \mathcal{L}_t(v_t, \mathbb{P})\mathrm{d}t = \int_0^1 \int_{(\mathbb{R}^d)^3} \left\| v_t(x_t) - \frac{x_1 - x_t}{1 - t} \right\|^2 \mathrm{d}\mathbb{P}_{0,1}(x_0, x_1)\mathrm{d}\mathbb{Q}_{t|0,1}(x_t|x_0, x_1)\mathrm{d}t,$$
$$(5)$$

where we recall that $\mathbb{Q}$ is induced by $(\sqrt{\varepsilon}\mathbf{B}_t)_{t \in [0,1]}$ for some $\varepsilon > 0$. This loss was already considered in (Peluchetti, 2021; Lipman et al., 2023; Liu et al., 2023a; Liu, 2022; Shi et al., 2023). We also define the path measure $\mathbb{P}_v \in \mathcal{P}(\mathcal{C})$ associated with

$$\mathrm{d}\mathbf{X}_t = v_t(\mathbf{X}_t)\mathrm{d}t + \sqrt{\varepsilon}\mathrm{d}\mathbf{B}_t, \qquad \mathbf{X}_0 \sim \pi_0. \qquad (6)$$

Consider first the sequence $(v^n)_{n \in \mathbb{N}}$ defined by

$$v^{n+1} = \text{argmin}_v \mathcal{L}(v, \mathbb{P}_{v^n}). \tag{7}$$

Using Definition 2.2, we have that $\mathbb{P}_{v^{n+1}} = \text{proj}_{\mathcal{M}}(\text{proj}_{\mathcal{R}}(\mathbb{P}_{v^n}))$, which corresponds to $\mathbb{P}^{n+1}$ in the IMF sequence. Therefore we have that $\lim_{n \to +\infty} \mathbb{P}_{v^n} = \mathbb{P}^\star$ under mild assumptions (Peluchetti, 2023, Theorem 2).

**Functional gradient descent.** We now introduce a relaxation of (7), where, instead of considering the argmin, we update the vector field with one gradient step. To define this relaxation, we recall that for a functional $F : \mathcal{F} \to \mathbb{R}$, where $\mathcal{F}$ is an appropriate function space, its functional derivative (Courant and Hilbert, 2008) with reference measure $\mu$ is denoted $\nabla_\mu F$ and is given for any $\phi \in \mathcal{F}$, when it exists, by

$$\lim_{\gamma \to 0}(F(f + \gamma\phi) - F(f))/\gamma = \int \langle \nabla_\mu F(f)(x), \phi(x) \rangle \mathrm{d}\mu(x).$$

Initialised with $v_t^0(x) = (\mathbb{E}_{\hat{\mathbb{P}}_{1|t}^0}[\mathbf{X}_1 \mid \mathbf{X}_t = x] - x)/(1-t)$, where $\hat{\mathbb{P}}^0 = (\pi_0 \otimes \pi_1)\mathbb{Q}_{|0,1}$, we now introduce a sequence of vector fields $(v^n)_{n \in \mathbb{N}}$. This corresponds to training a bridge matching model (see e.g. Liu et al. (2023a); Albergo et al. (2023)), giving $\mathbb{P}_{v^0} = \text{proj}_{\mathcal{M}}(\hat{\mathbb{P}}^0)$. Then for $n \in \mathbb{N}$, let

$$v_t^{n+1}(x) = v_t^n(x) - \delta_n \nabla_{\mu^n} \mathcal{L}_t(v_t^n, \mathbb{P}_{v^n})(x), \tag{8}$$

with $\delta_n > 0$ and $\mu^n \in \mathcal{P}(\mathcal{C})$. The parameters $(\delta_n, \mu^n)_{n \in \mathbb{N}}$ will be made explicit in Proposition 3.2. We emphasize that, in contrast to the IMF procedure, in the online update (8) we do not need to solve a Markovian projection problem at every step; instead we simply take a gradient step on the loss (5).

**Connection with $\alpha$-IMF.** The following proposition shows that $(\mathbb{P}_{v^n})_{n \in \mathbb{N}}$ defined by (8) is associated with $\alpha$-IMF defined in (4).

**Proposition 3.2 (Non-parametric updates are $\alpha$-IMF):** *Let $\alpha \in (0,1]$, $(\mathbb{P}^n, \hat{\mathbb{P}}^n)_{n \in \mathbb{N}}$ as in (4), $\delta_n = \alpha$ and $\mu^n = (1-\alpha)\hat{\mathbb{P}}^n + \alpha \text{proj}_{\mathcal{R}}(\mathbb{P}^n)$. Then, under mild assumptions, we have $\mathbb{P}_{v^n} = \mathbb{P}^n$ for all $n \in \mathbb{N}$.*

Combining Theorem 3.1 to Proposition 3.2, we get that $\lim_{n \to +\infty} \mathbb{P}_{v^n} = \mathbb{P}^\star$, i.e. the non-parametric procedure converges to the SB.

## 4 $\alpha$-Diffusion Schrödinger Bridge Matching

**From DSBM to $\alpha$-DSBM.** In Section 3, we introduced $\alpha$-IMF, a scheme which defines a sequence of path measures converging to the SB for all $\alpha \in (0,1]$. For $\alpha = 1$, this corresponds to the IMF, whose practical DSBM implementation (Shi et al., 2023) requires repeatedly solving an expensive minimisation problem (7). In contrast, for $\alpha < 1$ we are only required to take one (non-parametric) gradient step to update the vector field, see (8). This suggests the following practical implementation of $\alpha$-IMF, called $\alpha$-DSBM: First, pretrain a bridge matching model so that for $t \in [0,1]$ and $x \in \mathbb{R}^d$, $v_t^\theta(x) = (\mathbb{E}_{\hat{\mathbb{P}}_{1|t}^0}[\mathbf{X}_1 \mid \mathbf{X}_t = x] - x)/(1-t)$, where $\hat{\mathbb{P}}^0 = (\pi_0 \otimes \pi_1)\mathbb{Q}_{|0,1}$. Then, perform the parametric version of the update (8):

$$\theta \leftarrow \theta - \alpha \nabla_\theta \mathrm{L}(\theta, \mathbb{P}_{\bar{\theta}}); \ \mathrm{L}(\theta, \mathbb{P}) = \int_0^1 \int_{(\mathbb{R}^d)^3} \left\| v_t^\theta(x_t) - \frac{x_1 - x_t}{1-t} \right\|^2 \mathrm{d}\mathbb{P}_{0,1}(x_0, x_1) \mathrm{d}\mathbb{Q}_{|0,1}(x_t|x_0, x_1) \mathrm{d}t, \tag{9}$$

where $\mathbb{P}_{\bar{\theta}}$ is a stop-gradient version of $\mathbb{P}_{v^\theta}$. In Appendix D.2, we give a theoretical justification for this parametric equivalent of (5) and (8) by showing that, as $\alpha \to 0$, the update on the velocity fields $v^\theta$ given by (9) corresponds to a direction of descent for the non-parametric loss (8) on average. Once again, we emphasize that if we replace the gradient step in (9) with the minimisation $\theta \leftarrow \text{argmin}_\theta \mathrm{L}(\theta, \mathbb{P}_{\bar{\theta}})$, we recover DSBM.

**Bidirectional online procedure.** As with DSBM, directly implementing (9) leads to error quickly accumulating, see Appendix I for details. One way to circumvent this error accumulation issue is to consider a *bidirectional* procedure, in which we train both a forward and a backward

model. This is possible because the Markovian projection coincides for forward and backward path measures, see (Shi et al., 2023, Proposition 9). This suggests considering the loss $\mathcal{L}(v^{\rightarrow}, v^{\leftarrow}, \mathbb{P}^{\rightarrow}, \mathbb{P}^{\leftarrow}) = \int_0^1 \mathcal{L}_t(v_t^{\rightarrow}, v_t^{\leftarrow}, \mathbb{P}^{\rightarrow}, \mathbb{P}^{\leftarrow})\mathrm{d}t$, which is an extension of (5), where

$$\mathcal{L}_t(v_t^{\rightarrow}, v_t^{\leftarrow}, \mathbb{P}^{\rightarrow}, \mathbb{P}^{\leftarrow}) = \int_{(\mathbb{R}^d)^3} \left\| v_t^{\rightarrow}(x_t) - \frac{x_1 - x_t}{1-t} \right\|^2 \mathrm{d}\mathbb{P}_{0,1}^{\leftarrow}(x_0, x_1)\mathrm{d}\mathbb{Q}_{t|0,1}(x_t|x_0, x_1) \tag{10}$$

$$+ \int_{(\mathbb{R}^d)^3} \left\| v_{1-t}^{\leftarrow}(x_t) - \frac{x_0 - x_t}{t} \right\|^2 \mathrm{d}\mathbb{P}_{0,1}^{\rightarrow}(x_0, x_1)\mathrm{d}\mathbb{Q}_{t|0,1}(x_t|x_0, x_1).$$

Similarly to (6), we define $\mathbb{P}_{v^{\rightarrow}}, \mathbb{P}_{v^{\leftarrow}}$, associated with $(\mathbf{X}_t)_{t\in[0,1]}$ and $(\mathbf{Y}_{1-t})_{t\in[0,1]}$ respectively, which are defined by forward and backward SDEs

$$\text{(fwd): } \mathrm{d}\mathbf{X}_t = v_t^{\rightarrow}(\mathbf{X}_t)\mathrm{d}t + \sqrt{\varepsilon}\mathrm{d}\mathbf{B}_t, \ \mathbf{X}_0 \sim \pi_0, \ \text{(bwd): } \mathrm{d}\mathbf{Y}_t = v_t^{\leftarrow}(\mathbf{Y}_t)\mathrm{d}t + \sqrt{\varepsilon}\mathrm{d}\mathbf{B}_t, \ \mathbf{Y}_0 \sim \pi_1. \tag{11}$$

Similarly to (8), we define non-parametric updates for any $n \in \mathbb{N}$, $t \in [0, 1]$ and $x \in \mathbb{R}^d$

$$(v_t^{n+1,\rightarrow}(x), v_t^{n+1,\leftarrow}(x)) = (v_t^{n,\rightarrow}(x), v_t^{n,\leftarrow}(x)) - \delta_n \nabla_{\mu^n} \mathcal{L}_t (v_t^{n,\rightarrow}(x), v_t^{n,\leftarrow}(x), \mathbb{P}_{v^{n,\rightarrow}}, \mathbb{P}_{v^{n,\leftarrow}}) (x).$$

We have the following proposition which ensures our bidirectional procedure is still valid and that the results of Proposition 3.2 still hold.

---

**Proposition 4.1 (Bidirectional updates):** *Let $\alpha \in (0, 1]$. For any $n \in \mathbb{N}$, define $(\mathbb{P}^n, \hat{\mathbb{P}}^n)_{n\in\mathbb{N}}$ by (4). Then, under mild assumption and assuming that $\delta_n = \alpha$ and $\mu^n = (1-\alpha)\hat{\mathbb{P}}^n + \alpha\mathrm{proj}_{\mathcal{R}}(\mathbb{P}^n)$, we have that for any $n \in \mathbb{N}$, $\mathbb{P}_{v^{n,\rightarrow}} = \mathbb{P}_{v^{n,\leftarrow}} = \mathbb{P}^n$.*

---

In Appendix I, we show that in the Gaussian setting the bidirectional procedure (4.1) does not accumulate error when the vector field is approximated, while the unidirectional one (8) does.

**Vector field parameterisation.**  Contrary to existing procedures (Shi et al., 2023; Peluchetti, 2023; Liu, 2022), we do not parameterise $v^{\rightarrow}$ and $v^{\leftarrow}$ using two separate networks. Instead, we consider an additional input $s \in \{0, 1\}$ such that $v_\theta(1, \cdot) \approx v^{\rightarrow}$ and $v_\theta(0, \cdot) \approx v^{\leftarrow}$. This allows us to substantially reduce the number of parameters in the model. The conditioning on $s$ in the network is detailed in Appendix K. Before stating our full algorithm in Algorithm 1, we introduce a batched parametric version of (10). For ease of notation, we write $\mathrm{Interp}_t$ for the operation corresponding to sampling from $\mathbb{Q}_{t|0,1}$, i.e.

$$\mathrm{Interp}_t(\mathbf{X}_0, \mathbf{X}_1, \mathbf{Z}) = (1-t)\mathbf{X}_0 + t\mathbf{X}_1 + \sqrt{\varepsilon(1-t)t}\mathbf{Z}. \tag{12}$$

We are now ready to introduce the batched parametric version of (10). For a given batch of inputs $\mathbf{X}_0^{1:B}$ and $\mathbf{X}_1^{1:B}$, timesteps $t \sim \mathrm{Unif}([0,1])^{\otimes B}$, and $\mathbf{X}_t = \mathrm{Interp}_t(\mathbf{X}_0, \mathbf{X}_1, \mathbf{Z})$ with $\mathbf{Z} \sim \mathcal{N}(0, \mathrm{Id})^{\otimes B}$, we compute the empirical forward and backward losses as

$$\ell^{\rightarrow}(\theta; t, \mathbf{X}_1, \mathbf{X}_t) = \frac{1}{B} \sum_{i=1}^B \|v_\theta (1, t^i, \mathbf{X}_t^i) - (\mathbf{X}_1^i - \mathbf{X}_t^i)/(1-t^i)\|^2, \tag{13}$$

$$\ell^{\leftarrow}(\theta; t, \mathbf{X}_0, \mathbf{X}_t) = \frac{1}{B} \sum_{i=1}^B \left\| v_\theta (0, 1-t^i, \mathbf{X}_t^i) - (\mathbf{X}_0^i - \mathbf{X}_t^i)/t^i \right\|^2.$$

We present the resulting $\alpha$-DSBM in Algorithm 1. Note that in this algorithm, we maintain an Exponential Moving Average (EMA) of model parameters, as is common in diffusion models (Nichol and Dhariwal, 2021). During the finetuning stage, when we generate samples to use as model's inputs, we then have a choice of sampling using the EMA or non-EMA parameters. At test time, we always sample using the EMA parameters, as it is known to improve the visual quality (Song and Ermon, 2020). In Algorithm 1, we specify $\alpha \in (0, 1]$ as a stepsize parameter. In practice, we use Adam (Kingma and Ba, 2015) for optimization, thus the choice of $\alpha$ is implicit and adaptive throughout the training. To emphasize the importance of the parameter $\alpha$, we sweep over its value with an explicit solver SGD in a toy setting, see Appendix K.2. We refer to Appendix K for more details on our experimental setup.

---

**Algorithm 1** $\alpha$-Diffusion Schrödinger Bridge Matching

---

1: **Input:** datasets $\pi_0$ and $\pi_1$, entropic regularisation $\varepsilon$, number of pretraining and finetuning steps $N_{\text{pretraining}}$ and $N_{\text{finetuning}}$, batch size $B$ and half batch size $b = B/2$, EMA decay $\gamma$, initial parameters $\theta$ and initial EMA parameters $\theta^{\text{EMA}} = \theta$, $\alpha \in (0, 1]$
2: **for** $n \in \{1, \ldots, N_{\text{pretraining}}\}$ **do**
3:     Sample $(\mathbf{X}_0, \mathbf{X}_1) \sim (\pi_0 \otimes \pi_1)^{\otimes B}$
4:     Sample $t \sim \text{Unif}([0, 1])^{\otimes B}$ and $\mathbf{Z} \sim \mathcal{N}(0, \text{Id})^{\otimes B}$ and compute $\mathbf{X}_t = \text{Interp}_t(\mathbf{X}_0, \mathbf{X}_1, \mathbf{Z})$
5:     Update $\theta$ with a gradient step on $\frac{1}{2}\left[\ell^{\rightarrow}\left(t^{1:b}, \mathbf{X}_1^{1:b}, \mathbf{X}_t^{1:b}\right) + \ell^{\leftarrow}\left(t^{b+1:B}, \mathbf{X}_0^{b+1:B}, \mathbf{X}_t^{b+1:B}\right)\right]$
6:     Update EMA parameters: $\theta^{\text{EMA}} = \gamma\theta^{\text{EMA}} + (1 - \gamma)\theta$
7: **end for**
8: **for** $n \in \{1, \ldots, N_{\text{finetuning}}\}$ **do**
9:     Sample $(\mathbf{X}_0, \mathbf{X}_1) \sim (\pi_0 \otimes \pi_1)^{\otimes b}$
10:    Sample $\hat{\mathbf{X}}_1$ solving forward SDE (11)-(fwd) with $v_{\theta^{\text{EMA}}}(1, \cdot)$ or $v_\theta(1, \cdot)$ starting from $\mathbf{X}_0$
11:    Sample $\hat{\mathbf{X}}_0$ solving backward SDE (11)-(bwd) with $v_{\theta^{\text{EMA}}}(0, \cdot)$ or $v_\theta(0, \cdot)$ starting from $\mathbf{X}_1$
12:    Sample $t^{\rightarrow} \sim \text{Unif}([0, 1])^{\otimes b}$ and $\mathbf{Z}^{\rightarrow} \sim \mathcal{N}(0, \text{Id})^{\otimes b}$ and compute $\mathbf{X}_t^{\rightarrow} = \text{Interp}_{t^{\rightarrow}}(\hat{\mathbf{X}}_0, \mathbf{X}_1, \mathbf{Z}^{\rightarrow})$
13:    Sample $t^{\leftarrow} \sim \text{Unif}([0, 1])^{\otimes b}$ and $\mathbf{Z}^{\leftarrow} \sim \mathcal{N}(0, \text{Id})^{\otimes b}$ and compute $\mathbf{X}_t^{\leftarrow} = \text{Interp}_{t^{\leftarrow}}(\mathbf{X}_0, \hat{\mathbf{X}}_1, \mathbf{Z}^{\leftarrow})$
14:    Update $\theta$ with a gradient step on $\frac{1}{2}[\ell^{\rightarrow}(t^{\rightarrow}, \mathbf{X}_1, \mathbf{X}_t^{\rightarrow}) + \ell^{\leftarrow}(t^{\leftarrow}, \mathbf{X}_0, \mathbf{X}_t^{\leftarrow})]$ and stepsize $\alpha$
15:    Update EMA parameters: $\theta^{\text{EMA}} = \gamma\theta^{\text{EMA}} + (1 - \gamma)\theta$
16: **end for**
17: **Output:** $(\theta, \theta^{\text{EMA}})$ parameters of the finetuned model

---

## 5 Related work

**Solving Schrödinger Bridge problems.** Schrödinger Bridges (Schrödinger, 1932) have been thoroughly studied through the lens of probability theory (Léonard, 2014) and stochastic control (Dai Pra, 1991; Chen et al., 2021). They recently found applications in generative modeling and related fields leveraging recent advances in diffusion models (De Bortoli et al., 2021; Vargas et al., 2021; Chen et al., 2022). Extensions of these methods to other machine learning problems and modalities were studied in (Shi et al., 2022; Thornton et al., 2022; Liu et al., 2022; Chen et al., 2023; Tamir et al., 2023). Shi et al. (2023); Peluchetti (2023) concurrently introduced the DSBM algorithm which relies on a new procedure called IMF, while the DSB algorithm introduced in (De Bortoli et al., 2021) is based on the standard Iterative Proportional Fitting (IPF) scheme. Neklyudov et al. (2023a,b); Liu et al. (2022) generalise DSBM to arbitrary cost functions, albeit at the expense of having to learn the reciprocal projection which is no longer given by a Brownian bridge. These new methodologies translate to improved numerics when compared to their IPF counterparts, but they remain reliant on alternating between the optimisation of two losses. Finally, we note that the Schrödinger Bridge flow and the $\alpha$-IMF procedure can be linked to the Sinkhorn flow recently introduced by Karimi et al. (2024), see Appendix H.1 for a detailed discussion.

**Sampling-free methodologies.** Sampling-free methodologies have been proposed to solve OT related objectives. In (Liu et al., 2023a; Somnath et al., 2023; Diefenbacher et al., 2024; Cao et al., 2024), the authors perform one step of DSBM, i.e. only consider the pretraining stage of our algorithm. While the obtained bridge might enjoy transport properties, it does not solve an OT problem. In another line of work, Pooladian et al. (2023); Tong et al. (2024a,b); Eyring et al. (2024) have proposed simulation-free methods to minimise OT objectives. However, they target not the OT problem, but a minibatch version of it which coincides with OT only in the limit of infinite batch size, see (Pooladian et al., 2023, Theorem 4.2). Other sampling-free methods to solve the Schrödinger Bridge problem include Kim et al. (2024); Gushchin et al. (2024b) both of which rely on adversarial losses to solve the OT problem. In (De Bortoli et al., 2021; Vargas et al., 2021; Liu et al., 2022; Shi et al., 2023; Peluchetti, 2023) the adversarial objective is dropped and instead the procedure requires alternating objectives during training and is not sampling-free. We also highlight the line of work of Korotin et al. (2024); Gushchin et al. (2024a) in which the Schrödinger Bridge potentials are parameterised with mixtures of Gaussians, allowing for fast training in small dimensions. Finally, recently Deng et al. (2024) introduced a variation on Schrödinger Bridge for generative modeling, which while still not sampling-free, does not require learning a forward process.

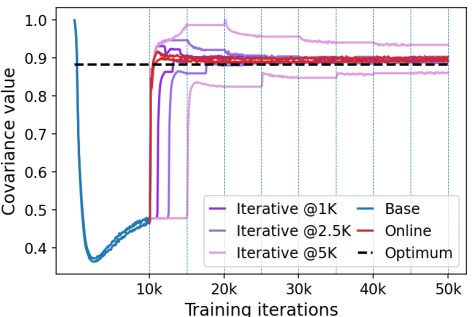 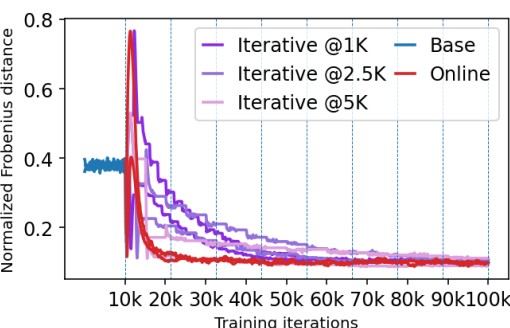

Figure 2: Evolution of the covariance during online and iterative DSBM finetuning for forward and backward networks. The finetuning starts after 10K steps of training a bridge matching model. For the iterative case, we alternate between forward and backward updates with varying frequencies, i.e. changing after 1K, 2.5K and 5K steps. **Left**: Gaussian with scalar covariance matrix. **Right**: Gaussian with full covariance matrix. We compute the normFrob between $C_\star$ and its estimate using Bridge Matching (Base), $\alpha$-DSBM (Online), and DSBM (Iterative with @xK training steps per model fit)

# 6 Experiments

In this section, we illustrate the efficiency of $\alpha$-DSBM on different tasks. In Section 6.1, we compare $\alpha$-DSBM to DSBM in a Gaussian setting where the EOT coupling is tractable and show that $\alpha$-DSBM recovers the solution faster than DSBM. In Section 6.2, we illustrate the scalability of our method through a range of unpaired image translation experiments.

## 6.1 Gaussian case

We compare $\alpha$-DSBM to DSBM in the Gaussian setting where $\pi_0 = \mathcal{N}(0, \sigma_0^2 \mathrm{Id})$, $\pi_1 = \mathcal{N}(0, \sigma_1^2 \mathrm{Id})$ and $\mathbb{Q}$ is associated with $(\sqrt{\varepsilon}\mathbf{B}_t)_{t \in [0,1]}$ with $\sqrt{\varepsilon} = 0.5$. In this case, the EOT coupling is $\mathcal{N}(0, \Sigma_\star)$, with $\Sigma_\star$ given by

$$\Sigma_\star = \begin{pmatrix} \sigma_0^2 \mathrm{Id} & \sigma_\star^2 \mathrm{Id} \\ \sigma_\star^2 \mathrm{Id} & \sigma_1^2 \mathrm{Id} \end{pmatrix}, \text{ where } \sigma_\star^2 = (1/2)((\sigma_0^2 \sigma_1^2 + \varepsilon^2)^{1/2} - \varepsilon),$$

with Id being a $d \times d$ identity matrix. We consider $d = 50$, $\sigma_0 = \sigma_1 = 1$, resulting in $\sigma_\star^2 \approx 0.88$. To showcase the robustness of $\alpha$-DSBM, we consider the initial coupling $\mathbb{P}_{0,1}$, where $(\mathbf{X}_0, \mathbf{X}_1) \sim \mathbb{P}_{0,1}$, $\mathbf{X}_0 \sim \mathcal{N}(0, \mathrm{Id})$, $\mathbf{X}_1 = -\mathbf{X}_0$, and let $\hat{\mathbb{P}}^0 = \mathbb{P}_{0,1} \mathbb{Q}_{|0,1}$. In this setting, the base model, i.e. bridge matching, significantly underestimates the true covariance $\sigma_\star^2$, as shown in Section 6.1. Additionally, the figure illustrates that online finetuning approaches the true solution faster than the original iterative DSBM finetuning. For the latter, we can set how often we alternate between updating the forward and backward networks, and as this frequency increases, the behaviour approaches that of the online finetuning.

**Full covariance Gaussian case.** Let $\pi_0 = \mathcal{N}(\mu_0, \Sigma_0)$, $\pi_1 = \mathcal{N}(\mu_1, \Sigma_1)$ with $\Sigma_i = \mathrm{Id} + \frac{1}{2} Z_i Z_i^\top$ for $i \in \{0, 1\}$ and $Z_0, Z_1$ independent $d \times d$ matrices with unit Gaussian entries. We also set $\mu_0 = \mu_1 = 0$. We consider the Entropic Optimal Transport (EOT) with regularization $\sigma = 0.5$ and $d = 3$, given by

$$\Pi = \mathcal{N}(\mu_\star, \Sigma_\star), \quad \Sigma_\star = \begin{pmatrix} \Sigma_0 & C_\star \\ C_\star^\top & \Sigma_1 \end{pmatrix},$$

with $C_\star = \frac{1}{2}[\Sigma_0^{1/2} D_\star \Sigma_0^{-1/2} - \sigma^2 \mathrm{Id}]$, with $D_\star = (4\Sigma_0^{1/2} \Sigma_1 \Sigma_0^{1/2} + \sigma^4 \mathrm{Id})^{1/2}$. Let normFrob $= \|A - B\|_{\mathrm{Fro}} / \|A\|_{\mathrm{Fro}}$ be the normalized Frobenius distance between matrices A and B. The results are presented in Section 6.1 and confirm those presented in the original manuscript considering a diagonal covariance.

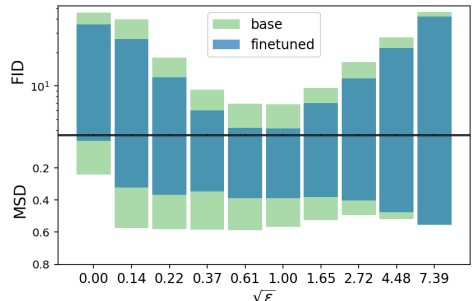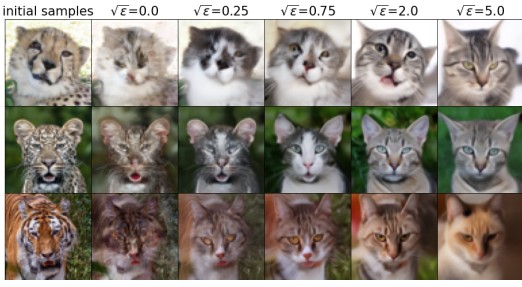

Figure 3: **Left**: FID and Mean Squared Distance (MSD) on EMNIST to MNIST translation before and after finetuning with different values of $\varepsilon$. **Right**: AFHQ-64 samples after the finetuning. For both, we use a bidirectional model with online finetuning. More results are in Appendix K.3 and K.4.

## 6.2 Image datasets

Similarly to Shi et al. (2023), we apply our method to image translation problems, such as MNIST digits to EMNIST letters (LeCun and Cortes, 2010; Cohen et al., 2017), Wild to Cat domains from the Animal Faces-HQ (AFHQ) dataset (Choi et al., 2020), downsampled to $64 \times 64$ and $256 \times 256$ resolutions and CelebA $64 \times 64$.

The whole training procedure can be framed as a two-stage process: first, we train a base model on the true data samples, performing bridge matching (Peluchetti, 2021; Albergo and Vanden-Eijnden, 2023; Lipman et al., 2023; Liu et al., 2023a), and then we finetune this model. We compare models that combine different vector field parameterisations (two networks vs. one bidirectional net), finetuning methods (iterative vs. online), and sample generation strategies during the finetuning stage.

Following the established practice (Choi et al., 2020), we evaluate our models using FID (Heusel et al., 2017) for visual quality, and mean squared distance (MSD) or LPIPS (Zhang et al., 2018) for alignment. It is important to note that for image translation tasks at hand, FID scores are not ideal, as FID was designed for natural RGB images, which is not the case for MNIST. It is also not well suited for small sample sizes as it is the case with AFHQ, where the test set in each domain has fewer than 500 examples. Thus quantitative results in Table 1 should be interpreted cautiously, and we recommend a visual inspection of samples to complement these quantitative measures, especially for the AFHQ models. Samples from the models along with the training and evaluation protocols are given in Appendix K.

Compared to the iterative DSBM, our online finetuning $\alpha$-DSBM reduces the number of tunable hyperparameters, i.e. inner and outer iterations, refresh rate and the size of the cache for storing generated samples. This simplifies implementation and makes the algorithm more practical. The primary remaining hyperparameter, the variance of a Brownian motion $\varepsilon$, requires careful tuning as it influences the trade-off between the visual quality and alignment, as was also observed in Shi et al. (2023). An appropriate $\varepsilon$ needs to balance the two: setting $\varepsilon$ too low results in poor visual quality, while high values of $\varepsilon$ cause poorly aligned and oversmoothed samples. Figure 3 illustrates how FID and MSD metrics vary with $\varepsilon$ for the case of MNIST. Additionally, it demonstrates the impact of $\varepsilon$ on the generated samples for the AFHQ-64 model.

We run $\alpha$-DSBM on CelebA with image size $64 \times 64$ with $\sigma = 2.0$. We do not change the training hyper-parameters compared to AFHQ. Visual results are reported in Figure 5 and Figure 6. In Figure 5, we show the influence of $\sigma$ during the pretraining. The visual quality of the transfer is much lower for $\sigma = 0$ than for $\sigma = 2.0$. The case $\sigma = 0$ corresponds to the first step of Rectified Flow (i.e. Flow Matching). Given the poor quality of the samples, we do not perform finetuning with $\sigma = 0$. In Figure 6, we compare the visual quality and alignment of DSBM and $\alpha$-DSBM after $4000$ training steps, corresponding to two outer DSBM iterations. In this case DSBM is trained with a bidirectional network and both procedures consist of finetuning the pretrained model obtained with $\sigma = 2.0$. We note that the alignment is better in the case of $\alpha$-DSBM.

| Method | EMNIST → MNIST | | AFHQ-64 Wild → Cat | |
| --- | --- | --- | --- | --- |
| | FID | MSD | FID | LPIPS |
| DSBM* | 10.59 | 0.375 | – | – |
| Pretrained two-networks model | 6.02 | 0.564 | 25.97 | 0.589 |
| (a) iterative finetuning | $5.25_{\pm0.15}$ | $0.345_{\pm0.001}$ | $25.41_{\pm0.84}$ | $0.485_{\pm0.003}$ |
| (b) online finetuning | $4.28_{\pm0.07}$ | $0.368_{\pm0.001}$ | $28.752_{\pm1.191}$ | $0.487_{\pm0.003}$ |
| (c) online finetuning without EMA | $4.23_{\pm0.171}$ | $0.361_{\pm0.002}$ | $32.665_{\pm0.647}$ | $0.445_{\pm0.002}$ |
| Pretrained bidirectional model | 6.33 | 0.572 | 29.44 | 0.584 |
| (d) online finetuning | $4.39_{\pm0.09}$ | $0.387_{\pm0.003}$ | $26.579_{\pm0.434}$ | $0.482_{\pm0.001}$ |
| (e) online finetuning without EMA | $4.57_{\pm0.17}$ | $0.369_{\pm0.003}$ | $30.638_{\pm1.023}$ | $0.451_{\pm0.002}$ |

Table 1: Results of image translation between EMNIST and MNIST, and AFHQ 64×64 between Wild and Cat domains. DSBM* results are from Shi et al. (2023). Our reimplementation of DSBM corresponds to row (a). For MNIST and AFHQ models, we used $\varepsilon = 1$ and $\varepsilon = 0.75^2$, respectively. Each finetuning run was done with 5 random seeds, and we report mean scores ± standard deviation.

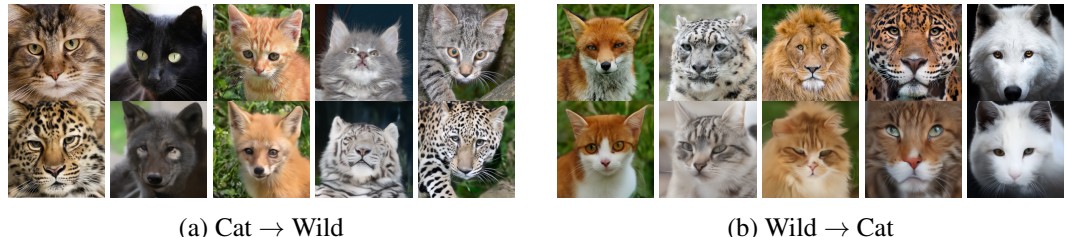

(a) Cat → Wild  (b) Wild → Cat

Figure 4: Online DSBM transfer results on AFHQ 256× 256 dataset between Cat and Wild domains. Top row—initial samples, bottom row—transferred samples.

# 7 Discussion

In this paper we have introduced $\alpha$-Diffusion Schrödinger Bridge Matching ($\alpha$-DSBM), a new methodology to solve Entropic Optimal Transport problems. $\alpha$-DSBM is an improved version of DSBM, which does not require training multiple DDM-type models. We have shown that a non-parametric version of this method recovers the Schrödinger Bridge (SB). In addition, $\alpha$-DSBM is easier to implement than existing SB methodologies while exhibiting similar performance. We illustrated the efficiency of our algorithm on a variety of unpaired transfer tasks.

While $\alpha$-DSBM solves one of the most critical limitations of DBSM, namely the alternative optimisation, several issues remain to be addressed in order for the method to scale comparably to generative DDMs. In particular, the method is not sampling-free, as during training it requires sampling from the model from the previous iteration to obtain the training data for the current iteration. While it seems difficult to derive a completely sampling-free method to solve SB problems without resorting to the Minibatch OT approximation, there is still room for improvement.

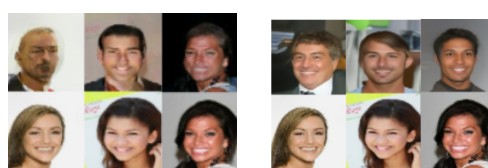 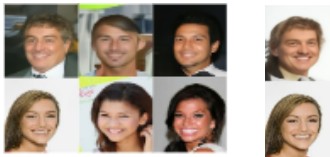

Figure 5: Translation Female → Male on CelebA. Left: pretraining with $\sigma = 0$. Right: pretraining with $\sigma = 2.0$.

Figure 6: Translation Female → Male on CelebA. Left: output after finetuning with DSBM. Right: output after $\alpha$-DSBM finetuning.

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

# A  Appendix organisation

The supplementary material is organised as follows. First in Appendix B, we analyze an Euclidean counterpart to the $\alpha$-IMF sequence identified in Section 3 and the associated flow. In Appendix C, we show that the Markovian projection can be recovered as the parameterisation of the vector field that minimises the accumulation of errors, extending the results of Chen et al. (2024). Theoretical results are gathered in Appendix D. In particular in Appendix D.1 we show that the proposed non-parametric method coincides with the $\alpha$-IMF and prove the convergence of the $\alpha$-IMF. In Appendix D.2, we show the connection between the non-parametric and the parametric updates. In Appendix E, we provide more background on DSBM and propose an extension of the DSBM methodology. Consistency losses similar to (Daras et al., 2023b; De Bortoli et al., 2024) are proposed in Appendix F. Model stitching procedures are described in Appendix G. We comment on extended related work in Appendix H. In particular we draw connections with Sinkhorn flows (Karimi et al., 2024), Reinforcement Learning policies, Expectation-Maximisation schemes following (Brekelmans and Neklyudov, 2023) and comment on finetuning of diffusion models. In Appendix I, we investigate the accumulation of bias in a Gaussian setting and compare forward-forward and forward-backward methods. In Appendix J, we derive the preconditioning of loss following the principles of (Karras et al., 2022) in the case of bridge matching. Additional results and experimental details are presented in Appendix K.

# B  Euclidean flow and iterative procedure

In this section, we study a simplified counterpart of the Schrödinger flow and of DSBM in a Euclidean setting. The goal of this section is to draw some conclusions in the Euclidean case which also remain true empirically when analyzing the Schrödinger Bridge problem.

We consider the set $A_1 = \{(x,y) \in \mathbb{R}^2 : y \geq x\}$ and the set $A_2 = \{(x,y) \in \mathbb{R}^2 : y \leq 0\}$. Loosely speaking, one can identify $A_1$ with the reciprocal class $\mathcal{R}(\mathbb{Q})$ and $A_2$ with the set of Markov path measures $\text{proj}_{\mathcal{M}}$. In that case, we have that for any $(x,y) \in \mathbb{R}^2$, $\text{proj}_{A_1}((x,y)) = ((x+y)/2, (x+y)/2)$ if $(x,y) \notin A_1$ and otherwise, $\text{proj}_{A_1}((x,y)) = (x,y)$. In addition, we have that for any $(x,y) \in \mathbb{R}^2$, $\text{proj}_{A_2}((x,y)) = (x,0)$ if $(x,y) \notin A_2$ and $\text{proj}_{A_2}((x,y)) = (x,y)$ otherwise. We consider the following flow $(x_t, y_t)_{t \geq 0}$ given by

$$\partial_t(x_t, y_t) = \text{proj}_{A_1}(\text{proj}_{A_2}((x_t, y_t))) - (x_t, y_t).$$

Let $(x_0, y_0) \notin A_1$ and $(x_0, y_0) \notin A_2$. Denote $T$ the explosion time of $(x_t, y_t)$, i.e. for any $t \geq T$ we have that $(x_t, y_t) = \infty$, where $\mathbb{R}^2 \cup \{\infty\}$ is the one-point compactification of $\mathbb{R}^2$. Finally, denote $\tau \leq T$ such that for any $t \in [0, \tau]$, $(x_t, y_t) \notin A_1$ and $(x_t, y_t) \notin A_2$. Then, we have

$$\partial_t(x_t, y_t) = (-x_t/2, x_t/2 - y_t).$$

Hence, we have that $x_t = x_0 \exp[-t/2]$ for any $t \in [0, \tau]$ and $y_t = x_0 \exp[-t/2] + (x_0 \exp[-t/2])^2(y_0 - x_0)/x_0^2$. Therefore, we get that $\tau = T = +\infty$ and we have that for any $t \geq 0$

$$x_t = x_0 \exp[-t/2], \qquad y_t = x_t + x_t^2(y_0 - x_0)/x_0^2.$$

Hence, $((x_t, y_t))_{t \geq 0}$ converges exponentially fast to $(0,0)$ with rate $1/2$.

We now investigate the rate of convergence of the alternate projection scheme, i.e. the Euclidean equivalent of DSBM. We define $((x_n, y_n))_{n \in \mathbb{N}}$ such that for any $n \in \mathbb{N}$,

$$(x_{n+1}, y_{n+1}) = \text{proj}_{A_1}(\text{proj}_{A_2}((x_n, y_n))) = (x_n/2, 0).$$

Hence, we get that $x_n = x_0 2^{-n}$ and therefore $((x_n, y_n))_{n \in \mathbb{N}}$ converges exponentially fast to $(0,0)$. Note that this procedure corresponds to a discretisation of the flow $((x_t, y_t))_{n \in \mathbb{N}}$ with stepsize $\alpha = 1$.

More generally, we define for any $\alpha \in (0,1]$, $((x_n^\alpha, y_n^\alpha))_{n \in \mathbb{N}}$ such that for any $n \in \mathbb{N}$,

$$(x_{n+1}^\alpha, y_{n+1}^\alpha) = \alpha \text{proj}_{A_1}(\text{proj}_{A_2}((x_n^\alpha, y_n^\alpha))) + (1-\alpha)(x_n^\alpha, y_n^\alpha).$$

Hence, we get that $x_n = x_0 2^{-n}$ and therefore $((x_n, y_n))_{n \in \mathbb{N}}$ converges exponentially fast to $(0,0)$. It can be shown that for any $n \in \mathbb{N}$, $x_n^\alpha = x_0^\alpha (1 - \alpha/2)^n$ and in addition, we have that

$$y_n^\alpha = (1-\alpha)^n y_0^\alpha + \alpha x_0^\alpha \sum_{k=0}^{n-1} (1-\alpha)^k (1-\alpha/2)^{n-k}.$$

Therefore, we get that

$$y_n = (1-\alpha)^n y_0^\alpha + 2(1 - (1 - \alpha/(2-\alpha))^n)(1 - \alpha/2)^n x_0^\alpha.$$

We now analyse the complexity of the different discretisations assuming that the cost of discretising the flow with stepsize $\alpha \in (0,1]$ is $C^\alpha$. In that case in order to reach the threshold value $\varepsilon$, i.e. $|x_n^\alpha| \le \varepsilon$, we get a total cost $C_n^\alpha = O(\log(1/\varepsilon)C^\alpha/\log(1/(1-\alpha/2)))$, where we have neglected the terms that do not depend on $\log(1/\varepsilon)$. Hence, if $C^\alpha$ is constant then the choice $\alpha = 1$ is the best possible one in the range $\alpha \in (0,1]$. Otherwise, one has to consider the ratio $C^\alpha/\log(1/(1-\alpha/2))$, where the lower is the better. The flow procedure and the iterative based one are presented in Figure 1.

Based on this simplified Euclidean experiment, we draw some conclusions which also remain true in our setting, see Appendix K for more experimental details. First, we have that different discretisations of the flow yield different convergence rates. Large stepsizes incur faster convergence. This suggests to choose $\alpha = 1$. However, if the cost of choosing $\alpha = 1$ is too high then one might turn to alternative schemes with $\alpha \in (0,1)$ assuming that $C^\alpha < C^1$ in that case. To draw a parallel with our setting, in the case of DSBM (case $\alpha = 1$), we need to solve the projection subproblem at each step which incurs a great cost. On the other hand, one step of the online algorithm only requires sampling once from the model and performing one gradient step.

## C   Minimisation of errors and Markovian projection

For a given non-Markovian (stochastic) interpolant process (see definition below), there exist an infinite number of Markov processes admitting the same marginals (Albergo and Vanden-Eijnden, 2023). In this section, when it is well-defined, we show that the Markovian projection corresponds to the process which minimises an error measure (defined further) in case one has access to the oracle of $x_t \mapsto \mathbb{E}[\mathbf{X}_1 \mid \mathbf{X}_t = x_t]$.

**Stochastic Interpolant.**   We first start by recalling the framework of Albergo and Vanden-Eijnden (2023). Consider a coupling $\Pi$ between $\pi_0$ and $\pi_1$, one builds a (stochastic) flow between $\pi_0$ and $\pi_1$ using the following interpolation procedure

$$\mathbf{X}_t = \mathrm{Interp}_t(\mathbf{X}_0, \mathbf{X}_1, \mathbf{Z}) = \alpha_t \mathbf{X}_0 + \beta_t \mathbf{X}_1 + \gamma_t \mathbf{Z}, \quad (\mathbf{X}_0, \mathbf{X}_1) \sim \Pi, \quad \mathbf{Z} \sim \mathcal{N}(0, \mathrm{Id}),$$

where $\alpha_1 = \beta_0 = \gamma_0 = \gamma_1 = 0$ and $\alpha_0 = \beta_1 = 1$. This defines a non-Markovian process. We denote by $\pi_t$ the induced unconditional distribution of $\mathbf{X}_t$. Let us now consider the Markov process $(\mathbf{X}_t^\varepsilon)$ given by

$$\mathrm{d}\mathbf{X}_t^\varepsilon = \mathbb{E}\left[\dot\alpha_t \mathbf{X}_0 + \dot\beta_t \mathbf{X}_1 + (\dot\gamma_t - \varepsilon_t^2/(2\gamma_t))\mathbf{Z} \mid \mathbf{X}_t = \mathbf{X}_t^\varepsilon\right] + \varepsilon_t \mathrm{d}\mathbf{B}_t, \qquad \mathbf{X}_0^\varepsilon \sim \pi_0, \qquad (14)$$

where $(\mathbf{B}_t)_{t\in[0,1]}$ is a $d$-dimensional Brownian motion and $\varepsilon_t$ is an additional hyperparameter. It can then be shown that $(\mathbf{X}_t^\varepsilon)_{t\in[0,1]}$ satisfies that $\mathbf{X}_t^\varepsilon \sim \pi_t$ for all $t \in [0,1]$; see e.g. (Albergo et al., 2023, Theorem 2.8, Corollary 2.10). Hence $(\mathbf{X}_t^\varepsilon)_{t\in[0,1]}$ is a (stochastic) flow mapping $\pi_0$ onto $\pi_1$. Note that $(\mathbf{X}_t^\varepsilon)_{t\in[0,1]}$ in (14) can be rewritten as

$$\mathrm{d}\mathbf{X}_t^\varepsilon = (\dot\alpha_t/\alpha_t)\mathbf{X}_t^\varepsilon + \mathbb{E}\left[(\dot\beta_t - \beta_t\dot\alpha_t/\alpha_t)\mathbf{X}_1 + (\dot\gamma_t - \gamma_t\dot\alpha_t/\alpha_t - \varepsilon_t^2/(2\gamma_t))\mathbf{Z} \mid \mathbf{X}_t = \mathbf{X}_t^\varepsilon\right] + \varepsilon_t \mathrm{d}\mathbf{B}_t.$$
$$(15)$$

In the specific case where $\alpha_t = 1 - t$, $\beta_t = t$ and $\gamma_t = \sigma_0\sqrt{t(1-t)}$ then (15) becomes

$$\mathbf{X}_t = \mathrm{Interp}_t(\mathbf{X}_0, \mathbf{X}_1, \mathbf{Z}) = (1-t)\mathbf{X}_0 + t\mathbf{X}_1 + \sigma_0\sqrt{t(1-t)}\mathbf{Z}.$$

This corresponds to the marginal distribution of the bridge associated with $(\sigma_0\mathbf{B}_t)_{t\in[0,1]}$. In this case, (15) becomes

$$\mathrm{d}\mathbf{X}_t^\varepsilon = \mathbb{E}\left[(\mathbf{X}_1 - \mathbf{X}_t)/(1-t)|\mathbf{X}_t = \mathbf{X}_t^\varepsilon\right]\mathrm{d}t + \sqrt{2}\sigma_0\mathrm{d}\mathbf{B}_t$$

for $\varepsilon_t^2 = (2\gamma_t)(\dot\gamma_t - \gamma_t\dot\alpha_t/\alpha_t) = 2\sigma_0^2$. In Proposition C.1, we will show that this choice of $(\varepsilon_t)_{t\in[0,1]}$ is optimal in some sense.

Consider $(\hat{\mathbf{X}}_t^\varepsilon)_{t\in[0,1]}$ given by

$$\mathrm{d}\hat{\mathbf{X}}_t^\varepsilon = (\dot\alpha_t/\alpha_t)\hat{\mathbf{X}}_t^\varepsilon + (\dot\beta_t - \beta_t\dot\alpha_t/\alpha_t)\mathbb{E}[\mathbf{X}_1 \mid \mathbf{X}_t = \hat{\mathbf{X}}_t^\varepsilon] + (\dot\gamma_t - \gamma_t\dot\alpha_t/\alpha_t - \varepsilon_t^2/(2\gamma_t))\hat{\mathbf{Z}}(t, \hat{\mathbf{X}}_t^\varepsilon) + \varepsilon_t\mathrm{d}\mathbf{B}_t,$$

with $\hat{\mathbf{X}}_0^\varepsilon \sim \pi_0$ and where $\hat{\mathbf{Z}}(t, x)$ is an approximation of $\mathbb{E}[\mathbf{Z}|\mathbf{X}_t = x]$. We have the following result.

**Proposition C.1 (Optimality and stochastic interpolant):** *Denote $\mathbb{P}^\varepsilon$, respectively $\hat{\mathbb{P}}^\varepsilon$ the path measures associated with $(\mathbf{X}_t^\varepsilon)_{t\in[0,1]}$ and $(\hat{\mathbf{X}}_t)_{t\in[0,1]}$ respectively. Consider $\ell(\varepsilon) = \mathrm{KL}(\mathbb{P}^\varepsilon|\hat{\mathbb{P}}^\varepsilon)$. Let $\varepsilon^\star = \mathrm{argmin}_\varepsilon \ell(\varepsilon)$. Then we have*

$$(\varepsilon_t^\star)^2 = 2\gamma_t \dot{\gamma}_t - 2\gamma_t^2 \dot{\alpha}_t/\alpha_t.$$

*In particular, if $\alpha_t = t$, $\beta_t = 1 - t$ and $\gamma_t = \sigma_0\sqrt{t(1-t)}$, then $\varepsilon_t = \sqrt{2}\sigma_0$. The value $(\varepsilon_t^\star)_{t\in[0,1]}$ corresponds to Markovian projection when it is well defined.*

*Proof.* We have that for any

$$\mathrm{KL}(\mathbb{P}^\varepsilon|\hat{\mathbb{P}}^\varepsilon) = \int_0^1 \frac{1}{\varepsilon_t^2}(\dot{\gamma}_t - \gamma_t \dot{\alpha}_t/\alpha_t - \varepsilon_t^2/(2\gamma_t))^2 \mathbb{E}[\Delta_t]\mathrm{d}t,$$

where $\Delta_t = \|\hat{\mathbf{Z}}(t, \mathbf{X}_t^\varepsilon) - \mathbb{E}[\mathbf{Z} \mid \mathbf{X}_t = \mathbf{X}_t^\varepsilon]\|^2$ and the expectation is w.r.t. $\mathbb{P}^\varepsilon$. We have that $\mathrm{KL}(\mathbb{P}^\varepsilon|\hat{\mathbb{P}}^\varepsilon) = 0$ if $\varepsilon = \varepsilon^\star$, which concludes the proof. $\square$

Proposition C.1 is related to (Chen et al., 2024, Section 3.4). Therein it is noticed that, in the case of Augmented Bridge matching (De Bortoli et al., 2023), the choice of $\varepsilon_t$ does not affect the joint distribution of $(\mathbf{X}_0^\varepsilon, \mathbf{X}_1^\varepsilon)$. The authors then select $(\varepsilon_t)$ so as to minimise an approximation error. They show that, in that case, they recover the Föllmer process.

We now show that Proposition C.1 can be further strengthened to establish that $\varepsilon^\star$ is also the optimal value if we interpolate between $\pi_s$ and $\pi_1$, or $\pi_0$ and $\pi_s$, for any $s \in [0, 1]$ and $\pi_s$ the distribution of $\mathbf{X}_s$. Consider in this context for any $s, t \in [0, 1]$ with $t \geq s$, $\gamma_t/\gamma_s \geq \alpha_t \geq \alpha_s$ the following interpolation model.

$$\mathbf{X}_t = (\alpha_t/\alpha_s)\mathbf{X}_s + (\beta_t - \alpha_t\beta_s/\alpha_s)\mathbf{X}_1 + \sqrt{\gamma_t^2 - \alpha_t^2\gamma_s^2/\alpha_s^2}\mathbf{Z}, \qquad (16)$$

where $\mathbf{X}_s \sim \pi_s$, $\mathbf{X}_1 \sim \pi_1$ and $\mathbf{Z} \sim \mathcal{N}(0, \mathrm{Id})$. Assume that $\alpha_t = 1-t$, $\beta_t = t$ and $\gamma_t = \sigma_0\sqrt{t(1-t)}$ for any $t \in [0, 1]$ then (16) corresponds to the Brownian bridge associated with $(\sigma_0\mathbf{B}_t)_{t\in[0,1]}$ with endpoints $\mathbf{X}_s$ at time $s$ and $\mathbf{X}_1$ at time 1. We have the following proposition.

**Proposition C.2 (Stochastic interpolant with intermediate time points):** *Define $(\mathbf{X}_{t,s}^\varepsilon)_{t\in[s,1]}$ given by*

$$\mathrm{d}\mathbf{X}_{t,s}^\varepsilon = (\dot{\alpha}_t/\alpha_t)\mathbf{X}_{t,s}^\varepsilon + \mathbb{E}\left[(\dot{\beta}_t - \beta_t\dot{\alpha}_t/\alpha_t)\mathbf{X}_1 \mid \mathbf{X}_t = \mathbf{X}_{t,s}^\varepsilon\right] \qquad (17)$$

$$+ \mathbb{E}\left[(\dot{\gamma}_{t,s} - \gamma_{t,s}\dot{\alpha}_t/\alpha_t - \varepsilon_{t,s}^2/(2\gamma_{t,s}))\mathbf{Z} \mid \mathbf{X}_t = \mathbf{X}_{t,s}^\varepsilon\right] + \varepsilon_{t,s}\mathrm{d}\mathbf{B}_t, \quad \mathbf{X}_{s,s}^\varepsilon \sim \pi_s,$$

*with $\gamma_{t,s} = \sqrt{\gamma_t^2 - \alpha_t^2\gamma_s^2/\alpha_s^2}$. Then for any $t \in [s, 1]$, $\mathbf{X}_{t,s}^\varepsilon$ and $\mathbf{X}_t$ defined by (16) have the same distribution.*

*Proof.* We let $s \in [0, 1]$ and $\mathbf{X}_1, \mathbf{X}_s \in \mathbb{R}^d$. From (16), we have directly that for any $t \in [s, 1]$

$$\mathrm{d}\mathbf{X}_t = [(\dot{\alpha}_t/\alpha_s)\mathbf{X}_s + (\dot{\beta}_t - \dot{\alpha}_t\beta_s/\alpha_s)\mathbf{X}_1 + \dot{\gamma}_{t,s}\mathbf{Z}]\mathrm{d}t,$$

where $\mathbf{Z} \sim \mathcal{N}(0, \mathrm{Id})$. In addition, rearranging (16), we also have that

$$\mathbf{X}_s = (\alpha_s/\alpha_t)\mathbf{X}_t - (\alpha_s\beta_t/\alpha_t - \beta_s)\mathbf{X}_1 - \gamma_{t,s}(\alpha_s/\alpha_t)\mathbf{Z}.$$

Hence, by combining these two expressions, we get that

$$\mathrm{d}\mathbf{X}_t = [(\dot{\alpha}_t/\alpha_t)\mathbf{X}_t + (\dot{\beta}_t - \dot{\alpha}_t\beta_t/\alpha_s)\mathbf{X}_1 + (\dot{\gamma}_{t,s} - \gamma_{t,s}(\dot{\alpha}_t/\alpha_t))\mathbf{Z}]\mathrm{d}t.$$

It follows that $(\mathbf{X}_{t,s})_{t\in[s,1]}$ given by

$$\mathrm{d}\mathbf{X}_{t,s} = (\dot{\alpha}_t/\alpha_t)\mathbf{X}_{t,s} + (\dot{\beta}_t - \beta_t\dot{\alpha}_t/\alpha_t)\mathbb{E}[\mathbf{X}_1 \mid \mathbf{X}_t = \mathbf{X}_{t,s}]$$

$$+ (\dot{\gamma}_{t,s} - \gamma_{t,s}\dot{\alpha}_t/\alpha_t)\mathbb{E}[\mathbf{Z} \mid \mathbf{X}_t = \mathbf{X}_{t,s}], \quad \mathbf{X}_{s,s} \sim \pi_s,$$

is such that for any $t \in [s, 1]$ the same distribution as $\mathbf{X}_t$ defined by (16). Then, we conclude similarly to (Albergo et al., 2023, Theorem 2.8, Corollary 2.10). $\square$

We now consider the following approximate version of (17)

$$d\hat{\mathbf{X}}_{t,s}^{\varepsilon} = (\dot{\alpha}_t/\alpha_t)\hat{\mathbf{X}}_{t,s}^{\varepsilon} + (\dot{\beta}_t - \beta_t\dot{\alpha}_t/\alpha_t)\mathbb{E}\left[\mathbf{X}_1 \mid \mathbf{X}_t = \hat{\mathbf{X}}_{t,s}^{\varepsilon}\right]$$

$$+ (\dot{\gamma}_{t,s} - \gamma_{t,s}\dot{\alpha}_t/\alpha_t - \varepsilon_{t,s}^2/(2\gamma_{t,s}))\hat{\mathbf{Z}}(t,\mathbf{X}_{t,s}^{\varepsilon}) + \varepsilon_{t,s}d\mathbf{B}_t, \quad \mathbf{X}_{s,s}^{\varepsilon} \sim \pi_s,$$

Similarly to Proposition C.1 we consider the best choice of $\varepsilon$ to minimise the interpolation cost.

---

**Proposition C.3 (Optimality and stochastic interpolant):** *Let $s \in [0,1]$. Denote $\mathbb{P}^\varepsilon$, respectively $\hat{\mathbb{P}}^\varepsilon$, the path measure associated with $(\mathbf{X}_{t,s}^\varepsilon)_{t\in[0,1]}$, respectively $(\hat{\mathbf{X}}_{t,s})_{t\in[0,1]}$. Consider $\ell(\varepsilon) = \mathrm{KL}(\mathbb{P}^\varepsilon|\hat{\mathbb{P}}^\varepsilon)$. Let $\varepsilon^\star = \mathrm{argmin}_\varepsilon \ell(\varepsilon)$. Then we have*

$$(\varepsilon_t^\star)^2 = 2\gamma_t\dot{\gamma}_t - 2\gamma_t^2\dot{\alpha}_t/\alpha_t.$$

*In particular, $\varepsilon^\star$ does not depend on $s \in [0,1]$ and for every $s_1, s_2 \in [0,1]$ with $s_1 \le s_2$, we have that $(\mathbf{X}_{t,s_1}^\varepsilon)_{t\in[s_2,1]}$ and $(\mathbf{X}_{t,s_2}^\varepsilon)_{t\in[s_2,1]}$ coincide.*

---

*Proof.* Similarly to Proposition C.1, we get first that for any $s,t \in [0,1]$ with $s \le t$

$$\varepsilon_{t,s}^\star = 2\gamma_{t,s}\dot{\gamma}_{t,s} - 2\gamma_{t,s}^2\dot{\alpha}_t/\alpha_t. \tag{18}$$

Second, we have that for any $s,t \in [0,1]$ with $s \le t$

$$2\dot{\gamma}_{t,s}\gamma_{t,s} = \dot{\gamma}_{t,s}^2 = 2\dot{\gamma}_t\gamma_t - 2\dot{\alpha}_t\alpha_t\gamma_s^2/\alpha_s^2. \tag{19}$$

Third, we have that

$$\gamma_{t,s}^2\dot{\alpha}_t/\alpha_t = \gamma_t^2\dot{\alpha}_t/\alpha_t - \dot{\alpha}_t\alpha_t\gamma_s^2/\alpha_s^2. \tag{20}$$

Combining (18), (19) and (20), we can conclude. $\square$

# D  Theoretical results

In this section, we prove the main theoretical results of the paper. In Appendix D.1, we first prove the convergence of the $\alpha$-IMF sequence, i.e. we prove Theorem 3.1. Second, we show that the non-parametric updates (8) correspond to the $\alpha$-IMF sequence, i.e. we prove Proposition 3.2. In Appendix D.2, we link the non-parametric updates to the parametric updates.

## D.1  Non-parametric sequence and convergence

Let $\mathbb{Q} \in \mathcal{P}(\mathcal{C})$ be associated with $(\sqrt{\varepsilon}\mathbf{B}_t)_{t\in[0,1]}$, where $(\mathbf{B}_t)_{t\in[0,1]}$ is a $d$-dimensional Brownian motion and $\varepsilon > 0$. In this section, we abuse notation and denote $\mathcal{P}(\mathcal{C})$ the set of *path measures* which are not necessarily *probability* path measures. In particular, we will consider $\mathbb{Q} \in \mathcal{P}(\mathcal{C})$ associated with $(\sqrt{\varepsilon}\mathbf{B}_t)_{t\in[0,1]}$ with $\mathbb{Q}_0 = \mathrm{Leb}$. In that case, the Kullback–Leibler divergence is still well-defined and we refer to (Léonard, 2014) for more details. We recall that we have defined $(\mathbb{P}^n, \hat{\mathbb{P}}^n)_{n\in\mathbb{N}}$ for any $n \in \mathbb{N}$ and $\alpha \in (0,1]$ by

$$\mathbb{P}^n = \mathrm{proj}_{\mathcal{M}}(\hat{\mathbb{P}}^n), \qquad \hat{\mathbb{P}}^{n+1} = (1-\alpha)\hat{\mathbb{P}}^n + \alpha\mathrm{proj}_{\mathcal{R}}(\mathrm{proj}_{\mathcal{M}}(\hat{\mathbb{P}}^n)).$$

In addition, for any $n \in \mathbb{N}$, $t \in [0,1)$ and $x \in \mathbb{R}^d$ we have defined

$$v_t^{n+1}(x) = v_t^n(x) - \delta_n \nabla_{\mu^n}\mathcal{L}_t(v_t^n, \mathbb{P}_{v^n})(x),$$

where

$$\mathcal{L}_t(v_t, \mathbb{P}) = \frac{1}{2}\int_{(\mathbb{R}^d)^3}\left\|v_t(x_t) - \frac{x_1 - x_t}{1-t}\right\|^2 d\mathbb{P}_{0,1}(x_0, x_1)d\mathbb{Q}_{t|0,1}(x_t|x_0, x_1) \tag{21}$$

$$= \frac{1}{2}\int_{(\mathbb{R}^d)^3}\left\|v_t(x_t) - \frac{x_1 - x_t}{1-t}\right\|^2 d\mathrm{proj}_{\mathcal{R}}(\mathbb{P})_{1,t}.$$

We define $(\mathbb{P}_{v^n})_{n\in\mathbb{N}}$ associated with (27), where for any suitable vector field $v$, $\mathbb{P}_v$ is associated with

$$d\mathbf{X}_t = v_t(\mathbf{X}_t)dt + \sqrt{\varepsilon}d\mathbf{B}_t,$$

where $(\mathbf{B}_t)_{t\in[0,1]}$ is a $d$-dimensional Brownian motion.

In order to rigorously prove Proposition D.1 detailed further, we introduce $\mathcal{P}_2(\mathcal{C})$, such that $\mathbb{P} \in \mathcal{P}_2(\mathcal{C})$ if $\mathbb{P} \in \mathcal{P}(\mathcal{C})$ and for

$$\int_{(\mathbb{R}^d)^2} \{\|x_0\|^2 + \|x_1\|^2\} \mathrm{d}\mathbb{P}_{0,1}(x_0, x_1) < +\infty.$$

Note that if $\mathbb{P} \in \mathcal{P}_2(\mathcal{C})$ then we have that for any $t \in [0,1]$

$$\int_{\mathbb{R}^d} \|x_t\|^2 \mathrm{dproj}_{\mathcal{R}}(\mathbb{P})_t < +\infty.$$

In addition, we recall that $\phi \in \mathrm{L}^2(\mu)$ for $\mu \in \mathcal{P}(\mathbb{R}^d)$ if $\phi : \mathbb{R}^d \to \mathbb{R}^d$ and

$$\int_{\mathbb{R}^d} \|\phi(x)\|^2 \mathrm{d}\mu(x) < +\infty.$$

Finally, we define

$$\mathsf{A}_2 = \{(\phi, \mathbb{P}) \ : \ \mathbb{P} \in \mathcal{P}_2(\mathcal{C}), \ \phi \in \mathrm{L}^2(\mathbb{P})\}.$$

Then for any $t \in [0,1)$, we define $\mathcal{L}_t : \mathsf{A}_2 \to \mathbb{R}$ given for any $(v, \mathbb{P}) \in \mathsf{A}_2$ by (21).

> **Proposition D.1 (Non-parametric updates are $\alpha$-IMF):** *Let $\alpha \in (0,1]$, $(\mathbb{P}^n, \hat{\mathbb{P}}^n)_{n\in\mathbb{N}}$ as in (4), $\delta_n = \alpha$ and $\mu^n = (1-\alpha)\hat{\mathbb{P}}^n + \alpha\mathrm{proj}_{\mathcal{R}}(\mathbb{P}^n)$. Assume that for any $n \in \mathbb{N}$, $\mathbb{P}_{v^n}$ is well-defined. Then, for any $n \in \mathbb{N}$, $\mathbb{P}_{v^n} = \mathbb{P}^n$.*

*Proof.* First, we have that for any $t \in [0,1)$, $v, \mathbb{P} \in \mathsf{A}_2$ and $\phi \in \mathrm{L}^2(\mathbb{P}_t)$ we have

$$\mathcal{L}_t(v_t + \varepsilon\phi, \mathbb{P}) = \mathcal{L}_t(v_t, \mathbb{P}) + \varepsilon \int_{(\mathbb{R}^d)^3} \langle \phi(x_t), v_t(x_t) - \tfrac{x_1 - x_1}{1-t}\rangle \mathrm{d}\mathbb{P}_{0,1}(x_0, x_1)\mathrm{d}\mathbb{Q}_{t|0,1}(x_t|x_0, x_1)$$

$$+ (\varepsilon^2/2)\int_{(\mathbb{R}^d)^3} \|\phi(x_t)\|^2 \mathrm{d}\mathbb{P}_{0,1}(x_0, x_1)\mathrm{d}\mathbb{Q}_{t|0,1}(x_t|x_0, x_1)$$

$$= \mathcal{L}_t(v_t, \mathbb{P}) + \varepsilon \int_{(\mathbb{R}^d)^2} \langle \phi(x_t), v_t(x_t)$$

$$- \left(\int_{\mathbb{R}^d} x_1 \mathrm{dproj}_{\mathcal{R}}(\mathbb{P})_{1|t}(x_1|x_t) - x_t\right)/(1-t)\rangle \mathrm{dproj}_{\mathcal{R}}(\mathbb{P})_t(x_t)$$

$$+ (\varepsilon^2/2)\int_{\mathbb{R}^d} \|\phi(x_t)\|^2 \mathrm{proj}_{\mathcal{R}}(\mathbb{P})_t(x_t).$$

Hence, we have that

$$\nabla_\mu \mathcal{L}_t(v_t, \mathbb{P}_v)(x_t) = (v_t(x_t) - (\mathbb{E}_{\mathrm{proj}_{\mathcal{R}}(\mathbb{P})}[\mathbf{X}_1 \mid \mathbf{X}_t = x_t] - x_t)/(1-t))(\mathrm{dproj}_{\mathcal{R}}(\mathbb{P}_v)_t/\mathrm{d}\mu_t)(x_t). \tag{22}$$

Assume that for some $n \in \mathbb{N}$ we have that for any $t \in [0,1)$ and $x_t \in \mathbb{R}^d$, we have $v_t^k(x_t) = (\mathbb{E}_{\hat{\mathbb{P}}^k}[\mathbf{X}_1 \mid \mathbf{X}_t = x_t] - x_t)/(1-t)$. We are going to show that for any $t \in [0,1)$ and $x_t \in \mathbb{R}^d$, we have $v_t^{n+1}(x_t) = (\mathbb{E}_{\hat{\mathbb{P}}^{n+1}}[\mathbf{X}_1 \mid \mathbf{X}_t = x_t] - x_t)/(1-t)$. For any $t \in [0,1)$ and $x_t \in \mathbb{R}^d$, we denote

$$\bar{\delta}_t^n(x_t) = \delta_n(\mathrm{dproj}_{\mathcal{R}}(\mathbb{P}^n)_t/\mathrm{d}\mu_t^n)(x_t).$$

Since we have that $\delta_n = \alpha$ and $\mu^n = (1-\alpha)\hat{\mathbb{P}}^n + \alpha\mathrm{proj}_{\mathcal{R}}(\mathbb{P}^n)$, we obtain for any $t \in [0,1]$ and $x_t \in \mathbb{R}^d$

$$\bar{\delta}_t^n(x_t) = \alpha(\mathrm{dproj}_{\mathcal{R}}(\mathbb{P}^n)_t/\mathrm{d}((1-\alpha)\hat{\mathbb{P}}_t^n + \alpha\mathrm{proj}_{\mathcal{R}}(\mathbb{P}^n)_t))(x_t), \tag{23}$$

so that

$$1 - \bar{\delta}_t^n(x_t) = (1-\alpha)(\mathrm{d}\hat{\mathbb{P}}_t^n/\mathrm{d}((1-\alpha)\hat{\mathbb{P}}_t^n + \alpha\mathrm{proj}_{\mathcal{R}}(\mathbb{P}^n)_t))(x_t). \tag{24}$$

Therefore, combining (8) with (23), (24), (22), we get that for any $t \in [0, 1)$ and $x_t \in \mathbb{R}^d$

$$
\begin{aligned}
v_t^{n+1}(x_t) &= (1 - \bar{\delta}_t^n(x_t))v_t^n(x_t) \\
&\quad + \bar{\delta}_t^n(x_t)\left(\mathbb{E}_{\mathrm{proj}_{\mathcal{R}}(\mathbb{P}^n)}[\mathbf{X}_1 \mid \mathbf{X}_t = x_t] - x_t\right)/(1 - t) \\
&= (1 - \bar{\delta}_t^n(x_t))\left(\mathbb{E}_{\hat{\mathbb{P}}^n}[\mathbf{X}_1 \mid \mathbf{X}_t = x_t] - x_t\right)/(1 - t) \\
&\quad + \bar{\delta}_t^n(x_t)\left(\mathbb{E}_{\mathrm{proj}_{\mathcal{R}}(\mathbb{P}^n)}[\mathbf{X}_1 \mid \mathbf{X}_t = x_t] - x_t\right)/(1 - t) \\
&= (1 - \bar{\delta}_t^n(x_t))\left(\int_{\mathbb{R}^d} x_1 \mathrm{d}\hat{\mathbb{P}}_{1|t}^n(x_1|x_t) - x_t\right)/(1 - t) \\
&\quad + \bar{\delta}_t^n(x_t)\left(\int_{\mathbb{R}^d} x_1 \mathrm{dproj}_{\mathcal{R}}(\mathbb{P}^n)_{1|t}(x_1|x_t) - x_t\right)/(1 - t) \\
&= \int_{\mathbb{R}^d}(x_1 - x_t)/(1 - t)\mathrm{d}[(1 - \bar{\delta}_t^n(x_t))\hat{\mathbb{P}}_{1|t}^n + \bar{\delta}_t^n(x_t)\mathrm{proj}_{\mathcal{R}}(\mathbb{P}^n)_{1|t}](x_1|x_t) \\
&= \int_{\mathbb{R}^d} x_1 \mathrm{d}[(1 - \alpha)\hat{\mathbb{P}}^n + \alpha \mathrm{proj}_{\mathcal{R}}(\mathbb{P}^n)]_{1|t}(x_1|x_t).
\end{aligned}
$$

Hence, we have that for any $t \in [0, 1)$ and $x_t \in \mathbb{R}^d$, $v_t^{n+1}(x_t) = \left(\mathbb{E}_{\hat{\mathbb{P}}^{n+1}}[\mathbf{X}_1 \mid \mathbf{X}_t = x_t] - x_t\right)/(1 - t)$. Since, for any $t \in [0, 1)$ and $x_t \in \mathbb{R}^d$, $v_t^0(x_t) = \left(\mathbb{E}_{\hat{\mathbb{P}}^0}[\mathbf{X}_1 \mid \mathbf{X}_t = x_t] - x_t\right/(1 - t)$ by definition, we get that for any $n \in \mathbb{N}$, $t \in [0, 1)$ and $x_t \in \mathbb{R}^d$, $v_t^n(x_t) = \left(\mathbb{E}_{\hat{\mathbb{P}}^n}[\mathbf{X}_1 \mid \mathbf{X}_t = x_t] - x_t\right)/(1 - t)$. Using, Definition 2.2, we get that $\mathbb{P}_{v^n} = \mathrm{proj}_{\mathcal{M}}(\hat{\mathbb{P}}^n)$, which concludes the proof. $\qquad\square$

Before stating our convergence theorem, we show the following result which is a direct consequence of (Léonard, 2014, Theorem 2.12) and (Léonard et al., 2014, Theorem 2.14). We recall that the differential entropy of a probability measure $\pi$ is given by

$$
\mathrm{H}(\pi) = -\int_{\mathbb{R}^d} \log((\mathrm{d}\pi/\mathrm{dLeb})(x))\mathrm{d}\pi(x),
$$

if $\pi$ admits a density with respect to the Lebesgue measure and $+\infty$ otherwise.

**Lemma D.2 (Characterisation of Schrödinger Bridge):** *Recall that $\mathbb{Q}$ is associated with $(\sqrt{\varepsilon}\mathbf{B}_t)_{t \in [0,1]}$ and assume that $\mathbb{Q}_0 = \mathrm{Leb}$. Let $\pi_0, \pi_1 \in \mathcal{P}(\mathbb{R}^d)$ such that*

$$
\int_{\mathbb{R}^d} \|x\|^2 \mathrm{d}\pi_i(x) < +\infty, \qquad \mathrm{H}(\pi_i) < +\infty,
$$

*for $i \in \{0, 1\}$. Let $\mathbb{P}^\star$ such that $\mathbb{P}^\star$ is Markov, $\mathbb{P}^\star \in \mathcal{R}(\mathbb{Q})$, $\mathbb{P}_0^\star = \pi_0$ and $\mathbb{P}_1^\star = \pi_1$. Then $\mathbb{P}^\star$ is the Schrödinger Bridge, i.e. the unique solution to (2).*

*Proof.* First, we have that $\mathbb{Q}_{0,1}$ is equivalent to $\mathrm{Leb} \otimes \mathrm{Leb}$. Indeed, we have that for any $x_0, x_1 \in \mathbb{R}^d$

$$
(\mathrm{d}\mathbb{Q}_{0,1}/\mathrm{d}(\mathrm{Leb} \otimes \mathrm{Leb}))(x_0, x_1) = (2\pi\varepsilon)^{-d/2}\exp[-\|x_0 - x_1\|^2/(2\varepsilon)].
$$

Similarly, we have that for any $t \in (0, 1)$ and $x_t \in \mathbb{R}^d$, $\mathbb{Q}_{0,1|t}(\cdot|x_t)$ is equivalent to $\mathrm{Leb} \otimes \mathrm{Leb}$. Indeed, we have that for any $t \in (0, 1)$ and $x_0, x_t, x_1 \in \mathbb{R}^d$

$$
\begin{aligned}
(\mathrm{d}\mathbb{Q}_{0,1|t}(\cdot|x_t)/\mathrm{d}(\mathrm{Leb} \otimes \mathrm{Leb}))(x_0, x_1) &= (2\pi\varepsilon t)^{-d/2}\exp[-\|x_0 - x_t\|^2/(2\varepsilon t)] \\
&\quad \times (2\pi\varepsilon(1 - t))^{-d/2}\exp[-\|x_t - x_1\|^2/(2\varepsilon(1 - t))].
\end{aligned}
$$

Hence, for any $t \in (0, 1)$ and $x_t \in \mathbb{R}^d$, $\mathbb{Q}_{0,1|t}(\cdot|x_t)$ is equivalent to $\mathbb{Q}_{0,1}$. Since $\mathbb{P}^\star$ is Markov and $\mathbb{P}^\star \in \mathcal{R}(\mathbb{Q})$ we get that there exist $\varphi_0^\circ$ and $\varphi_1^\star$ which are Lebesgue measurable such that for any $\omega \in \mathcal{C}$ we have that

$$
(\mathrm{d}\mathbb{P}^\star/\mathrm{d}\mathbb{Q})(\omega) = \varphi_0^\circ(\omega_0)\varphi_1^\star(\omega_1). \tag{25}
$$

Second we verify that the conditions (i)-(vii) of (Léonard, 2014, Theorem 2.12) are satisfied. First, $\mathbb{Q}$ is Markov and hence (i) is satisfied. Then, (ii) is satisfied since for any $t \in (0, 1)$ and $x_t \in \mathbb{R}^d$,

$\mathbb{Q}_{0,1|t}(\cdot|x_t)$ is equivalent to $\mathbb{Q}_{0,1}$. We have that $\mathbb{Q}_0 = \mathbb{Q}_1 = \text{Leb}$ and (iii) is satisfied. We have that for any $x_0, x_1 \in \mathbb{R}^d$

$$(\mathrm{d}\mathbb{Q}_{0,1}/\mathrm{d}(\text{Leb} \otimes \text{Leb}))(x_0, x_1) = (2\pi\varepsilon)^{-d/2} \exp[-\|x_0 - x_1\|^2/(2\varepsilon)]$$
$$\geq (2\pi\varepsilon)^{-d/2} \exp[-\|x_0\|^2/\varepsilon - \|x_1\|^2/\varepsilon].$$

Hence, (iv) is satisfied and we let $A : \mathbb{R}^d \to \mathbb{R}_+$ be given for any $x \in \mathbb{R}^d$ by $A(x) = \|x\|^2/\varepsilon$. In addition, we have that for any $x_0, x_1 \in \mathbb{R}^d$

$$\int_{(\mathbb{R}^d)^2} \exp[-\|x_0\|^2/\varepsilon - \|x_1\|^2/\varepsilon]\mathrm{d}\mathbb{Q}_{0,1}(x_0, x_1) < +\infty.$$

Hence, (v) is satisfied and we let $B : \mathbb{R}^d \to \mathbb{R}_+$ given for any $x \in \mathbb{R}^d$ by $B(x) = \|x\|^2/\varepsilon$. By assumption (vi) and (vii) are satisfied. We conclude the proof upon using (Léonard, 2014, Theorem 2.12-(b)) and (25). $\qquad\square$

We are now ready to state our main convergence result.

> **Proposition D.3 (Convergence of $\alpha$-IMF):** *Let* $\alpha \in (0, 1]$ *and* $(\mathbb{P}^n, \hat{\mathbb{P}}^n)_{n\in\mathbb{N}}$ *defined by* (4). *Under mild assumptions, we have that* $\lim_{n\to+\infty} \mathbb{P}^n = \mathbb{P}^\star$, *where* $\mathbb{P}^\star$ *is the solution of the Schrödinger Bridge problem* (2).

*Proof.* Using the convexity of the Kullback–Leibler divergence with respect to its first argument (see e.g. (Dupuis and Ellis, 2011)), the data processing inequality (see e.g. (Ambrosio et al., 2008, Lemma 9.4.5)), the fact that the Schrödinger Bridge is Markov and in the reciprocal class of $\mathbb{Q}$ (see e.g. (Léonard, 2014, Theorem 2.12) and (Léonard et al., 2014, Theorem 3.2)), and the Pythagorean theorem for the Markovian projection (Shi et al., 2023, Lemma 6), we have that for any $n \in \mathbb{N}$

$$\begin{aligned}
\mathrm{KL}(\hat{\mathbb{P}}^{n+1}|\mathbb{P}^\star) &= \mathrm{KL}((1-\alpha)\hat{\mathbb{P}}^n + \alpha\mathrm{proj}_{\mathcal{R}}(\mathrm{proj}_{\mathcal{M}}(\hat{\mathbb{P}}^n))|\mathbb{P}^\star) \\
&\leq (1-\alpha)\mathrm{KL}(\hat{\mathbb{P}}^n|\mathbb{P}^\star) + \alpha\mathrm{KL}(\mathrm{proj}_{\mathcal{R}}(\mathrm{proj}_{\mathcal{M}}(\hat{\mathbb{P}}^n))|\mathbb{P}^\star) \\
&\leq (1-\alpha)\mathrm{KL}(\hat{\mathbb{P}}^n|\mathbb{P}^\star) + \alpha\mathrm{KL}(\mathrm{proj}_{\mathcal{M}}(\hat{\mathbb{P}}^n)_{0,1}|\mathbb{P}^\star_{0,1}) \\
&\leq (1-\alpha)\mathrm{KL}(\hat{\mathbb{P}}^n|\mathbb{P}^\star) + \alpha\mathrm{KL}(\mathrm{proj}_{\mathcal{M}}(\hat{\mathbb{P}}^n)|\mathbb{P}^\star) \\
&\leq (1-\alpha)\mathrm{KL}(\hat{\mathbb{P}}^n|\mathbb{P}^\star) + \alpha\mathrm{KL}(\hat{\mathbb{P}}^n|\mathbb{P}^\star) - \alpha\mathrm{KL}(\hat{\mathbb{P}}^n|\mathrm{proj}_{\mathcal{M}}(\hat{\mathbb{P}}^n)). \quad (26)
\end{aligned}$$

Therefore, we get that

$$\alpha\mathrm{KL}(\hat{\mathbb{P}}^n|\mathrm{proj}_{\mathcal{M}}(\hat{\mathbb{P}}^n)) \leq \mathrm{KL}(\hat{\mathbb{P}}^n|\mathbb{P}^\star) - \mathrm{KL}(\hat{\mathbb{P}}^{n+1}|\mathbb{P}^\star).$$

Hence, it follows that

$$\sum_{n\in\mathbb{N}} \mathrm{KL}(\hat{\mathbb{P}}^n|\mathrm{proj}_{\mathcal{M}}(\hat{\mathbb{P}}^n)) \leq 2\mathrm{KL}(\hat{\mathbb{P}}^0|\mathbb{P}^\star) < +\infty.$$

So we obtain $\lim_{n\to+\infty} \mathrm{KL}(\hat{\mathbb{P}}^n|\mathrm{proj}_{\mathcal{M}}(\hat{\mathbb{P}}^n)) = 0$. In addition, using (26) we have that $\mathrm{KL}(\hat{\mathbb{P}}^n|\mathbb{P}^\star) \leq \mathrm{KL}(\hat{\mathbb{P}}^0|\mathbb{P}^\star) < +\infty$ for all $n \in \mathbb{N}$. Using (Shi et al., 2023, Lemma 6), we also get that $\mathrm{KL}(\mathrm{proj}_{\mathcal{M}}(\hat{\mathbb{P}}^n)|\mathbb{P}^\star) \leq \mathrm{KL}(\hat{\mathbb{P}}^0|\mathbb{P}^\star) < +\infty$ for any $n \in \mathbb{N}$. Hence both the sequences $(\hat{\mathbb{P}}^n)_{n\in\mathbb{N}}$ and $(\mathbb{P}^n)_{n\in\mathbb{N}} = (\mathrm{proj}_{\mathcal{M}}(\hat{\mathbb{P}}^n))_{n\in\mathbb{N}}$ are relatively compact in $\mathcal{P}(\mathcal{C})$. Let $\bar{\mathbb{P}} \in \mathcal{P}(\mathcal{C})$ be an adherent point to the sequence $(\hat{\mathbb{P}}^n)_{n\in\mathbb{N}}$ and $\varphi : \mathbb{N} \to \mathbb{N}$ increasing such that $\lim_{n\to+\infty} \mathbb{P}^{\varphi(n)} = \bar{\mathbb{P}}$. Similarly, let $\phi : \mathbb{N} \to \mathbb{N}$ increasing such that $(\phi(n))_{n\in\mathbb{N}}$ is a subsequence of $(\varphi(n))_{n\in\mathbb{N}}$ such that $\lim_{n\to+\infty} \mathrm{proj}_{\mathcal{M}}(\hat{\mathbb{P}}^n) = \bar{\mathbb{P}}'$, with $\bar{\mathbb{P}}'$ and adherent point to the sequence $(\mathrm{proj}_{\mathcal{M}}(\hat{\mathbb{P}}^n))_{n\in\mathbb{N}}$. Using the lower semi-continuity of the Kullback-Leibler divergence in both arguments (Dupuis and Ellis, 2011), we get that

$$\mathrm{KL}(\bar{\mathbb{P}}|\bar{\mathbb{P}}') \leq \liminf_{n\to+\infty} \mathrm{KL}(\hat{\mathbb{P}}^{\phi(n)}|\mathrm{proj}_{\mathcal{M}}(\hat{\mathbb{P}}^{\phi(n)})) = 0.$$

Since the set of Markov measures and the set of reciprocal measures w.r.t. $\mathbb{Q}$ are both closed, we have that $\bar{\mathbb{P}}$ is Markov and in the reciprocal class of $\mathbb{Q}$. Since we also have that $\bar{\mathbb{P}}_0 = \pi_0$ and $\bar{\mathbb{P}}_1 = \pi_1$, we get that $\bar{\mathbb{P}} = \mathbb{P}^\star$ using Appendix D.1. Since every adherent point of $(\hat{\mathbb{P}}^n)_{n\in\mathbb{N}}$ is $\mathbb{P}^\star$, we have that $\lim_{n\to+\infty} \hat{\mathbb{P}}^n = \mathbb{P}^\star$. Similarly, using that $\lim_{n\to+\infty} \mathrm{KL}(\hat{\mathbb{P}}^n|\mathrm{proj}_{\mathcal{M}}(\hat{\mathbb{P}}^n)) = 0$ and again the lower semi-continuity of the Kullback–Leibler divergence in both arguments, we get that every adherent point of $(\mathrm{proj}_{\mathcal{M}}(\hat{\mathbb{P}}^n))_{n\in\mathbb{N}}$ is $\mathbb{P}^\star$. Hence, we have that $\lim_{n\to+\infty} \mathbb{P}^n = \mathbb{P}^\star$, which concludes the proof. $\qquad\square$

## D.2 From parametric to non-parametric.

In this section, we show that the parametric updates considered in (9) are a preconditioned version of the non-parametric updates considered in (8). We first recall the non-parametric loss

$$\mathcal{L}(v, \mathbb{P}) = \int_0^1 \mathcal{L}_t(v_t, \mathbb{P}) \mathrm{d}t = \frac{1}{2} \int_0^1 \int_{(\mathbb{R}^d)^2} \left\| v_t(x_t) - \frac{x_1 - x_t}{1 - t} \right\|^2 \mathrm{dproj}_{\mathcal{R}}(\mathbb{P})_{t,1}(x_t, x_1) \mathrm{d}t$$

and the parametric loss

$$\mathrm{L}(\theta, \mathbb{P}) = \frac{1}{2} \int_0^1 \int_{\mathbb{R}^d \times \mathbb{R}^d} \left\| v_t^\theta(x_t) - \frac{x_1 - x_t}{1 - t} \right\|^2 \mathrm{dproj}_{\mathcal{R}(\mathbb{Q})}(\mathbb{P})_{t,1}(x_t, x_1) \mathrm{d}t.$$

The non-parametric sequence $(v^n)_{n \in \mathbb{N}}$ is given by (27), i.e. we have for any $n \in \mathbb{N}$, $t \in [0, 1]$ and $x \in \mathbb{R}^d$

$$v_t^{n+1}(x) = v_t^n(x) - \delta_n \nabla_{\mu^n} \mathcal{L}_t(v_t^n, \mathbb{P}_{v^n})(x). \tag{27}$$

Similarly the sequence of parametric updates is given for any $n \in \mathbb{N}$, $t \in [0, 1]$ and $x \in \mathbb{R}^d$ by

$$\theta_{n+1} = \theta_n - \alpha \nabla_\theta \mathrm{L}(\theta_n, \mathbb{P}^{\bar{\theta}_n}).$$

We recall that $\mathbb{P}^{\bar{\theta}_n}$ is a stop gradient version of $\mathbb{P}_{v^{\bar{\theta}_n}}$. We are going to show that on average the parametric algorithm yields direction of descent for the non-parametric loss. We assume that the set of parameters $\Theta$ is an open subset of $\mathbb{R}^p$ for some $p \in \mathbb{N}$. For any $t \in [0, 1]$ and $x \in \mathbb{R}^d$ we assume that $\theta \mapsto v_t^\theta(x)$ is twice continuously differentiable and denote $\mathrm{D}_\theta v_t^\theta(x) \in \mathbb{R}^{d \times p}$ its Jacobian and $\mathrm{D}_\theta^2 v_t^\theta(x)$ its Hessian. For any $\theta \in \Theta$, we denote

$$h_\theta = \nabla_\theta \mathrm{L}(\theta, \mathbb{P}^{\bar{\theta}}).$$

We show the following result.

**Proposition D.4 (Velocity field parametric update):** *Assume that there exists $C > 0$ such that for any $\theta \in \Theta$ and $x \in \mathbb{R}^d$*

$$\int_0^1 (1 - s) \mathrm{D}_\theta^2 v_s^{\theta - \alpha s h_\theta}(x)(h_\theta, h_\theta) \mathrm{d}s < C, \tag{28}$$

*where $h_\theta = \nabla_\theta \mathrm{L}(\theta, \mathbb{P}^{\bar{\theta}})$. We have that for any $n \in \mathbb{N}$, $t \in [0, 1)$ and $x \in \mathbb{R}^d$*

$$v_t^{\theta_{n+1}}(x) = v_t^{\theta_n}(x)$$

$$- \alpha \mathrm{D}_\theta v_t^{\theta_n}(x) \int_0^1 \int_{\mathbb{R}^d} \mathrm{D}_\theta v_s^\theta(\tilde{x})^\top \nabla_{\mu^n} \mathcal{L}_s(v^{\theta_n}, \mathbb{P}_{v^{\theta_n}})(\tilde{x}) \mathrm{d}\mu_s^n(\tilde{x}) \mathrm{d}s + o(\alpha),$$

*where $\mu^n = (1 - \alpha)\mathbb{P}^n + \alpha \mathbb{P}_{v^{\theta_n}}$.*

*Proof.* First, we have that for any $\mu \in \mathcal{P}(\mathcal{C})$

$$\nabla_\theta \mathrm{L}(\theta, \mathbb{P}^{\bar{\theta}}) = \int_0^1 \int_{\mathbb{R}^d} \mathrm{D}_\theta v_s^\theta(x_s)^\top (v_s^\theta(x_s) - (x_1 - x_s)/(1 - s)) \mathrm{dproj}_{\mathcal{R}}(\mathbb{P}^{\bar{\theta}})_{s,1}(x_s, x_1) \mathrm{d}s.$$

Therefore, using (22), we get that

$$\nabla_\theta \mathrm{L}(\theta, \mathbb{P}^{\bar{\theta}}) = \int_0^1 \int_{\mathbb{R}^d} \mathrm{D}_\theta v_s^\theta(x_s)^\top \nabla_\mu \mathcal{L}_s(v_s^\theta, \mathbb{P}^{\bar{\theta}})(x_s) \mathrm{d}\mu_s(x_s) \mathrm{d}s.$$

Let $\theta \in \Theta$ and denote $\theta' = \theta - \alpha \nabla_\theta \mathrm{L}(\theta, \mathbb{P}^{\bar{\theta}})$. Using a Taylor expansion, we get that for any $\theta \in \Theta$, we have that

$$v_t^{\theta'}(x) = v_t^\theta(x) - \alpha \mathrm{D}_\theta v_t^\theta(x) \int_0^1 \int_{\mathbb{R}^d} \mathrm{D}_\theta v_s^\theta(x_s)^\top \nabla_\mu \mathcal{L}_s(v_s^\theta, \mathbb{P}^{\bar{\theta}})(x_s) \mathrm{d}\mu_s(x_s) \mathrm{d}s$$

$$+ \alpha^2 \int_0^1 (1 - s) \mathrm{D}_\theta^2 v_s^{\theta - \alpha s h_\theta}(x)(h_\theta, h_\theta) \mathrm{d}s.$$

Since the functional gradient is not applied on the second coordinate, we can drop the stop gradient operator and therefore we have for any $\theta \in \Theta$

$$v_t^{\theta'}(x) = v_t^\theta(x) - \alpha D_\theta v_t^\theta(x) \int_0^1 \int_{\mathbb{R}^d} D_\theta v_s^\theta(x_s)^\top \nabla_\mu \mathcal{L}_s(v_s^\theta, \mathbb{P}_{v^\theta})(x_s) \mathrm{d}\mu_s(x_s)\mathrm{d}s$$

$$+ \alpha^2 \int_0^1 (1-s) D_\theta^2 v_s^{\theta - \alpha s h_\theta}(x)(h_\theta, h_\theta)\mathrm{d}s.$$

Combining this result with (28), we conclude the proof.

$\square$

The corresponding update on the velocity field is given for any $n \in \mathbb{N}$, $t \in [0,1]$ and $x \in \mathbb{R}^d$ by

$$d_t^n(x) = -\alpha D_\theta v_t^{\theta_n}(x) \int_0^1 \int_{\mathbb{R}^d} D_\theta v_s^\theta(\tilde{x})^\top \nabla_{\mu^n} \mathcal{L}_s(v^{\theta_n}, \mathbb{P}_{v^{\theta_n}})(\tilde{x}) \mathrm{d}\mu_s^n(\tilde{x}) + o(\alpha).$$

We immediately have the following corollary.

**Proposition D.5 (Parametric direction of descent):** *For any $n \in \mathbb{N}$, if*

$$\int_0^1 \int_{\mathbb{R}^d} D_\theta v_s^\theta(\tilde{x})^\top \nabla_{\mu^n} \mathcal{L}_s(v^{\theta_n}, \mathbb{P}_{v^{\theta_n}})(\tilde{x}) \mathrm{d}\mu_s^n(\tilde{x}) \neq 0,$$

*then we have*

$$\lim_{\alpha \to 0} \int_0^1 \int_{\mathbb{R}^d} \langle \nabla_{\mu^n} \mathcal{L}_t(v^{\theta_n}, \mathbb{P}_{v^{\theta_n}})(x), d_t^n(x) \rangle \mathrm{d}\mu_t^n(x) \leq 0.$$

## E    Background material on DSBM and extensions

In this section we recall some basics on Markovian and reciprocal projections in Appendix E.1. We explain the link between the concept of *iterative refinement* and Schrödinger Bridges in Appendix E.2. Then, we briefly present Diffusion Schrödinger Bridge Matching (DSBM) (Shi et al., 2023) in Appendix E.3 and propose some new extensions in Appendix E.4.

### E.1    Markov and reciprocal projections in practice

In this section, we recall the definition of the reciprocal and Markov projection. We provide more details on how these different projections can be performed and illustrate them on simple examples.

**Markov projection.**    First, we recall the definition of the Markovian projection.

**Definition E.1 (Markov projection):** *Assume that $\mathbb{Q}$ is induced by $(\sqrt{\varepsilon}\mathbf{B}_t)_{t \in [0,1]}$ for $\varepsilon > 0$. Then, when it is well-defined, for any $\mathbb{P} \in \mathcal{R}(\mathbb{Q})$, the Markovian projection $\mathbb{M} = \mathrm{proj}_\mathcal{M}(\mathbb{P}) \in \mathcal{M}$ is the path measure induced by the diffusion*

$$\mathrm{d}\mathbf{X}_t^\star = v_t^\star(\mathbf{X}_t^\star)\mathrm{d}t + \sqrt{\varepsilon}\mathrm{d}\mathbf{B}_t, \qquad v_t^\star(x_t) = \left(\mathbb{E}_{\mathbb{P}_{1|t}}[\mathbf{X}_1 \mid \mathbf{X}_t = x_t] - x_t\right)/(1-t), \qquad \mathbf{X}_0^\star \sim \mathbb{P}_0.$$

In Figure 7 and Figure 8, we illustrate the effect of the Markovian projection, following the example of (Liu, 2022). We consider two distributions $\pi_0$ and $\pi_1$ such that

$$\pi_0 = \frac{1}{2}\mathcal{N}([-2,-2], \mathrm{Id}) + \frac{1}{2}\mathcal{N}([-2,2], \mathrm{Id}), \quad \pi_1 = \frac{1}{2}\mathcal{N}([2,-2], \mathrm{Id}) + \frac{1}{2}\mathcal{N}([2,2], \mathrm{Id}).$$

In Figure 7, we display samples from the distributions $\pi_0$ and $\pi_1$ as well as trajectories from the path measure $\mathbb{P} = (\pi_0 \otimes \pi_1)\mathbb{Q}_{|0,1}$. Practically, this means that we sample $\mathbf{X}_0 \sim \pi_0$ and $\mathbf{X}_1 \sim \pi_1$ independently and then consider a Brownian bridge between $\mathbf{X}_0$ and $\mathbf{X}_1$. The SDE associated with the Brownian bridge with scale $\varepsilon > 0$ is given for any $t \in [0,1]$ by

$$\mathrm{d}\mathbf{X}_t = (\mathbf{X}_1 - \mathbf{X}_t)/(1-t)\mathrm{d}t + \sqrt{\varepsilon}\mathrm{d}\mathbf{B}_t. \tag{29}$$

Note that the measure $\mathbb{P} = (\pi_0 \otimes \pi_1)\mathbb{Q}_{|0,1}$ is in the reciprocal class, i.e. $\mathbb{P} \in \mathcal{R}(\mathbb{Q})$.

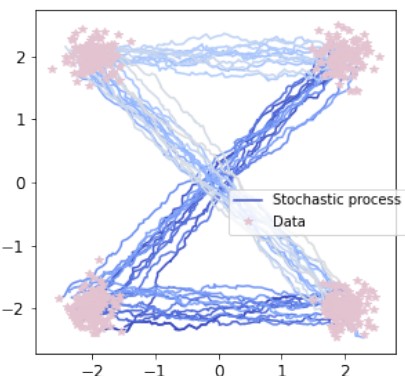

Figure 7: Samples from the original distributions $\pi_0$ (left) and $\pi_1$ (right) are shown in red, while sample paths from $\mathbb{P} = (\pi_0 \otimes \pi_1)\mathbb{Q}_{|0,1}$ are shown in blue.

Next in Figure 8, we display samples from the distributions $\pi_0$ and $\pi_1$ as well as trajectories from the path measure $\mathbb{P}^\star = \mathrm{proj}_{\mathcal{M}}(\mathbb{P})$. Note that in Figure 8, contrary to Figure 7, we observe less crossings between the trajectories. Indeed in the limit case where $\varepsilon \to 0$ the Markov measures $\mathbb{P}^\star$ is an ODE with regular coefficients and therefore admits a unique solution for every starting point in the space so no crossing is possible. In particular, note that most of the trajectories starting from the upper-left Gaussian end at the upper-right Gaussian. Similarly, most of the trajectories starting from the lower-left Gaussian end at the lower-right Gaussian.

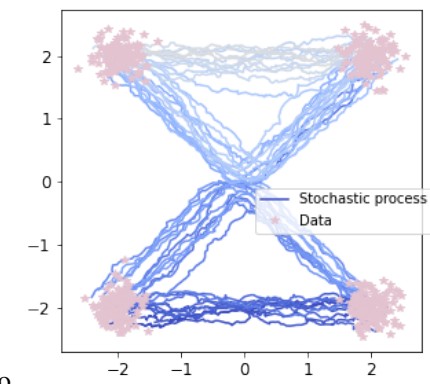

o

Figure 8: Samples from the original distributions are shown in red, while sample paths from $\mathbb{M} = \mathrm{proj}_{\mathcal{M}}(\mathbb{P})$ are shown in blue.

In practice, computing the Markov projection involves finding the optimal drift $v_t^\star$. This optimal drift is the minimizer of a regression problem, see (Shi et al., 2023) for more details. Hence, computing the Markovian projection requires training a neural network to define a vector field.

**Reciprocal projection.** First, we recall the definition the reciprocal projection.

> **Definition E.2 (Reciprocal projection):** $\mathbb{P} \in \mathcal{P}(\mathcal{C})$ *is in the* reciprocal class $\mathcal{R}(\mathbb{Q})$ *of* $\mathbb{Q}$ *if* $\mathbb{P} = \mathbb{P}_{0,1}\mathbb{Q}_{|0,1}$. *We define the* reciprocal projection *of* $\mathbb{P} \in \mathcal{P}(\mathcal{C})$ *as* $\mathbb{P}^\star = \mathrm{proj}_{\mathcal{R}(\mathbb{Q})}(\mathbb{P}) = \mathbb{P}_{0,1}\mathbb{Q}_{|0,1}$. *We will write* $\mathrm{proj}_{\mathcal{R}}$ *instead of* $\mathrm{proj}_{\mathcal{R}(\mathbb{Q})}$ *to simplify notation.*

To sample from $\mathbb{P}^\star = \mathrm{proj}_{\mathcal{R}}(\mathbb{P})$, we only need to sample $(\mathbf{X}_0, \mathbf{X}_1) \sim \mathbb{P}_{0,1}$ and then to sample from the Brownian bridge conditioned on $(\mathbf{X}_0, \mathbf{X}_1)$. This means that in order to sample $\mathbf{X}_t^\star \sim \mathbb{P}_t^\star$, we only need to sample $(\mathbf{X}_0, \mathbf{X}_1) \sim \mathbb{P}_{0,1}$ and then compute

$$\mathbf{X}_t = (1-t)\mathbf{X}_0 + t\mathbf{X}_1 + \sqrt{\varepsilon t(1-t)}\mathbf{Z}, \tag{30}$$

with $\mathbf{Z} \sim \mathcal{N}(0, \mathrm{Id})$. In particular, sampling from $\mathbb{P}^\star = \mathrm{proj}_\mathcal{R}(\mathbb{P})$ does *not* require training any neural network. However, in practice, in order to obtain samples $(\mathbf{X}_0, \mathbf{X}_1) \sim \mathbb{P}$, we have that $\mathbb{P}$ is associated with an SDE and therefore obtaining $(\mathbf{X}_0, \mathbf{X}_1)$ requires unrolling the SDE associated with $\mathbb{P}$. In Algorithm 1, the measure $\mathbb{P}$ is associated with an SDE with parametric drift $v^\theta$.

In Figure 9, we continue our study of the example of (Liu, 2022) that we used to explain the concept of Markovian projection. We consider the path measure $\mathbb{M}$ obtained as the Markov projection of $\mathbb{P} = (\pi_0 \otimes \pi_1)\mathbb{Q}_{|0,1}$. In Figure 9, we display samples from the distributions $\pi_0$ and $\pi_1$ as well as trajectories from the path measure $\mathbb{P}^\star = \mathbb{M}_{0,1}\mathbb{Q}_{|0,1}$. In order to sample from $\mathbb{P}^\star$ we first sample $(\mathbf{X}_0, \mathbf{X}_1) \sim \mathbb{M}_{0,1}$. This involves unrolling the SDE associated with $\mathbb{M}$. Once we have access to samples $(\mathbf{X}_0, \mathbf{X}_1)$, we draw trajectories from the Brownian bridge following the SDE (29). We can also sample from any time $t$ without having to unroll the SDE (29) by simply sampling from (30). This is what is done in Algorithm 1.

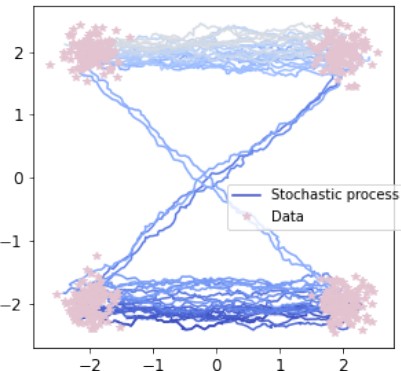

Figure 9: Samples from the original distributions are shown in red, while sample paths from $\mathbb{P}^\star = \mathrm{proj}_\mathcal{R}(\mathbb{M})$ are shown in blue.

## E.2 Iterative refinement and Schrödinger Bridge

The Schrödinger Bridge problem (2) can be solved leveraging techniques from diffusion models and bridge matching. De Bortoli et al. (2021); Vargas et al. (2021) consider an alternating projection algorithm, corresponding to a dynamic version of the celebrated Sinkhorn algorithm. Peluchetti (2023); Shi et al. (2023) introduce the Iterative Markovian Fitting procedure which corresponds to perform an alternating projection algorithm on the class of Markov processes and the reciprocal class of the Brownian motion. It can be shown that the solution of this iterative algorithm converges to the Schrödinger Bridge under mild assumptions, see (Peluchetti, 2023; Shi et al., 2023). We highlight that in the case where $\varepsilon \to 0$ then DSBM is equivalent to the Rectified Flow algorithm (Liu et al., 2023b). One of the main limitation of those previously introduced procedures which provably converge to the solution of the Schrödinger Bridge problem is that they rely on these expensive iterative solvers and requires to consider two networks, one parameterising the forward process $\pi_0 \to \pi_1$ and one parameterising the backward $\pi_1 \to \pi_0$.

## E.3 Diffusion Schrödinger Bridge Matching

Diffusion Schrödinger Bridge Matching corresponds to the practical implementation of the Iterative Markovian Fitting procedure proposed in Shi et al. (2023); Peluchetti (2023). The IMF procedure alternates between projecting on the Markov class $\mathcal{M}$ and the reciprocal class $\mathcal{R}_\mathbb{Q}$. In what follows, we denote $\mathbb{M}^{n+1} = \mathbb{P}^{2n+1} \in \mathcal{M}$ and $\Pi^n = \mathbb{P}^{2n} \in \mathcal{R}(\mathbb{Q})$. We also recall that $\mathbb{Q}$ is a (rescaled) Brownian motion associated with $(\sigma_0 \mathbf{B}_t)_{t \in [0,1]}$ and that therefore sampling from $\mathbb{Q}_{|0,1}(\cdot|x_0, x_1)$ corresponds to sampling from

$$\mathrm{d}\mathbf{X}_t = (x_1 - \mathbf{X}_t)/(1-t)\mathrm{d}t + \sigma_0 \mathrm{d}\mathbf{B}_t, \qquad \mathbf{X}_0 = x_0.$$

We recall that the main computational bottleneck of the DSBM lies in the approximation of the Markovian projection. Indeed, using (Shi et al., 2023, Definition 1, Proposition 2), we have that

$\mathbb{M}^\star = \text{proj}_\mathcal{M}(\Pi)$ is associated with the process

$$d\mathbf{X}_t = (\mathbb{E}_\Pi[\mathbf{X}_1 \mid \mathbf{X}_t] - \mathbf{X}_t)/(1-t)dt + \sigma_0 d\mathbf{B}_t \qquad \mathbf{X}_0 \sim \pi_0.$$

We also have using (Shi et al., 2023, Proposition 2) that $\mathbb{M}^\star$ can be approximated using $\mathbb{M}^{\theta^\star}$ given by

$$d\mathbf{X}_t = v_{\theta^\star}(t, \mathbf{X}_t)dt + \sigma_0 d\mathbf{B}_t, \qquad \mathbf{X}_0 \sim \pi_0, \tag{31}$$

$$\theta^\star = \text{argmin}_{\theta \in \Theta} \int_0^1 \mathbb{E}_{\Pi_{t,1}}[\|(\mathbf{X}_1 - \mathbf{X}_t)/(1-t) - v_\theta(t, \mathbf{X}_t)\|^2]dt, \tag{32}$$

where $\{v_\theta \ : \ \theta \in \Theta\}$ is a parametric family of functions, usually given by a neural network.

Hence, since we can approximate $\text{proj}_{\mathcal{R}(\mathbb{Q})}(\mathbb{M})$ and $\text{proj}_\mathcal{M}(\Pi)$ we can approximate the IMF procedure. This is the DSBM algorithm introduced in Shi et al. (2023); Peluchetti (2023). We describe the first few iterations. Let $\Pi^0 = \Pi^0_{0,1}\mathbb{Q}_{|0,1}$ where $\Pi^0_0 = \pi_0$, $\Pi^0_1 = \pi_1$. Learn $\mathbb{M}^1 \approx \text{proj}_\mathcal{M}(\Pi^0)$ given by (31) with $v_{\theta^\star}$ given by (32). Next, sample from $\Pi^1 = \text{proj}_{\mathcal{R}(\mathbb{Q})}(\mathbb{M}^1) = \mathbb{M}^1_{0,1}\mathbb{Q}_{|0,1}$ by sampling from $\mathbb{M}^1_{0,1}$ and reconstructing the bridge $\mathbb{Q}_{|0,1}$. Upon iterating the previous procedure, we obtain a sequence $(\Pi^n, \mathbb{M}^{n+1})_{n \in \mathbb{N}}$. To mitigate the bias accumulation problem caused by approximating *only* the forward process, we alternate between a *forward* Markovian projection and a *backward* Markovian projection. We give more details on the advantage of using a forward-backward parameterisation instead of a forward-forward in Appendix I. This procedure is valid using (Shi et al., 2023, Proposition 9). The optimal backward process is approximated with

$$d\mathbf{Y}_t = v_{\phi^\star}(1-t, \mathbf{Y}_t)dt + \sigma_0 d\mathbf{B}_t, \qquad \mathbf{Y}_0 \sim \pi_1, \tag{33}$$

$$\phi^\star = \text{argmin}_{\phi \in \Phi} \int_0^1 \mathbb{E}_{\Pi_{0,t}}[\|(\mathbf{X}_0 - \mathbf{X}_t)/t - v_\phi(t, \mathbf{X}_t)\|^2]dt. \tag{34}$$

We recall the full DSBM algorithm in Algorithm 2.

---

**Algorithm 2** Diffusion Schrödinger Bridge Matching

1: **Input:** Joint distribution $\Pi^0_{0,1}$, tractable bridge $\mathbb{Q}_{|0,1}$, number of outer iterations $N \in \mathbb{N}$.
2: Let $\Pi^0 = \Pi^0_{0,1}\mathbb{Q}_{|0,1}$.
3: **for** $n \in \{0, \dots, N-1\}$ **do**
4:     Learn $v_{\phi^\star}$ using (34) with $\Pi = \Pi^{2n}$.
5:     Let $\mathbb{M}^{2n+1}$ be given by (33).
6:     Let $\Pi^{2n+1} = \mathbb{M}^{2n+1}_{0,1}\mathbb{Q}_{|0,1}$.
7:     Learn $v_{\theta^\star}$ using (5) with $\Pi = \Pi^{2n+1}$.
8:     Let $\mathbb{M}^{2n+2}$ be given by (31).
9:     Let $\Pi^{2n+2} = \mathbb{M}^{2n+2}_{0,1}\mathbb{Q}_{|0,1}$.
10: **end for**
11: **Output:** $v_{\theta^\star}, v_{\phi^\star}$

---

### E.4 A Reflection-projection extension

First, we consider a reflection-projection method similar to the one investigated in Bauschke and Kruk (2004). We recall that the DSBM algorithm is associated with a sequence $(\mathbb{P}^n)_{n \in \mathbb{N}}$ such that for any $n \in \mathbb{N}$

$$\mathbb{P}^{n+1/2} = \text{proj}_\mathcal{M}(\mathbb{P}^n), \qquad \mathbb{P}^{n+1} = \text{proj}_{\mathcal{R}(\mathbb{Q})}(\mathbb{P}^{n+1/2}).$$

In a reflection-projection scheme, one of the projection is replaced by a reflection. As noted in Bauschke and Kruk (2004), this can yield faster convergence rates in practice. We consider the sequence $(\mathbb{P}^n)_{n \in \mathbb{N}}$ such that for any $n \in \mathbb{N}$

$$\mathbb{P}^{n+1/2} = \text{proj}_\mathcal{M}(\mathbb{P}^n), \qquad \mathbb{P}^{n+1} = \text{proj}_{\mathcal{R}(\mathbb{Q})}(2\mathbb{P}^{n+1/2} - \mathbb{P}^n). \tag{35}$$

In what follows, we make the assumption that $2\mathbb{P}^{n+1/2} - \mathbb{P}^n$ is a probability measure, even though it is not clear if this path measure is non-negative. However, even by making this strong assumption, we show that we can recover DSBM in Algorithm 4. By considering a relaxation of the reflection-projection scheme.

First, note that for any $n \in \mathbb{N}$, $\mathbb{P}^n_{|0}$ is Markov, see De Bortoli et al. (2023) for instance. Hence, we assume that $\mathbb{P}^n$ is associated with

$$\mathrm{d}\mathbf{X}^n_t = v^n_{t,0}(\mathbf{X}_t, \mathbf{X}_0)\mathrm{d}t + \sigma_0\mathrm{d}\mathbf{B}_t, \qquad \mathbf{X}_0 \sim \pi_0. \tag{36}$$

**Estimating $\mathbb{P}^{n+1}$.** First, we compute $v^{n+1}_{t,0}$ assuming that we can sample from $\mathbb{P}^n$ and $\mathbb{P}^{n+1/2}$. Since $\mathbb{P}^{n+1}$ is in the reciprocal class, we have that $\mathbb{P}^{n+1}$ is associated with

$$\mathrm{d}\mathbf{X}_t = (\mathbb{E}_{\mathbb{P}^{n+1}_{1|0,t}}[\mathbf{X}_1 \mid \mathbf{X}_t, \mathbf{X}_0] - \mathbf{X}_t)/(1 - t)\mathrm{d}t + \sigma_0\mathrm{d}\mathbf{B}_t.$$

We refer to De Bortoli et al. (2023) for a proof of this fact. Hence, using (35), we have that

$$
\begin{aligned}
v^{n+1}_{t,0} &= \operatorname{argmin}_v \int_0^1 \int_{\mathbb{R}^d \times \mathbb{R}^d \times \mathbb{R}^d} \|v_{t,0}(t, x_t, x_0) - (x_1 - x_t)/(1 - t)\|^2 \mathrm{d}\mathbb{P}^{n+1}_{0,t,1}(x_0, x_t, x_1) \\
&= \operatorname{argmin}_v 2 \int_0^1 \int_{\mathbb{R}^d \times \mathbb{R}^d \times \mathbb{R}^d} \|v_{t,0}(t, x_t, x_0) - (x_1 - x_t)/(1 - t)\|^2 \mathrm{d}\mathbb{P}^{n+1/2}_{0,1} \mathrm{d}\mathbb{Q}_{t|0,1}(x_t|x_0, x_1) \\
&\qquad - \int_0^1 \int_{\mathbb{R}^d \times \mathbb{R}^d \times \mathbb{R}^d} \|v_{t,0}(t, x_t, x_0) - (x_1 - x_t)/(1 - t)\|^2 \mathrm{d}\mathbb{P}^n_{0,1} \mathrm{d}\mathbb{Q}_{t|0,1}(x_t|x_0, x_1).
\end{aligned}
\tag{37}
$$

Next, we turn to the estimation of $\mathbb{P}^{n+3/2}$.

**Estimating $\mathbb{P}^{n+3/2}$.** Next, we assume that for any $n \in \mathbb{N}$, $\mathbb{P}^{n+1/2}$ is associated with

$$\mathrm{d}\mathbf{X}^n_t = v^n_t(\mathbf{X}_t)\mathrm{d}t + \sigma_0\mathrm{d}\mathbf{B}_t, \qquad \mathbf{X}_0 \sim \pi_0.$$

Using (36), we have that $v^{n+1}_t$ is given by

$$v^{n+1}_t = \operatorname{argmin}_v \int_0^1 \int_{\mathbb{R}^d \times \mathbb{R}^d} \|v_t(t, x_t) - v^{n+1}_{t,0}(t, x_t, x_0)\|^2 \mathrm{d}\mathbb{P}^{n+1}_{0,t}(x_0, x_t).$$

We note also that using (Shi et al., 2023, Proposition 2) and the fact $\mathbb{P}^n$ is in the reciprocal class of $\mathbb{Q}$, we also have that

$$
\begin{aligned}
v^{n+1}_t &= \operatorname{argmin}_v 2 \int_0^1 \int_{\mathbb{R}^d \times \mathbb{R}^d \times \mathbb{R}^d} \|v_t(t, x_t) - (x_1 - x_t)/(1 - t)\|^2 \mathrm{d}\mathbb{P}^{n+1/2}_{0,1} \mathrm{d}\mathbb{Q}_{t|0,1}(x_t|x_0, x_1) \\
&\qquad - \int_0^1 \int_{\mathbb{R}^d \times \mathbb{R}^d \times \mathbb{R}^d} \|v_t(t, x_t) - (x_1 - x_t)/(1 - t)\|^2 \mathrm{d}\mathbb{P}^n_{0,1} \mathrm{d}\mathbb{Q}_{t|0,1}(x_t|x_0, x_1).
\end{aligned}
\tag{38}
$$

Hence, assuming that we can sample from $\mathbb{P}^n$ and $\mathbb{P}^{n+1/2}$ then we can estimate $v^{n+1}_{t,0}$ and $v^{n+1}_t$, i.e. sample from $\mathbb{P}^{n+1}$ and $\mathbb{P}^{n+3/2}$. Note that the losses (37) and (38) only differ by the conditioning with respect to the initial condition $x_0$ and therefore the optimisation can be conducted in parallel. We are now able to propose the following projection-reflection algorithm, see Algorithm 3.

---

**Algorithm 3** Reflection Diffusion Schrödinger Bridge Matching

---

1: **Input:** Vector field and conditional vector field $v_t^0$ and $v_{t,0}^0$, noise level $\sigma_0$ and associated bridge $\mathbb{Q}_{|0,1}$, number of outer iterations $N \in \mathbb{N}$, batch size $B$
2: **for** $n \in \{0, \dots, N-1\}$ **do**
3:     **while** not converged **do**
4:         Sample $\mathbf{X}_0^{1:B} \sim \pi_0^{\otimes B}$
5:         Sample $\mathbf{X}_1^{1:B}$ using $\mathrm{d}\mathbf{X}_t^{1:B} = v_t^n(\mathbf{X}_t^{1:B})\mathrm{d}t + \sigma_0\mathrm{d}\mathbf{B}_t$
6:         Sample $\hat{\mathbf{X}}_1^{1:B}$ using $\mathrm{d}\hat{\mathbf{X}}_t^{1:B} = v_{t,0}^n(\hat{\mathbf{X}}_t^{1:B}, \mathbf{X}_0^{1:B})\mathrm{d}t + \sigma_0\mathrm{d}\mathbf{B}_t$
7:         $\mathcal{L} = \int_0^1 [\sum_{k=1}^B \|v_t(\mathbf{X}_t^k) - \frac{\mathbf{X}_1^k - \mathbf{X}_t^k}{1-t}\|^2 - (1/2)\sum_{k=1}^B \|v_t(\mathbf{X}_t^k) - \frac{\mathbf{X}_1^k - \mathbf{X}_t^k}{1-t}\|^2]\mathrm{d}t$
8:         $\mathcal{L}_0 = \int_0^1 [\sum_{k=1}^B \|v_{t,0}(\mathbf{X}_t^k, \mathbf{X}_0^k) - \frac{\mathbf{X}_1^k - \mathbf{X}_t^k}{1-t}\|^2 - (1/2)\sum_{k=1}^B \|v_{t,0}(\mathbf{X}_t^k, \mathbf{X}_0^k) - \frac{\mathbf{X}_1^k - \mathbf{X}_t^k}{1-t}\|^2]\mathrm{d}t$
9:         $v_t^{n+1} = \mathrm{Gradientstep}(\mathcal{L})$
10:        $v_{t,0}^{n+1} = \mathrm{Gradientstep}(\mathcal{L}_0)$
11:     **end while**
12: **end for**
13: **Output:** $v_t^{N+1}, v_{t,0}^{N+1}$

---

Note that in Algorithm 3 we only consider the optimisation of a forward process but similarly to Algorithm 2, one can construct a forward backward extension to alleviate some of the bias accumulation problems. Finally, we can interpolate between DSBM and this new reflection algorithm and DSBM by introducing an hyperparameter $\alpha \geq 0$ and consider the following extension given in Algorithm 4

---

**Algorithm 4** Reflection Diffusion Schrödinger Bridge Matching

---

1: **Input:** Vector field and conditional vector field $v_t^0$ and $v_{t,0}^0$, noise level $\sigma_0$ and associated bridge $\mathbb{Q}_{|0,1}$, number of outer iterations $N \in \mathbb{N}$, batch size $B$
2: **for** $n \in \{0, \dots, N-1\}$ **do**
3:     **while** not converged **do**
4:         Sample $\mathbf{X}_0^{1:B} \sim \pi_0^{\otimes B}$
5:         Sample $\mathbf{X}_1^{1:B}$ using $\mathrm{d}\mathbf{X}_t^{1:B} = v_t^n(\mathbf{X}_t^{1:B})\mathrm{d}t + \sigma_0\mathrm{d}\mathbf{B}_t$
6:         Sample $\hat{\mathbf{X}}_1^{1:B}$ using $\mathrm{d}\hat{\mathbf{X}}_t^{1:B} = v_{t,0}^n(\hat{\mathbf{X}}_t^{1:B}, \mathbf{X}_0^{1:B})\mathrm{d}t + \sigma_0\mathrm{d}\mathbf{B}_t$
7:         $\mathcal{L} = \int_0^1 [\sum_{k=1}^B \|v_t(\mathbf{X}_t^k) - \frac{\mathbf{X}_1^k - \mathbf{X}_t^k}{1-t}\|^2 - \alpha\sum_{k=1}^B \|v_t(\mathbf{X}_t^k) - \frac{\mathbf{X}_1^k - \mathbf{X}_t^k}{1-t}\|^2]\mathrm{d}t$
8:         $\mathcal{L}_0 = \int_0^1 [\sum_{k=1}^B \|v_{t,0}(\mathbf{X}_t^k, \mathbf{X}_0^k) - \frac{\mathbf{X}_1^k - \mathbf{X}_t^k}{1-t}\|^2 - \alpha\sum_{k=1}^B \|v_{t,0}(\mathbf{X}_t^k, \mathbf{X}_0^k) - \frac{\mathbf{X}_1^k - \mathbf{X}_t^k}{1-t}\|^2]\mathrm{d}t$
9:         $v_t^{n+1} = \mathrm{Gradientstep}(\mathcal{L})$
10:        $v_{t,0}^{n+1} = \mathrm{Gradientstep}(\mathcal{L}_0)$
11:     **end while**
12: **end for**
13: **Output:** $v_t^{N+1}, v_{t,0}^{N+1}$

---

Using different values of $\alpha \geq 0$ in Algorithm 4, we recover different existing algorithms. If $\alpha = 1$, we recover DSBM Shi et al. (2023). Finally, if $\alpha = 1/2$, we recover the reflection algorithm Algorithm 3.

## F   Consistency in Schrödinger Bridge

The idea of training both the forward and the backward jointly was mentioned in (Shi et al., 2023, Section G). However, it was still assumed that, while being trained jointly, the forward and backward vector fields were obtained using an argmin operation, see (Shi et al., 2023, Equation (43), (44)). In addition, in (Shi et al., 2023, Section G) a consistency loss was proposed in order to enforce that the forward and backward processes match, see (Shi et al., 2023, Equation (49)). In this section, we leverage new results from Daras et al. (2023a); De Bortoli et al. (2024) in order to enforce the internal consistency of the model.

First, note that for any $(\mathbf{X}_t)_{t\in[0,1]}$ associated with $\mathbb{P} \in \mathcal{R}(\mathbb{Q})$ we have for any $0 \leq t_0 \leq t \leq t_1 \leq 1$ that

$$\mathbf{X}_t = \frac{t - t_0}{t_1 - t_0}\mathbf{X}_{t_1} + \frac{t_1 - t}{t_1 - t_0}\mathbf{X}_{t_0} + \sigma_{t_0,t,t_1}\mathbf{Z}, \qquad \mathbf{Z} \sim \mathcal{N}(0, \mathrm{Id}),$$

where

$$\sigma_{t_0,t,t_1} = \sqrt{\frac{(t - t_0)(t_1 - t)}{t_1 - t_0}}.$$

Let $p_t$ be the density of $\mathbf{X}_t$ with respect to the Lebesgue measure, we have that for any $0 \leq t_0 \leq t \leq t_1 \leq 1$ and $x_t \in \mathbb{R}^d$

$$p_t(x_t) = \int_{\mathbb{R}^d \times \mathbb{R}^d} (2\pi\sigma_{t_0,t,t_1}^2)^{-d/2} \exp\left(-\frac{\|x_t - \frac{t-t_0}{t_1-t_0}x_{t_1} - \frac{t_1-t}{t_1-t_0}x_{t_0}\|^2}{2\sigma_{t_0,t,t_1}^2}\right) p_{t_0}(x_{t_0})p_{t_1}(x_{t_1})\mathrm{d}x_{t_0}\mathrm{d}x_{t_1}.$$

Using the change of variable $x_{t_0} \to x_{t_0} + x_t$ and $x_{t_1} \to x_{t_1} + x_t$ we get that for any $0 \leq t_0 \leq t \leq t_1 \leq 1$ and $x_t \in \mathbb{R}^d$

$$\nabla \log p_t(x_t) = \int_{\mathbb{R}^d \times \mathbb{R}^d} \{\nabla \log p_{t_0}(x_{t_0}) + \nabla \log p_{t_1}(x_{t_1})\} p_{t_0,t_1|t}(x_{t_0}, x_{t_1}|x_t)\mathrm{d}x_t. \qquad (39)$$

This identity for the score has already been presented in a bridge matching context in (De Bortoli et al., 2024, Section 3.3). Let $\mathbb{P} \in \mathcal{R}(\mathbb{Q})$ then we have that $\mathrm{proj}_{\mathcal{M}}(\mathbb{P})$ is such that for any $t \in [0, 1]$, $\mathrm{proj}_{\mathcal{M}}(\mathbb{P})_t = \mathbb{P}_t$, see (Shi et al., 2023, Proposition 2). We have that for any $t \in [0, 1]$, $\vec{v}_t(x) + v_{1-t}^{\mathrm{e}}(x) = \sigma_0^2 \nabla \log p_t(x)$. Combining this result and (39), this suggests considering the following consistency loss

$$\ell_{\mathrm{cons},(t_0,t,t_1)}(\theta) = \mathbb{E}[\|v_\theta(t, 1, \mathbf{X}_t) + v_\theta(1 - t, 0, \mathbf{X}_t) \qquad\qquad\qquad (40)$$
$$- v_\theta(t_0, 1, \mathbf{X}_t) - v_\theta(1 - t_0, 0, \mathbf{X}_{t_0}) - v_\theta(t_1, 1, \mathbf{X}_{t_1}) - v_\theta(1 - t_1, 0, \mathbf{X}_{t_1})\|^2].$$

Similarly to (13), we can consider an empirical version of (40).

# G    Model stitching

In Algorithm 1, the finetuning stage requires a pretrained bridge matching model interpolating between $\pi_0$ and $\pi_1$ (lines 2-7). However, for large datasets with complex distributions $\pi_0$ and $\pi_1$, e.g. ImageNet, training this bridge model from scratch can be computationally expensive. To improve efficiency, we can leverage existing diffusion models targeting $\pi_0$ and $\pi_1$. Specifically, we assume access to generative models transferring between $\mathcal{N}(0, \mathrm{Id})$ and $\pi_0$, and between $\mathcal{N}(0, \mathrm{Id})$ and $\pi_1$. In the rest of this section, we show how one can adapt Algorithm 1 to this setting. We then comment on the link between the proposed algorithm and Dual Diffusion Implicit Bridges (Su et al., 2023).

**Setting.**    For simplicity, assume that we have two pretrained diffusion models for $\pi_0$ and $\pi_1$. We describe our procedure for $\pi_0$. Consider a forward process of the form $\mathbf{X}_t = \mathbf{X}_0 + \sigma_t \mathbf{Z}$, with $\mathbf{Z} \sim \mathcal{N}(0, \mathrm{Id})$, where $\sigma_t$ is a hyperparameter. Note that we could have considered an interpolant of the form $\mathbf{X}_t = \alpha_t \mathbf{X}_0 + \sigma_t \mathbf{Z}$ instead, see Song et al. (2021a) for instance.

We assume that the model $\mathbf{X}_t = \mathbf{X}_0 + \sigma_t \mathbf{Z}$ is associated with the forward diffusion model

$$\mathrm{d}\mathbf{X}_t = g_t \mathrm{d}\mathbf{B}_t, \qquad\qquad\qquad (41)$$

where we assume that $g_t \geq 0$ for all $t \in [0, 1]$. Note that we have that for any $t \in [0, 1]$, $\sigma_t^2 = \int_0^t g_s^2 \mathrm{d}s$. In particular, we have that for any $s, t \in [0, 1]$ with $s \leq t$

$$\mathbf{X}_t = \mathbf{X}_s + \sqrt{\sigma_t^2 - \sigma_s^2}\mathbf{Z}, \qquad \mathbf{Z} \sim \mathcal{N}(0, \mathrm{Id}).$$

Our goal is to solve the following Entropic Optimal Transport problem

$$\Pi^\star = \mathrm{argmin}_{\Pi \in \mathcal{P}(\mathbb{R}^{2d})}\left\{\int_{\mathbb{R}^d \times \mathbb{R}^d} c(x, y)\mathrm{d}\Pi(x, y) - \varepsilon\mathrm{H}(\Pi) \; ; \; \Pi_0 = \pi_0, \; \Pi_1 = \pi_1\right\}, \qquad (42)$$

where $\varepsilon > 0$ is some entropic regularisation. We assume that $\varepsilon > 0$ is fixed and assume that there exists $t' \in [0, 1]$ such that $\sigma_{t'}^2 = \varepsilon/2$. We now consider a dynamic version of (42) with

$$\mathbb{P}^\star = \mathrm{argmin}_{\mathbb{P}\in\mathcal{P}(\mathcal{C})}\{\mathrm{KL}(\mathbb{P}|\mathbb{Q}) \; ; \; \mathbb{P}_0 = \pi_0, \; \mathbb{P}_{t'} = \pi_1\}, \qquad (43)$$

where $\mathbb{Q}$ is associated with $(\mathbf{X}_t)_{t \in [0,t']}$ (41). Note that contrary to the setting presented in the main paper, here we do not consider the integration between time 0 and 1 but between time 0 and $t'$. It can be shown that for any $t \in [0,t']$, $(\mathbf{X}_t)_{t \in [0,t']}$ associated with $\mathbb{Q}_{t|0,t'}$ is given by

$$\mathbf{X}_t = \text{Interp}_t(\mathbf{X}_0, \mathbf{X}_{t'}, \mathbf{Z}) = \left(1 - \frac{\sigma_t^2}{\sigma_{t'}^2}\right)\mathbf{X}_0 + \frac{\sigma_t^2}{\sigma_{t'}^2}\mathbf{X}_{t'} + \sigma_t\sqrt{1 - \frac{\sigma_t^2}{\sigma_{t'}^2}}\mathbf{Z}, \quad \mathbf{Z} \sim \mathcal{N}(0, \text{Id}).$$

Solving (43) is equivalent to solving (42). We now propose an algorithm to solve (43). It corresponds to the finetuning stage of Algorithm 1 with a specific initialisation, similar to DSBM-IPF in Shi et al. (2023).

By $v_\phi$, we denote a DDM model associated with $\pi_1$:

$$\mathrm{d}\mathbf{X}_t = v_\phi(t, \mathbf{X}_t)\mathrm{d}t + g_t\mathrm{d}\mathbf{B}_t, \quad \mathbf{X}_0 \sim \mathcal{N}(0, \text{Id}), \quad \mathbf{X}_1 \sim \pi_1. \tag{44}$$

Similarly, $v_\theta$ denotes a diffusion model associated with $\pi_0$:

$$\mathrm{d}\mathbf{Y}_t = v_\theta(t, \mathbf{Y}_t)\mathrm{d}t + g_t\mathrm{d}\mathbf{B}_t, \quad \mathbf{Y}_0 \sim \mathcal{N}(0, \text{Id}), \quad \mathbf{Y}_1 \sim \pi_0 \tag{45}$$

In analogy to Equation (11), the two equations above correspond to the forward and backward SDEs.

For a given batch of inputs $\mathbf{X}_0^{1:B}$ and $\mathbf{X}_1^{1:B}$, timesteps $t \sim \text{Unif}([0,t'])^{\otimes B}$, and interpolations $\mathbf{X}_t^{\leftarrow}$ and $\mathbf{X}_t^{\rightarrow}$, we compute the empirical forward and backward losses as the following modification of Equation (13):

$$\ell^{\rightarrow}(\phi; t, \mathbf{X}_1, \mathbf{X}_t^{\rightarrow}) = \frac{1}{B}\sum_{i=1}^{B}\left\|v_\phi\left(t^i, \mathbf{X}_t^{\rightarrow i}\right) - \left(\mathbf{X}_1^i - \mathbf{X}_t^{\rightarrow i}\right)/\sigma_t^i\right\|^2,$$

$$\ell^{\leftarrow}(\theta; t, \mathbf{X}_0, \mathbf{X}_t^{\leftarrow}) = \frac{1}{B}\sum_{i=1}^{B}\|v_\theta\left(t^i, \mathbf{X}_t^{\leftarrow i}\right) - \left(\mathbf{X}_0^i - \mathbf{X}_t^{\leftarrow i}\right)/\sqrt{\sigma_{t'}^2 - \sigma_{t^i}^2}\|^2.$$

Algorithm 5 corresponds to an online version of DSBM-IPF (Shi et al., 2023) with the initialisation given by two generative models. In Algorithm 5, we finetune the trained vector fields to solve the interpolation task. At inference time, the SDE associated with vector field $v_\theta$ interpolates between $\pi_1 \to \pi_0$, while the SDE associated with the vector field $v_\phi$ interpolates between $\pi_0 \to \pi_1$.

---

**Algorithm 5** $\alpha$-Diffusion Schrödinger Bridge Matching for DDM finetuning

1: **Input:** datasets $\pi_0$ and $\pi_1$, number finetuning steps $N_{\text{finetuning}}$, batch size $B$, DDM parameters $\phi$ and $\theta$.
2: **for** $n \in \{1, \ldots, N_{\text{finetuning}}\}$ **do**
3:     Sample $(\mathbf{X}_0, \mathbf{X}_1) \sim (\pi_0 \otimes \pi_1)^{\otimes B}$, $t \sim \text{Unif}([0,1])$, $\mathbf{Z}^{1:B} \sim \mathcal{N}(0, \text{Id})^{\otimes B}$
4:     Sample $\hat{\mathbf{X}}_{t'}^{\leftarrow}$ by solving (44) starting from $\mathbf{X}_0$
5:     Sample $\hat{\mathbf{X}}_{t'}^{\rightarrow}$ by solving (45) starting from $\mathbf{X}_1$
6:     Sample $t^{\leftarrow} \sim \text{Unif}([0,t'])^{\otimes B}$, $\mathbf{Z}^{\leftarrow} \sim \mathcal{N}(0, \text{Id})^{\otimes B}$, and compute $\mathbf{X}_t^{\leftarrow} = \text{Interp}_{t^{\leftarrow}}(\mathbf{X}_0, \hat{\mathbf{X}}_{t'}^{\leftarrow}, \mathbf{Z}^{\leftarrow})$
7:     Sample $t^{\rightarrow} \sim \text{Unif}([0,t'])^{\otimes B}$, $\mathbf{Z}^{\rightarrow} \sim \mathcal{N}(0, \text{Id})^{\otimes B}$, and compute $\mathbf{X}_t^{\rightarrow} = \text{Interp}_{t^{\rightarrow}}(\mathbf{X}_1, \hat{\mathbf{X}}_{t'}^{\rightarrow}, \mathbf{Z}^{\rightarrow})$
8:     Update $\theta$ with gradient step on $\ell^{\leftarrow}(\theta; t^{\leftarrow}, \mathbf{X}_0, \mathbf{X}_t^{\leftarrow})$
9:     Update $\phi$ with gradient step on $\ell^{\rightarrow}(\phi; t^{\rightarrow}, \mathbf{X}_1, \mathbf{X}_t^{\rightarrow})$
10: **end for**
11: **Output:** $\theta, \phi$ parameters of the finetuned models

---

Our model stitching approach is related to Dual Diffusion Implicit Bridges (DDIB) (Su et al., 2023), which uses pretrained diffusion models, but without further finetuning. As highlighted in Shi et al. (2023), DDIB is inferior to DSBM in terms of quality and alignment of the samples.

# H Extended related work

We highlight links between our proposed flow and Sinkhorn flows in Appendix H.1. We draw connection between our practical approach and Reinforcement Learning in Appendix H.2. We discuss how $\alpha$-IMF is related to (incremental) Expectation-Maximisation in Appendix H.3. Finally, we discuss how our algorithm can be seen as an instance of continual learning in Appendix H.5.

## H.1 Links with Sinkhorn flow

In this section, we discuss the links between our approach and the Sinkhorn flow introduced by Karimi et al. (2024). We start by recalling how Sinkhorn flows are defined and then discuss how they are related to our approach.

$\gamma$**-Sinkhorn and Sinkhorn flows.** We first consider the static EOT problem and recall the Sinkhorn procedure, also called Iterative Proportional Fitting. We define a sequence of coupling $(\bar{\Pi}^n, \Pi^n)_{n \in \mathbb{N}}$, i.e. for any $n \in \mathbb{N}$, $\Pi^n \in \mathcal{P}(\mathbb{R}^d \times \mathbb{R}^d)$. We let $\Pi^0 = \mathbb{Q}_{0,1}$ and we consider for any $n \in \mathbb{N}$,

$$\Pi^n = \mathrm{argmin}\{\mathrm{KL}(\Pi \mid \bar{\Pi}^n) \ : \ \Pi \in \mathcal{P}(\mathbb{R}^d \times \mathbb{R}^d), \ \Pi_0 = \pi_0\}, \tag{46}$$
$$\bar{\Pi}^{n+1} = \mathrm{argmin}\{\mathrm{KL}(\Pi \mid \Pi^n) \ : \ \Pi \in \mathcal{P}(\mathbb{R}^d \times \mathbb{R}^d), \ \Pi_1 = \pi_1\},$$

In Karimi et al. (2024), the authors generalise (46) by introducing an extra hyperparameter $\gamma \in (0, 1]$ and defining

$$\Pi^n = \mathrm{argmin}\{\mathrm{KL}(\Pi \mid \bar{\Pi}^n) \ : \ \Pi \in \mathcal{P}(\mathbb{R}^d \times \mathbb{R}^d), \ \Pi_1 = \pi_1\}, \tag{47}$$
$$\bar{\Pi}^{n+1} = \mathrm{argmin}\{\gamma \mathrm{KL}(\Pi \mid \Pi^n) + (1 - \gamma)\mathrm{KL}(\Pi \mid \bar{\Pi}^n) \ : \ \Pi \in \mathcal{P}(\mathbb{R}^d \times \mathbb{R}^d), \ \Pi_0 = \pi_0\},$$

Using (Karimi et al., 2024, Lemma 2), we have that for any $\gamma \in (0, 1]$, any $n \in \mathbb{N}$ and any $x_0, x_1 \in \mathbb{R}^d$

$$(\mathrm{d}\bar{\Pi}^n/\mathrm{d}\mathbb{Q}_{0,1})(x_0, x_1) = \exp[f_\gamma^n(x_0) + g_\gamma^n(x_1)], \tag{48}$$

with $f_\gamma^0 = g_\gamma^0 = 0$ and for any $n \in \mathbb{N}$, $\gamma \in (0, 1]$ and $x_1 \in \mathbb{R}^d$

$$g_\gamma^{n+1}(x_1) = g_\gamma^n(x_1) - \gamma \log(\mathrm{d}\bar{\Pi}_1^n/\mathrm{d}\pi_1)(x_1). \tag{49}$$

In addition, using (Karimi et al., 2024, Equation (9)) we have that for any $n \in \mathbb{N}$, $\gamma \in (0, 1]$ and $x_0 \in \mathbb{R}^d$

$$f_\gamma^n(x_0) = -\log\left(\int_{\mathbb{R}^d} \exp[g_\gamma^n(x_1) - (1/(2\varepsilon))\|x_0 - x_1\|^2]\mathrm{d}\pi_1(x_1)\right).$$

When letting $\gamma \to 0$, (48) and (49) suggest to consider for any $s \geq 0$, $x_0, x_1 \in \mathbb{R}^d$

$$(\mathrm{d}\bar{\Pi}^s/\mathrm{d}\mathbb{Q}_{0,1})(x_0, x_1) = \exp[f^s(x_0) + g^s(x_1)], \qquad \Pi^s = \mathrm{argmin}\{\mathrm{KL}(\Pi \mid \bar{\Pi}^s) \ : \ \Pi, \ \Pi_1 = \pi_1\},$$

where for any $s \geq 0$, $x_1 \in \mathbb{R}^d$

$$\partial_s g^s(x_1) = -\log(\mathrm{d}\bar{\Pi}_1^s/\mathrm{d}\pi_1)(x_1), \qquad \partial_s f^s(x_0) = \int_{\mathbb{R}^d} \log(\mathrm{d}\bar{\Pi}_1^s/\mathrm{d}\pi_1)(x_1)\mathrm{d}\bar{\Pi}^s(x_1|x_0).$$

**Comparison with Schrödinger Bridge flows.** In order to compare our approach with the one of Karimi et al. (2024), we start by rewriting the $\gamma$-Sinkhorn algorithm defined by (47). To do so, we introduce the projection on the measures with fixed marginal.

**Definition H.1 (Projection on marginals):** *Let $\Pi \in \mathcal{P}(\mathbb{R}^d \times \mathbb{R}^d)$ and $\pi_0 \in \mathcal{P}(\mathbb{R}^d)$, we define* $\mathrm{proj}_{0,\pi_0}(\Pi)$ *as follows*

$$\mathrm{proj}_{0,\pi_0}(\Pi) = \mathrm{argmin}\{\mathrm{KL}(\tilde{\Pi} \mid \Pi) \ : \ \tilde{\Pi} \in \mathcal{P}(\mathbb{R}^d \times \mathbb{R}^d), \tilde{\Pi}_0 = \pi_0\}.$$

*Similarly, for any $\Pi \in \mathcal{P}(\mathbb{R}^d \times \mathbb{R}^d)$ and $\pi_1 \in \mathcal{P}(\mathbb{R}^d)$, we define* $\mathrm{proj}_{1,\pi_1}(\Pi)$ *as follows*

$$\mathrm{proj}_{1,\pi_1}(\Pi) = \mathrm{argmin}\{\mathrm{KL}(\tilde{\Pi} \mid \Pi) \ : \ \tilde{\Pi} \in \mathcal{P}(\mathbb{R}^d \times \mathbb{R}^d), \tilde{\Pi}_1 = \pi_1\}.$$

| | $\gamma$-Sinkhorn | $\gamma$-IMF |
|---|---|---|
| Loss function | $\mathrm{KL}(\Pi \,|\mathrm{proj}_{1,\pi_1}(\tilde{\Pi}))$ | $\int_0^1 \mathbb{E}_{\mathrm{proj}_{\mathcal{R}(\mathbb{Q})}(\mathbb{P})}[\|v_t(\mathbf{X}_t) - \frac{\mathbf{X}_1 - \mathbf{X}_t}{1-t}\|^2]\mathrm{d}t$ |
| regularisation | $\mathrm{KL}(\Pi \,|\tilde{\Pi})$ | $\int_0^1 \int_{\mathbb{R}^d} \|f_t(x_t) - \tilde{f}_t(x_t)\|^2 \mathrm{d}\mu_t(x_t)\mathrm{d}t$ |
| Update | Implicit | Explicit |

Table 2: Comparison between $\gamma$-Sinkhorn and $\gamma$-IMF.

With these definitions, we have that for any $n \in \mathbb{N}$, $\bar{\Pi}^{n+1} = \mathrm{proj}_{0,\pi_0}(\mathrm{proj}_{1,\pi_1}(\bar{\Pi}^n))$, with $(\bar{\Pi}^n)_{n\in\mathbb{N}}$ the original Sinkhorn sequence defined by (46). Similarly, we have that the original Iterative Markovian Fitting (IMF) sequence $(\hat{\mathbb{P}}^n)_{n\in\mathbb{N}}$ as defined in (4) with $\alpha = 1$ satisfies for any $n \in \mathbb{N}$, $\hat{\mathbb{P}}^{n+1} = \mathrm{proj}_{\mathcal{R}(\mathbb{Q})}(\mathrm{proj}_{\mathcal{M}}(\mathbb{P}^n))$. The analogy between the Sinkhorn iterates and the IMF sequence was already highlighted in Shi et al. (2023); Peluchetti (2023) and further studied in Brekelmans and Neklyudov (2023). We know show that similarly, we can draw an analogy between the sequences defined in (4) with $\alpha \in (0,1)$ and the sequences obtained in $\gamma$-Sinkhorn. To do so, we start by introducing for any $\Pi, \tilde{\Pi} \in \mathcal{P}(\mathbb{R}^d \times \mathbb{R}^d)$

$$\mathcal{L}^{\mathrm{IPF}}(\Pi, \tilde{\Pi}) = \mathrm{KL}(\Pi \,|\mathrm{proj}_{1,\pi_1}(\tilde{\Pi})), \qquad R^{\mathrm{IPF}}(\Pi, \tilde{\Pi}) = \mathrm{KL}(\Pi \,|\tilde{\Pi}).$$

With this notation, we can now rewrite (47) for any $n \in \mathbb{N}$ as

$$\bar{\Pi}^{n+1} = \mathrm{argmin}\{\mathcal{L}^{\mathrm{IPF}}(\Pi, \bar{\Pi}^n) + ((1-\gamma)/\gamma)R^{\mathrm{IPF}}(\Pi, \bar{\Pi}^n) \,:\, \Pi \in \mathcal{P}(\mathbb{R}^d \times \mathbb{R}^d), \, \Pi_0 = \pi_0\}.$$

Now, we are going to see that (50) is linked with the discretisation of the path measure flow described in (4). Recall that for any suitable $v$, we define the path measure $\mathbb{P}_v$ associated with

$$\mathrm{d}\mathbf{X}_t = v_t(\mathbf{X}_t)\mathrm{d}t + \sqrt{\varepsilon}\mathrm{d}\mathbf{B}_t, \qquad \mathbf{X}_0 \sim \pi_0.$$

We define

$$\mathcal{L}(v, \mathbb{P}) = \int_0^1 \mathcal{L}(v_t, \mathbb{P})\mathrm{d}t = \int_0^1 \int_{\mathbb{R}^d \times \mathbb{R}^d} \left\|v_t(x_t) - \frac{x_1 - x_t}{1-t}\right\|^2 \mathrm{dproj}_{\mathcal{R}(\mathbb{Q})}(\mathbb{P})_{t,1}(x_t, x_1)\mathrm{d}s.$$

Similarly, for any $\mu \in \mathcal{P}(\mathcal{C})$, we define

$$R_\mu(\mathbb{P}_v, \mathbb{P}_{\tilde{v}}) = \int_0^1 \int_{\mathbb{R}^d} \|v_t(x_t) - \tilde{v}_t(x_t)\|^2 \mathrm{d}\mu_t(x_t)\mathrm{d}t.$$

Next, we define the sequence of path measures $(\bar{\mathbb{P}}^n)_{n\in\mathbb{N}}$ such that for any $n \in \mathbb{N}$

$$\bar{\mathbb{P}}^{n+1} = \mathrm{argmin}\{\mathcal{L}(\mathbb{P}, \bar{\mathbb{P}}^n) + (1/\alpha)R_{\mu^n}(\Pi, \bar{\mathbb{P}}^n) \,:\, \mathbb{P} = \mathbb{P}_v, \text{ for some } v\}. \tag{50}$$

Now, if we denote $(v^n)_{n\in\mathbb{N}}$ the sequence such that for any $n \in \mathbb{N}$, $\bar{\mathbb{P}}^n = \mathbb{P}_{v^n}$ then we have that for any $n \in \mathbb{N}$, $t \in [0,1]$ and $x \in \mathbb{R}^d$

$$v_t^{n+1}(x) = v_t^n(x) - \delta\nabla_{\mu^n}\mathcal{L}_t(v^{n+1}, \bar{\mathbb{P}}^n)(x). \tag{51}$$

Recall that $(\mathbb{P}^n)_{n\in\mathbb{N}}$ given by (4) is associated with $(v^n)_{n\in\mathbb{N}}$ such that for any $n \in \mathbb{N}$, $t \in [0,1]$ and $x \in \mathbb{R}^d$

$$v_t^{n+1}(x) = v_t^n(x) - \delta\nabla_\mu^n\mathcal{L}_t(v^n, \mathbb{P}^n)_t(x), \tag{52}$$

see Proposition 3.2. Therefore, the only difference between (52) and (51) is that (52) is an explicit update whereas (51) is an implicit update. We summarise the differences between $\gamma$-Sinkhorn and the discretisation we introduce in Table 2.

## H.2 Links with Reinforcement Learning

In this section, we draw some connection between Algorithm 1 and self-play in Reinforcement Learning. In particular, we introduce a generalisation of Algorithm 1 which uses the concept of replay buffer commonly used in Reinforcement learning, see Mnih et al. (2015) for instance.

We first present a generalisation of Algorithm 1 called Replay Buffer Diffusion Schrödinger Bridge Matching Algorithm 6. We define a buffer $\mathcal{B}$ as a collection of samples $\{(\mathbf{X}_0^k, \mathbf{X}_1^k)\}_{k=1}^N$, where

$N \in \mathbb{N}$ is the size of the buffer equipped with two functions Add and Sample. We have that Add : $\Omega \times \bigsqcup_{k \in \mathbb{N}} (\mathbb{R}^{2d})^k \times (\mathbb{R}^{2d})^N \to (\mathbb{R}^{2d})^N$, where $\Omega$ is a probability space. In practice Add takes a random number (the function can be stochastic), any number of proposed samples as well as the current buffer. As an output Add returns the updated buffer. We also define Sample : $\Omega \times \mathbb{N} \times (\mathbb{R}^{2d})^N \to \bigsqcup_{k \in \mathbb{N}} (\mathbb{R}^{2d})^k$. This function takes a random number (the function can be stochastic), a natural number $k$ representing the number of samples to return as well as the current buffer. As an output Sample returns a batch of $k$ samples from the buffer.

---

**Algorithm 6** Replay Buffer Diffusion Schrödinger Bridge Matching

---

1: **Input:** $\pi_0$, $\pi_1$, $\varepsilon$ (entropic regularisation), $N_{\text{pretraining}}$ (number of pretraining steps), $N_{\text{finetuning}}$ (number of finetuning steps), $B$ (batch size), $\gamma$ (EMA parameter), $\theta$ (initial parameters), $\mathcal{B}^{\text{fwd}}$ (forward buffer), $\mathcal{B}^{\text{bwd}}$ (backward buffer)
2: $\bar{\theta} = \theta$
3: **for** $n \in \{0, \ldots, N_{\text{pretraining}}\}$ **do**
4:     Sample $(\mathbf{X}_0^{1:B}, \mathbf{X}_1^{1:B}) \sim (\pi_0 \otimes \pi_1)^{\otimes B}$, $t \sim \text{Unif}([0,1])$, $\mathbf{Z}^{1:B} \sim \mathcal{N}(0, \text{Id})^{\otimes B}$
5:     Compute $\mathbf{X}_t^{1:B} = \text{Interp}_t(\mathbf{X}_0^{1:B}, \mathbf{X}_1^{1:B}, \mathbf{Z}^{1:B})$ using (12)
6:     Update $\theta$ with gradient step on $\ell_t^B$, $\bar{\theta} = (1 - \gamma)\bar{\theta} + \gamma\theta$
7: **end for**
8: **for** $n \in \{0, \ldots, N_{\text{finetuning}}\}$ **do**
9:     **if** $n \equiv 0[n_{\text{refresh}}]$ **then**
10:         Sample $(\hat{\mathbf{X}}_0^{1:B}, \hat{\mathbf{Y}}_0^{1:B}) \sim (\pi_0 \otimes \pi_1)^{\otimes B}$, $t \sim \text{Unif}([0,1])$, $\mathbf{Z}^{1:B} \sim \mathcal{N}(0, \text{Id})^{\otimes B}$
11:         Sample $(\mathbf{X}_1^{1:B}, \mathbf{Y}_1^{1:B})$ using (11) with initialisation $(\hat{\mathbf{X}}_0^{1:B}, \hat{\mathbf{Y}}_0^{1:B})$
12:         $\mathcal{B}^{\text{fwd}} = \text{Add}((\hat{\mathbf{X}}_0^{1:B}, \mathbf{X}_1^{1:B}), \mathcal{B}^{\text{fwd}})$
13:         $\mathcal{B}^{\text{bwd}} = \text{Add}((\mathbf{Y}_1^B, \hat{\mathbf{Y}}_0^{1:B}), \mathcal{B}^{\text{bwd}})$
14:     **end if**
15:     $(\hat{\mathbf{X}}_0^{1:B}, \mathbf{X}_1^{1:B}) = \text{Sample}(B, \mathcal{B}^{\text{fwd}})$
16:     $(\mathbf{Y}_1^{1:B}, \hat{\mathbf{Y}}_0^{1:B}) = \text{Sample}(B, \mathcal{B}^{\text{bwd}})$
17:     Compute $\mathbf{X}_t^{1:B} = \text{Interp}_t(\hat{\mathbf{X}}_0^{1:B}, \mathbf{X}_1^{1:B}, \mathbf{Z}^{1:B})$ using (12)
18:     Compute $\mathbf{Y}_{1-t}^{1:B} = \text{Interp}_t(\mathbf{Y}_1^{1:B}, \hat{\mathbf{Y}}_0^{1:B}, \mathbf{Z}^{1:B})$ using (12)
19:     Update $\theta$ with gradient step on $\ell_t^B$, $\bar{\theta} = (1 - \gamma)\bar{\theta} + \gamma\theta$
20: **end for**
21: **Output:** $(\theta, \bar{\theta})$ parameters of the finetuned model

---

In Algorithm 6, we allow for more flexibility than the online procedure by leveraging the concept of replay buffer originally introduced in Reinforcement Learning Mnih et al. (2015). The concept of replay buffer has been used previously in Schrödinger Bridge works, with the notion of cache where every $n_{\text{refresh}}$ steps a cache is emptied and filled with new samples. If $n_{\text{refresh}} = 1$, $N = B$ for both $\mathcal{B}^{\text{fwd}}$ and $\mathcal{B}^{\text{fwd}}$ we have that for any $\omega \in \Omega$ and $(\mathbf{X}_0^{1:B}, \mathbf{X}_1^{1:B}) \in (\mathbb{R}^{2d})^B$

$$\text{Add}(\omega, (\mathbf{X}_0^{1:B}, \mathbf{X}_1^{1:B}), \mathcal{B}) = (\mathbf{X}_0^{1:B}, \mathbf{X}_1^{1:B}),$$
$$\text{Sample}(\omega, B, (\mathbf{X}_0^{1:B}, \mathbf{X}_1^{1:B})) = (\mathbf{X}_0^{1:B}, \mathbf{X}_1^{1:B}).$$

This means that the Add simply fills the buffer with the new samples while Sample just return the whole current buffer. In that case we recover Algorithm 1. For more general update rules, the replay buffers $\mathcal{B}^{\text{fwd}}$ and $\mathcal{B}^{\text{bwd}}$ allow us to collect previous samples and therefore to keep a memory of the past experiences. In future work, we plan to investigate popular choice in experience replay and their impact on the performance of Algorithm 6.

### H.3 Links with Expectation Maximisation

In this section, we make a connection between DSBM and the Expectation Maximisation (EM) algorithm, and show that the discretisation of the Schrödinger Flow proposed in Algorithm 1 corresponds to some incremental version of an idealised algorithm, as discussed in Neal and Hinton (1998). We would like to emphasize that the link between the EM algorithm and Diffusion Schrödinger Bridge based methodologies was already highlighted by Vargas et al. (2024); Brekelmans and Neklyudov (2023). Below, we follow the framework of Brekelmans and Neklyudov (2023) and recall the following definitions.

**Definition H.2 (Projections and maximisations):** *Let* $A$ *be a subset of* $\mathcal{P}(\mathcal{C})$. *Then, for any* $\mathbb{P} \in \mathcal{P}(\mathcal{C})$, *when it is well-defined, we define its* E-*projection on* $A$ *as* $\mathbb{P}^{\star} = \mathrm{argmin}_{\mathbb{Q} \in A} \mathrm{KL}(\mathbb{Q} \mid \mathbb{P})$. *Similarly, for any* $\mathbb{P} \in \mathcal{P}(\mathcal{C})$, *when it is well-defined, we define its* M-*projection on* $A$ *as* $\mathbb{P}^{\star} = \mathrm{argmin}_{\mathbb{Q} \in A} \mathrm{KL}(\mathbb{P} \mid \mathbb{Q})$.

In Brekelmans and Neklyudov (2023), the authors choose M-projection because this corresponds to the *Maximisation* step in an EM algorithm while the E-projection corresponds to the *expectation* step in the EM algorithm. In Brekelmans and Neklyudov (2023), the authors highlight that the Iterative Proportional Fitting procedure is a Expectation-Expectation procedure, i.e. the alternating projections are both E-projections. In contrast, the Iterative Markovian Fitting procedure is a Maximisation-Maximisation procedure, i.e. the alternating projections are both M-projections. In particular, we can define the following sequence of path measures $(\mathbb{P}^n)_{n \in \mathbb{N}}$, where for any $n \in \mathbb{N}$ we have

$$\mathbb{P}^{n+1/2} = \mathrm{argmin}_{\mathbb{P} \in \mathrm{proj}_{\mathcal{M}}} \mathrm{KL}(\mathbb{P}^n \mid \mathbb{P}), \qquad \mathbb{P}^{n+1} = \mathrm{argmin}_{\mathbb{P} \in \mathcal{R}(\mathbb{Q})} \mathrm{KL}(\mathbb{P}^{n+1/2} \mid \mathbb{P}).$$

In addition, we have that

$$\mathbb{P}^{n+1/2} = \mathbb{P}_{v_{\star}^{n+1}}, \qquad v_{\star}^{n+1} = \mathrm{argmin}_v \mathcal{L}(v, \mathbb{P}^n), \qquad \mathbb{P}^{n+1} = \mathrm{argmin}_{\mathbb{P} \in \mathcal{R}(\mathbb{Q})} \mathrm{KL}(\mathbb{P}_{v_{\star}^{n+1}} \mid \mathbb{P}),$$

since we have that $\mathbb{P}^n = \mathbb{P}^{v_{\star}^n}$. Hence, our online procedure Algorithm 1, which corresponds to the discretisation of the flow of path measures (3) can be rewritten as

$$\mathbb{P}^{n+1/2} = \mathbb{P}_{v_{\star}^{n+1}}, \quad v_{\star}^{n+1} = \mathrm{Gradientstep}(\mathcal{L}(v, \mathbb{P}^n)), \quad \mathbb{P}^{n+1} = \mathrm{argmin}_{\mathbb{P} \in \mathcal{R}(\mathbb{Q})} \mathrm{KL}(\mathbb{P}_{v_{\star}^{n+1}} \mid \mathbb{P}).$$

Therefore, our proposed algorithm can be seen as an incremental version of the Maximisation-Maximisation algorithm associated with DSBM instead of an incremental version of the Expectation-Maximisation algorithm discussed in (Neal and Hinton, 1998).

### H.4  Links with finetuning of diffusion models

Algorithm 1 can be seen as a method to finetune bridge matching. Finetuning of diffusion models and flow matching procedures is an active research area. Most of the existing methodologies optimise for an external cost after a pretraining phase. These procedures rely on Reinforcement Learning strategies (Lee et al., 2023; Black et al., 2023; Fan et al., 2024). Recently Direct Preference optimisation (DPO) (Rafailov et al., 2024) has been applied to the finetuning of diffusion models in (Yang et al., 2023; Rafailov et al., 2024). Our approach departs from these works as the objective we minimise is given by the EOT cost. However all of these approaches involve some level of self-play, i.e. are not simulation free.

### H.5  Links with continual learning

Continual learning develops techniques to train models when the dataset changes during the training, usually to solve different tasks De Lange et al. (2021); Parisi et al. (2019); Zając et al. (2023). In the context of diffusion models, continual learning has been investigated in Masip et al. (2023); Zając et al. (2023); Smith et al. (2023). In (Masip et al., 2023), the authors consider a weighted loss between a diffusion model loss and a distillation loss which ensures some consistency between the model being trained and the previous task model. Similarly to our approach this distillation loss is not simulation-free but, contrary to our loss, the clean samples are not obtained by unrolling the diffusion model but by applying a one-step prediction operator. In (Zając et al., 2023), consider different replay buffer techniques to train continual diffusion models and observe that experience replay with a small coefficient can bring improvements. Finally, in (Smith et al., 2023), the authors consider the continual training of a text-to-image diffusion model with LoRA (Hu et al., 2021).

## I  Forward-Forward, Forward-Backward and accumulation of error

In this section, we investigate how error accumulates in the context of DSBM. In practice, we observe similar conclusions in the case of the online version of DSBM. We compare two methods: one which only trains a forward model and one which trains a forward and a backward model.

In what follows, we assume that $\pi_0 = \pi_1 = \mathcal{N}(0, \mathrm{Id})$, we also assume that $\mathbb{Q}$ is associated with $(\sqrt{2}\mathbf{B}_t)_{t \in [0,1]}$. We recall that for any $t \in [0,1]$, we have that

$$\mathbf{X}_t = (1-t)\mathbf{X}_0 + t\mathbf{X}_1 + \sqrt{2t(1-t)}\mathbf{Z}, \qquad \mathbf{Z} \sim \mathcal{N}(0, \mathrm{Id}).$$

We are going to consider to approximate schemes to implement IMF.

**Forward-forward.** First, we consider the following sequence of path measures $(\mathbb{P}^n)_{n \in \mathbb{N}}$. We set $\mathbb{P}^0 = (\pi_0 \otimes \pi_1)\mathbb{Q}_{|0,1}$. For any $n \in \mathbb{N}$, we define $\mathbb{P}^{2n+2} = \mathbb{P}^{2n+1}_{0,1}\mathbb{Q}_{|0,1}$, i.e. $\mathbb{P}^{2n+2} = \mathrm{proj}_{\mathcal{R}}(\mathbb{P}^{2n+1})$. In addition, we define $\mathbb{P}^{2n+1} = \mathrm{proj}^{\varepsilon,\rightarrow}_{\mathcal{M}}(\mathbb{P}^{2n})$ such that $\mathbb{P}^{2n+1}$ is associated with $(\mathbf{X}_t)_{t \in [0,1]}$ where for any $t \in [0,1]$

$$d\mathbf{X}_t = \{(\mathbb{E}_{\mathbb{P}^{2n}_{1|t}}[\mathbf{X}_1 \mid \mathbf{X}_t] - \mathbf{X}_t)/(1-t) + \varepsilon\mathbf{X}_t\}dt + \sqrt{2}d\mathbf{B}_t, \qquad \mathbf{X}_0 \sim \pi_0, \qquad (53)$$

with $\varepsilon \in \mathbb{R}$. Recall that if we define $\bar{\mathbb{P}}^{2n+1} = \mathrm{proj}_{\mathcal{M}}(\mathbb{P}^{2n})$ we have that for any $t \in [0,1]$, $\bar{\mathbb{P}}^{2n+1}$ is associated with $(\mathbf{X}_t)_{t \in [0,1]}$ where for any $t \in [0,1]$

$$d\mathbf{X}_t = (\mathbb{E}_{\mathbb{P}^{2n}_{1|t}}[\mathbf{X}_1 \mid \mathbf{X}_t] - \mathbf{X}_t)/(1-t)dt + \sqrt{2}d\mathbf{B}_t.$$

Hence, $\mathrm{proj}^{\varepsilon,\rightarrow}_{\mathcal{M}}$ corresponds to making an error of order $x \mapsto \varepsilon x$ on the estimated velocity field. Doing so, we now longer have that for any $n \in \mathbb{N}$, $\mathbb{P}^n_1 = \pi_1$. In what follows, we are going to show how the error accumulates for the sequence $\mathbb{P}^n_{0,1}$.

Before stating Proposition I.1, we introduce $f : \mathbb{R}^4 \to \mathbb{R}$ such that for any $c_{0,0}, c_{1,1}, c_{0,1} > 0$ and $t \in [0,1]$

$$f(c_{0,0}, c_{1,1}, c_{0,1}, t) = [-(1-t)c_{0,0} + tc_{1,1} + (1-2t)c_{0,1} - 2t]$$
$$/[(1-t)^2 c_{0,0} + t^2 c_{1,1} + 2t(1-t)c_{0,1} + 2t(1-t)].$$

We define $F(c_{0,0}, c_{1,1}, c_{0,1}, \varepsilon, t) = 2\int_0^t f(c_{0,0}, c_{1,1}, c_{0,1}, s)ds + 2\varepsilon t$. Finally, we define

$$f_{\mathrm{cov}}(c_{0,0}, c_{1,1}, c_{0,1}, \varepsilon) = \exp[\tfrac{1}{2}F(c_{0,0}, c_{1,1}, c_{0,1}, \varepsilon, 1)],$$

as well as

$$f_{\mathrm{var}}(c_{0,0}, c_{1,1}, c_{0,1}, \varepsilon) = \exp[\tfrac{1}{2}F(c_{0,0}, c_{1,1}, c_{0,1}, \varepsilon, 1)](1 + 2\int_0^1 \exp[-F(c_{0,0}, c_{1,1}, c_{0,1}, \varepsilon, s)]ds).$$

**Proposition I.1 (Forward-Forward updates):** *For any* $n \in \mathbb{N}$, *we have that* $\mathbb{P}^{2n+1}_{0,1} = \mathcal{N}(0, \Sigma^{n+1}\mathrm{Id})$ *where*

$$\Sigma^{n+1} = \begin{pmatrix} \mathrm{Id} & c^{n+1}_{0,1}\mathrm{Id} \\ c^{n+1}_{0,1}\mathrm{Id} & c^{n+1}_{1,1}\mathrm{Id} \end{pmatrix},$$

*and for any* $n \in \mathbb{N}$

$$c^{n+1}_{1,1} = f_{\mathrm{var}}(1, c^n_{1,1}, c^n_{0,1}, \varepsilon),$$
$$c^{n+1}_{0,1} = f_{\mathrm{cov}}(1, c^n_{1,1}, c^n_{0,1}, \varepsilon).$$

*Proof.* Let $\mathbb{P} = (\mathbb{P}_{0,1})\mathbb{Q}_{|0,1}$ where $\mathbb{P}_{0,1}$ is a Gaussian random variable with zero mean and covariance matrix $\Sigma \in \mathbb{R}^{2d \times 2d}$ such that

$$\Sigma = \begin{pmatrix} \mathrm{Id} & c_{0,1}\mathrm{Id} \\ c_{0,1}\mathrm{Id} & c_{1,1}\mathrm{Id} \end{pmatrix},$$

where $\mathrm{Id}$ is the $d$-dimensional identity matrix and we assume that $c_{0,1}, c_{1,1} > 0$. We denote $\mathbb{P}^\star = \mathrm{proj}^{\varepsilon,\rightarrow}_{\mathcal{M}}(\mathbb{P})$. We have that $\mathbb{P}_{1|t}$ is a Gaussian random variable with zero mean. We now compute its covariance matrix. First, we have that

$$\mathbb{E}[\mathbf{X}_t\mathbf{X}_1^\top] = (1-t)\mathbb{E}[\mathbf{X}_0\mathbf{X}_1^\top] + t\mathbb{E}[\mathbf{X}_1\mathbf{X}_1^\top] = [(1-t)c_{0,1} + tc_{1,1}]\mathrm{Id}.$$

We also have that

$$\mathbb{E}[\mathbf{X}_t\mathbf{X}_t^\top] = (1-t)^2\mathbb{E}[\mathbf{X}_0\mathbf{X}_0^\top] + t(1-t)(\mathbb{E}[\mathbf{X}_1\mathbf{X}_0^\top] + \mathbb{E}[\mathbf{X}_1\mathbf{X}_0^\top]) + t^2\mathbb{E}[\mathbf{X}_1\mathbf{X}_1^\top] + 2t(1-t)\mathrm{Id}$$
$$= [(1-t)^2 + t^2c_{1,1} + 2t(1-t)c_{0,1} + 2t(1-t)]\mathrm{Id}$$
$$= [1 - t^2 + t^2c_{1,1} + 2t(1-t)c_{0,1}]\mathrm{Id}.$$

Therefore, we get that for any $t \in [0,1]$ and $x_t \in \mathbb{R}^d$

$$\mathbb{E}_{\mathbb{P}_{1|t}}[\mathbf{X}_1 \mid \mathbf{X}_t = x_t] = ([(1-t)c_{0,1} + tc_{1,1}]/[1 - t^2 + t^2c_{1,1} + 2t(1-t)c_{0,1}])x_t.$$

Hence, we have that for any $t \in [0,1]$ and $x_t \in \mathbb{R}^d$

$$\mathbb{E}_{\mathbb{P}_{1|t}}[\mathbf{X}_1 \mid \mathbf{X}_t = x_t] - x_t = ([(1-t)c_{0,1} + tc_{1,1}]/[1 - t^2 + t^2c_{1,1} + 2t(1-t)c_{0,1}] - 1)x_t$$
$$= ([(1-t)c_{0,1} + tc_{1,1} - 1 + t^2 - t^2c_{1,1} - 2t(1-t)c_{0,1}]/[1 - t^2 + t^2c_{1,1} + 2t(1-t)c_{0,1}])x_t$$
$$= ([(1-t)(1-2t)c_{0,1} + t(1-t)c_{1,1} - 1 + t^2]/[1 - t^2 + t^2c_{1,1} + 2t(1-t)c_{0,1}])x_t$$
$$= (1-t)([(1-2t)c_{0,1} + tc_{1,1} - 1 - t]/[1 - t^2 + t^2c_{1,1} + 2t(1-t)c_{0,1}])x_t.$$

So it follows that

$$(\mathbb{E}_{\mathbb{P}_{1|t}}[\mathbf{X}_1 \mid \mathbf{X}_t = x_t] - x_t)/(1-t)$$
$$= ([(1-2t)c_{0,1} + tc_{1,1} - 1 - t]/[1 - t^2 + t^2c_{1,1} + 2t(1-t)c_{0,1}])x_t. \quad (54)$$

Note that if we set $c_{0,1} = c^2$ and $c_{1,1} = 1$, we recover (Shi et al., 2023, Lemma 13) with $\sigma = 2$. Denote $\mathbb{P}^\star = \mathrm{proj}_{\mathcal{M}}^{\varepsilon,\rightarrow}(\mathbb{P})$. Combining (54) and (53) we get that $\mathbb{P}^\star$ is associated with $(\mathbf{X}_t)_{t\in[0,1]}$ such that for any $t \in [0,1]$ we have

$$\mathrm{d}\mathbf{X}_t = \{([(1-2t)c_{0,1} + tc_{1,1} - 1 - t]/[1 - t^2 + t^2c_{1,1} + 2t(1-t)c_{0,1}]) + \varepsilon\}\mathbf{X}_t\mathrm{d}t + \sqrt{2}\mathrm{d}\mathbf{B}_t.$$

Hence, we get that

$$\mathbf{X}_t = \exp[\tfrac{1}{2}G(t, c_{0,1}, c_{1,1}, \varepsilon)]\mathbf{X}_0 + \left(2\int_0^t \exp[-G(s, c_{0,1}, c_{1,1}, \varepsilon)]\mathrm{d}s \exp[G(t, c_{0,1}, c_{1,1}, \varepsilon)]\right)^{1/2}\mathbf{Z},$$

where $\mathbf{Z} \sim \mathcal{N}(0, \mathrm{Id})$ is independent from $\mathbf{X}_0$ and for any $t \in [0,1]$, $c_{0,1}, c_{1,1}, \varepsilon > 0$ we have

$$G(t, c_{0,1}, c_{1,1}, \varepsilon) = 2\int_0^t [(1-2t)c_{0,1} + tc_{1,1} - 1 - t]/[1 - t^2 + t^2c_{1,1} + 2t(1-t)c_{0,1}]\mathrm{d}t + 2\varepsilon t.$$

In addition, we define

$$g_{\mathrm{cov}}(c_{0,1}, c_{1,1}, \varepsilon) = \exp[G(1, c_{0,1}, c_{1,1}, \varepsilon)],$$
$$g_{\mathrm{var}}(c_{0,1}, c_{1,1}, \varepsilon) = \exp[G(1, c_{0,1}, c_{1,1}, \varepsilon)]\left(1 + 2\int_0^1 \exp[-G(t, c_{0,1}, c_{1,1}, \varepsilon)]\mathrm{d}t\right).$$

Hence, we have that

$$\mathbb{E}[\mathbf{X}_0\mathbf{X}_1^\top] = g_{\mathrm{cov}}(c_{0,1}, c_{1,1}, \varepsilon)\mathrm{Id}, \qquad \mathbb{E}[\mathbf{X}_1\mathbf{X}_1^\top] = g_{\mathrm{var}}(c_{0,1}, c_{1,1}, \varepsilon)\mathrm{Id}.$$

Therefore, since for any $n \in \mathbb{N}$, we have that $\mathbb{P}^{2n+1} = \mathrm{proj}_{\mathcal{M}}^{\varepsilon,\rightarrow}(\mathbb{P}^{2n})$ and $\mathbb{P}^{2n+2} = \mathbb{P}_{0,1}^{2n+1}\mathbb{Q}_{|0,1}$, we define $(c_{0,1}^n, c_{1,1}^n)_{n\in\mathbb{N}}$ such that for any $n \in \mathbb{N}$

$$\mathbb{E}_{\mathbb{P}^{2n}}[\mathbf{X}_0\mathbf{X}_1^\top] = c_{0,1}^n\mathrm{Id}, \qquad \mathbb{E}_{\mathbb{P}^{2n}}[\mathbf{X}_1\mathbf{X}_1^\top] = c_{1,1}^n\mathrm{Id}.$$

Note that for any $n \in \mathbb{N}$, we have that

$$\mathbb{E}_{\mathbb{P}^{2n+1}}[\mathbf{X}_0\mathbf{X}_1^\top] = c_{0,1}^{n+1}\mathrm{Id}, \qquad \mathbb{E}_{\mathbb{P}^{2n+1}}[\mathbf{X}_1\mathbf{X}_1^\top] = c_{1,1}^{n+1}\mathrm{Id}.$$

Since $\mathbb{P}^0 = (\pi_0 \otimes \pi_1)\mathbb{Q}_{|0,1}$ we get that $c_{0,1}^0 = 0$ and $c_{1,1} = 1$. We have that for any $n \in \mathbb{N}$

$$c_{0,1}^{n+1} = g_{\mathrm{cov}}(c_{0,1}^n, c_{1,1}^n, \varepsilon), \qquad c_{1,1}^{n+1} = g_{\mathrm{var}}(c_{1,1}^n, c_{1,1}^n, \varepsilon),$$

which concludes the proof. □

**Forward-backward.** Next, we consider the following sequences of path measures $(\mathbb{P}^{n,\rightarrow})_{n\in\mathbb{N}}$ and $(\mathbb{P}^{n,\leftarrow})_{n\in\mathbb{N}}$. We set $\mathbb{P}^{0,\rightarrow} = \mathbb{P}^{0,\leftarrow} = (\pi_0 \otimes \pi_1)\mathbb{Q}_{|0,1}$. For any $n \in \mathbb{N}$, we define $\mathbb{P}^{2n+2,\rightarrow} = \mathbb{P}^{2n+1,\leftarrow}_{0,1}\mathbb{Q}_{|0,1}$ and $\mathbb{P}^{2n+2,\leftarrow} = \mathbb{P}^{2n+1,\rightarrow}_{0,1}\mathbb{Q}_{|0,1}$, i.e. $\mathbb{P}^{2n+2,\rightarrow} = \mathrm{proj}_{\mathcal{R}}(\mathbb{P}^{2n+1,\leftarrow})$ and $\mathbb{P}^{2n+2,\leftarrow} = \mathrm{proj}_{\mathcal{R}}(\mathbb{P}^{2n+1,\rightarrow})$. In addition, we define $\mathbb{P}^{2n+1,\rightarrow} = \mathrm{proj}_{\mathcal{M}}^{\varepsilon,\rightarrow}(\mathbb{P}^{2n,\leftarrow})$ such that for any $t \in [0,1]$, $\mathbb{P}^{2n+1,\rightarrow}$ is associated with $(\mathbf{X}_t)_{t\in[0,1]}$ where

$$d\mathbf{X}_t = \{(\mathbb{E}_{\mathbb{P}^{2n,\rightarrow}_{1|t}}[\mathbf{X}_1 \mid \mathbf{X}_t] - \mathbf{X}_t)/(1-t) + \varepsilon\mathbf{X}_t\}dt + \sqrt{2}d\mathbf{B}_t, \qquad \mathbf{X}_0 \sim \pi_0,$$

with $\varepsilon \in \mathbb{R}$. Similarly, we define $\mathbb{P}^{2n+1,\leftarrow} = \mathrm{proj}_{\mathcal{M}}^{\varepsilon,\leftarrow}(\mathbb{P}^{2n,\rightarrow})$ such that for any $t \in [0,1]$, $\mathbb{P}^{2n+1,\leftarrow}$ is associated with $(\mathbf{Y}_{1-t})_{t\in[0,1]}$ where

$$d\mathbf{Y}_t = \{(\mathbb{E}_{\mathbb{P}^{2n,\leftarrow}_{0|t}}[\mathbf{X}_0 \mid \mathbf{Y}_t] - \mathbf{Y}_t)/(1-t) + \varepsilon\mathbf{Y}_t\}dt + \sqrt{2}d\mathbf{B}_t, \qquad \mathbf{Y}_0 \sim \pi_1.$$

**Proposition I.2 (Forward-Backward updates):** *For any $n \in \mathbb{N}$, we have that $\mathbb{P}^{2n+1,\rightarrow}_{0,1} = \mathcal{N}(0, \Sigma^{n+1,\rightarrow}\mathrm{Id})$ and $\mathbb{P}^{2n+1,\leftarrow}_{0,1} = \mathcal{N}(0, \Sigma^{n+1,\leftarrow}\mathrm{Id})$ where*

$$\Sigma^{n+1,\rightarrow} = \begin{pmatrix} \mathrm{Id} & c_{0,1}^{n+1,\rightarrow}\mathrm{Id} \\ c_{0,1}^{n+1,\rightarrow}\mathrm{Id} & c_{1,1}^{n+1,\rightarrow}\mathrm{Id} \end{pmatrix}, \qquad \Sigma^{n+1,\leftarrow} = \begin{pmatrix} c_{0,0}^{n+1,\leftarrow}\mathrm{Id} & c_{0,1}^{n+1,\leftarrow}\mathrm{Id} \\ c_{0,1}^{n+1,\leftarrow}\mathrm{Id} & \mathrm{Id} \end{pmatrix},$$

*and for any $n \in \mathbb{N}$*

$$c_{1,1}^{n+1,\rightarrow} = f_{\mathrm{var}}(c_{0,0}^{n,\leftarrow}, 1, c_{0,1}^{n,\leftarrow}, \varepsilon),$$
$$c_{0,1}^{n+1,\rightarrow} = f_{\mathrm{cov}}(c_{0,0}^{n,\leftarrow}, 1, c_{0,1}^{n,\leftarrow}, \varepsilon),$$
$$c_{0,0}^{n+1,\leftarrow} = f_{\mathrm{var}}(1, c_{1,1}^{n,\rightarrow}, c_{0,1}^{n,\rightarrow}, \varepsilon),$$
$$c_{0,1}^{n+1,\leftarrow} = f_{\mathrm{cov}}(1, c_{1,1}^{n,\rightarrow}, c_{0,1}^{n,\rightarrow}, \varepsilon).$$

The proof is similar to the one of Proposition I.1.

*Proof.* Let $\mathbb{P} = (\mathbb{P}_{0,1})\mathbb{Q}_{|0,1}$ where $\mathbb{P}_{0,1}$ is a Gaussian random variable with zero mean and covariance matrix $\Sigma \in \mathbb{R}^{2d \times 2d}$ such that

$$\Sigma = \begin{pmatrix} c_{0,0}\mathrm{Id} & c_{0,1}\mathrm{Id} \\ c_{0,1}\mathrm{Id} & \mathrm{Id} \end{pmatrix},$$

where Id is the $d$-dimensional identity matrix and $c_{0,1}, c_{0,0} > 0$. We denote $\mathbb{P}^\star = \mathrm{proj}_{\mathcal{M}}^{\varepsilon,\rightarrow}(\mathbb{P})$. We have that $\mathbb{P}_{1|t}$ is a Gaussian random variable with zero mean. We now compute its covariance matrix. First, we have that

$$\mathbb{E}[\mathbf{X}_t\mathbf{X}_1^\top] = (1-t)\mathbb{E}[\mathbf{X}_0\mathbf{X}_1^\top] + t\mathbb{E}[\mathbf{X}_1\mathbf{X}_1^\top] = [(1-t)c_{0,1} + t]\mathrm{Id}.$$

We also have that

$$\mathbb{E}[\mathbf{X}_t\mathbf{X}_t^\top] = (1-t)^2\mathbb{E}[\mathbf{X}_0\mathbf{X}_0^\top] + t(1-t)(\mathbb{E}[\mathbf{X}_1\mathbf{X}_0^\top] + \mathbb{E}[\mathbf{X}_1\mathbf{X}_0^\top]) + t^2\mathbb{E}[\mathbf{X}_1\mathbf{X}_1^\top] + 2t(1-t)\mathrm{Id}$$
$$= [(1-t)^2c_{0,0} + t^2 + 2t(1-t)c_{0,1} + 2t(1-t)]\mathrm{Id}$$
$$= [2t - t^2 + (1-t)^2c_{0,0} + 2t(1-t)c_{0,1}]\mathrm{Id}.$$

Therefore, we get that for any $t \in [0,1]$ and $x_t \in \mathbb{R}^d$

$$\mathbb{E}_{\mathbb{P}_{1|t}}[\mathbf{X}_1 \mid \mathbf{X}_t = x_t] = ([(1-t)c_{0,1} + t]/[2t - t^2 + (1-t)^2c_{0,0} + 2t(1-t)c_{0,1}])x_t.$$

Hence, we have that for any $t \in [0,1]$ and $x_t \in \mathbb{R}^d$

$$\mathbb{E}_{\mathbb{P}_{1|t}}[\mathbf{X}_1 \mid \mathbf{X}_t = x_t] - x_t = ([(1-t)c_{0,1} + t]/[2t - t^2 + (1-t)^2c_{0,0} + 2t(1-t)c_{0,1}] - 1)x_t$$
$$= ([(1-t)c_{0,1} + t - 2t + t^2 - (1-t)^2c_{0,0} - 2t(1-t)c_{0,1}]$$
$$/[2t - t^2 + (1-t)^2c_{0,0} + 2t(1-t)c_{0,1}])x_t$$
$$= ([(1-t)(1-2t)c_{0,1} - (1-t)^2c_{0,0} - t(1-t)]/[2t - t^2 + (1-t)^2c_{0,0} + 2t(1-t)c_{0,1}])x_t$$
$$= (1-t)([(1-2t)c_{0,1} - (1-t)c_{0,0} - t]/[2t - t^2 + (1-t)^2c_{0,0} + 2t(1-t)c_{0,1}])x_t.$$

Finally, we have that for any $t \in [0,1]$ and $x_t \in \mathbb{R}^d$

$$(\mathbb{E}_{\mathbb{P}_{1|t}}[\mathbf{X}_1 \mid \mathbf{X}_t = x_t] - x_t)/(1-t)$$
$$= ([[(1-2t)c_{0,1} - (1-t)c_{0,0} - t]/[2t - t^2 + (1-t)^2 c_{0,0} + 2t(1-t)c_{0,1}])x_t. \quad (55)$$

Note that if we set $c_{0,1} = c^2$ and $c_{0,0} = 1$, we recover (Shi et al., 2023, Lemma 13) with $\sigma = 2$. Denote $\mathbb{P}^\star = \mathrm{proj}_{\mathcal{M}}^{\varepsilon,\to}(\mathbb{P})$. Combining (55) and (53) we get that $\mathbb{P}^\star$ is associated with $(\mathbf{X}_t)_{t \in [0,1]}$ such that for any $t \in [0,1]$ we have

$$d\mathbf{X}_t = \{([(1-2t)c_{0,1} - (1-t)c_{0,0} - t]/[2t - t^2 + (1-t)^2 c_{0,0} + 2t(1-t)c_{0,1}]) + \varepsilon\}\mathbf{X}_t dt + \sqrt{2}d\mathbf{B}_t.$$

Hence, we get that

$$\mathbf{X}_t = \exp[\tfrac{1}{2}H(t, c_{0,1}, c_{0,0}, \varepsilon)]\mathbf{X}_0 + (2\int_0^t \exp[-H(s, c_{0,1}, c_{0,0}, \varepsilon)]ds \exp[H(t, c_{0,1}, c_{0,0}, \varepsilon)])^{1/2}\mathbf{Z},$$

where $\mathbf{Z} \sim \mathcal{N}(0, \mathrm{Id})$ is independent from $\mathbf{X}_0$ and for any $t \in [0,1]$, $c_{0,1}, c_{1,1}, \varepsilon > 0$ we have

$$H(t, c_{0,1}, c_{0,0}, \varepsilon) = 2\int_0^t [(1-2t)c_{0,1} - (1-t)c_{0,0} - t]/[2t - t^2 + (1-t)^2 c_{0,0} + 2t(1-t)c_{0,1}]dt + 2\varepsilon t.$$

In addition, we define

$$g_{\mathrm{cov}}(c_{0,1}, c_{0,0}, \varepsilon) = \exp[\tfrac{1}{2}H(1, c_{0,1}, c_{0,0}, \varepsilon)],$$

$$g_{\mathrm{var}}(c_{0,1}, c_{0,0}, \varepsilon) = \exp[H(1, c_{0,1}, c_{0,0}, \varepsilon)]\Big(1 + 2\int_0^1 \exp[-H(t, c_{0,1}, c_{0,0}, \varepsilon)]dt\Big).$$

Hence, we have that

$$\mathbb{E}[\mathbf{X}_0\mathbf{X}_1^\top] = g_{\mathrm{cov}}(c_{0,1}, c_{0,0}, \varepsilon)\mathrm{Id}, \qquad \mathbb{E}[\mathbf{X}_1\mathbf{X}_1^\top] = g_{\mathrm{var}}(c_{0,1}, c_{0,0}, \varepsilon)\mathrm{Id}. \quad (56)$$

Remember that $\mathbb{P}^{0,\to} = \mathbb{P}^{0,\leftarrow} = (\pi_0 \otimes \pi_1)\mathbb{Q}_{|0,1}$. In addition, for any $n \in \mathbb{N}$, we have $\mathbb{P}^{2n+2,\to} = \mathbb{P}^{2n+1,\leftarrow}_{0,1}\mathbb{Q}_{|0,1}$ and $\mathbb{P}^{2n+2,\leftarrow} = \mathbb{P}^{2n+1,\to}_{0,1}\mathbb{Q}_{|0,1}$, i.e. $\mathbb{P}^{2n+2,\to} = \mathrm{proj}_{\mathcal{R}}(\mathbb{P}^{2n+1,\leftarrow})$ and $\mathbb{P}^{2n+2,\leftarrow} = \mathrm{proj}_{\mathcal{R}}(\mathbb{P}^{2n+1,\to})$. In addition, we also have $\mathbb{P}^{2n+1,\to} = \mathrm{proj}_{\mathcal{M}}^{\varepsilon,\to}(\mathbb{P}^{2n,\leftarrow})$ and $\mathbb{P}^{2n+1,\leftarrow} = \mathrm{proj}_{\mathcal{M}}^{\varepsilon,\leftarrow}(\mathbb{P}^{2n,\to})$. We also define $(c_{0,1}^{n,\to}, c_{1,1}^{n,\to})_{n \in \mathbb{N}}$ such that for any $n \in \mathbb{N}$

$$\mathbb{E}_{\mathbb{P}^{2n,\to}}[\mathbf{X}_0\mathbf{X}_1^\top] = c_{0,1}^{n,\to}\mathrm{Id}, \qquad \mathbb{E}_{\mathbb{P}^{2n,\to}}[\mathbf{X}_1\mathbf{X}_1^\top] = c_{1,1}^{n,\to}\mathrm{Id}.$$

Finally, we define $(c_{0,1}^{n,\leftarrow}, c_{1,1}^{n,\leftarrow})_{n \in \mathbb{N}}$ such that for any $n \in \mathbb{N}$

$$\mathbb{E}_{\mathbb{P}^{2n,\leftarrow}}[\mathbf{X}_0\mathbf{X}_1^\top] = c_{0,1}^{n,\leftarrow}\mathrm{Id}, \qquad \mathbb{E}_{\mathbb{P}^{2n,\leftarrow}}[\mathbf{X}_0\mathbf{X}_0^\top] = c_{0,0}^{n,\leftarrow}\mathrm{Id}.$$

Using this definition and (56) we get that for any $n \in \mathbb{N}$

$$c_{0,1}^{n+1,\to} = g_{\mathrm{cov}}(c_{0,1}^{n,\leftarrow}, c_{0,0}^{n,\leftarrow}, \varepsilon), \qquad c_{1,1}^{n+1,\to} = g_{\mathrm{var}}(c_{0,1}^{n,\leftarrow}, c_{0,0}^{n,\leftarrow}, \varepsilon),$$
$$c_{0,1}^{n+1,\leftarrow} = g_{\mathrm{cov}}(c_{0,1}^{n,\to}, c_{1,1}^{n,\to}, \varepsilon), \qquad c_{0,0}^{n+1,\leftarrow} = g_{\mathrm{var}}(c_{0,1}^{n,\to}, c_{1,1}^{n,\to}, \varepsilon).$$

In addition, we have that $c_{0,1}^{n,\leftarrow} = c_{0,1}^{n,\to} = 0$ and $c_{1,1}^{0,\to} = c_{0,0}^{0,\leftarrow} = 1$. This concludes the proof. $\qquad\square$

**Error accumulation.** In Proposition I.1 and Proposition I.2, we derive the sequences corresponding to the evolution of the variance and the covariance throughout the DSBM iterations in forward-forward mode or forward-backward mode. In what follows, we showcase the behavior of these sequences for different values of $\varepsilon > 0$. We recall that $\varepsilon$ corresponds to the error made in the Markov projection, i.e. $\mathrm{proj}_{\mathcal{M}}$ is replaced by $\mathrm{proj}_{\mathcal{M}}^{\varepsilon,\to}$ in the forward-forward mode and $\mathrm{proj}_{\mathcal{M}}$ is replaced by $\mathrm{proj}_{\mathcal{M}}^{\varepsilon,\to}$ and $\mathrm{proj}_{\mathcal{M}}^{\varepsilon,\leftarrow}$ in the forward-backward mode. First, if we consider the perfect scenario, i.e. $\varepsilon = 0$, then we observe that both the forward-forward mode and the forward-backward mode satisfy that $\mathbb{E}_{\mathbb{P}^{2n}}[\mathbf{X}_1\mathbf{X}_1^\top] = \mathrm{Id}$, see Figure 10 and Figure 11. Additionally, we can show that in the perfect scenario, i.e. $\varepsilon = 0$, then both the forward-forward mode and the forward-backward mode satisfy that $\lim_{n \to +\infty} \mathbb{E}_{\mathbb{P}^{2n}}[\mathbf{X}_1\mathbf{X}_0^\top] = (\sqrt{2} - 1)\mathrm{Id}$, see Figure 10 and Figure 11. However, as $\varepsilon$ increases the behavior between the forward-forward sequence and the forward-backward sequence significantly differs. More precisely, the error explodes as $\varepsilon$ increases along the DSBM iteration for the forward-forward mode. On the contrary, in the forward-backward mode, the error remains bounded along the DSBM iterations, see Figure 10 and Figure 11.

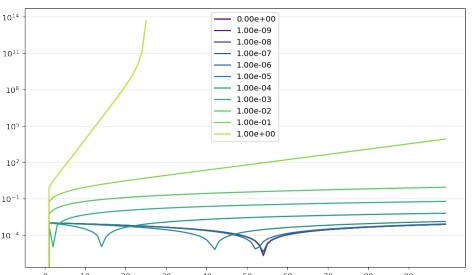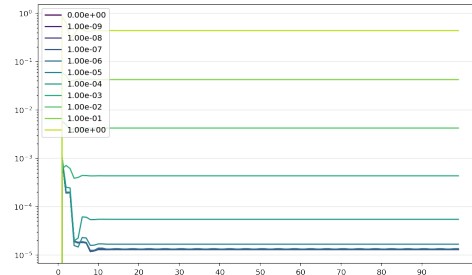

Figure 10: Evolution of $(\|\mathbb{E}_{\mathbb{P}^{2n}}[\mathbf{X}_1 \mathbf{X}_1^\top] - \mathrm{Id}\|)_{n \in \mathbb{N}}$ in log-space along DSBM iterations (x-axis). Different curves correspond to different values of $\varepsilon$, i.e. the larger $\varepsilon$ the larger the error in the Markovian projection. Left: evolution in the forward-forward mode. Right: evolution in the forward-backward mode.

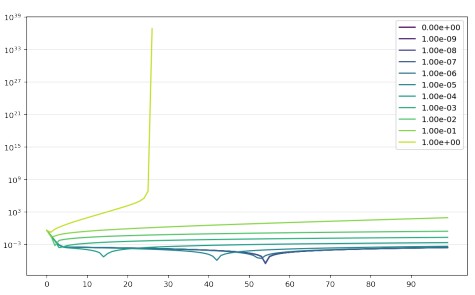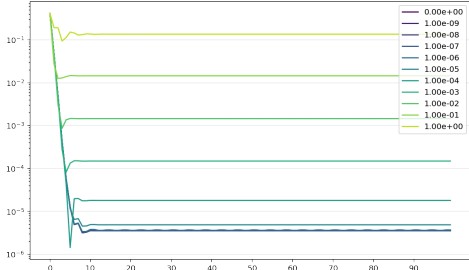

Figure 11: Evolution of $(\|\mathbb{E}_{\mathbb{P}^{2n}}[\mathbf{X}_1 \mathbf{X}_0^\top] - \mathrm{Id}\|)_{n \in \mathbb{N}}$ in log-space along DSBM iterations (x-axis). Different curves correspond to different values of $\varepsilon$, i.e. the larger $\varepsilon$ the larger the error in the Markovian projection. Left: evolution in the forward-forward mode. Right: evolution in the forward-backward mode.

## J Preconditioning of the loss function

In this section, we provide details on the scaling of the loss function we implement when training our online version of DSBM. We adapt the method of (Karras et al., 2022, Appendix B.2) to the case of bridge matching. We only present our derivations in the case of the forward training of the online version of DSBM, i.e. (9). The preconditioning of the loss described in this setting can be readily extended to the forward-backward loss we consider in practice, i.e. the parametric version of (10).

We consider the following objective function for any $t \in [0, 1]$

$$\ell_t = \lambda_t \mathbb{E}_{\mathbb{P}}[\|c_t^o \mathrm{nn}_t^\theta(c_t^i \mathbf{X}_t) + c_t^s \mathbf{X}_t - \tfrac{\mathbf{X}_1 - \mathbf{X}_t}{1-t}\|^2]. \tag{57}$$

We also define for any $t \in [0, 1]$ and $x_t \in \mathbb{R}^d$, $v_t^\theta(x_t) = c_t^o \mathrm{nn}_t^\theta(c_t^i x_t) + c_t^s x_t$. Hence, $c_t^i$ is an input scaling function, $c_t^o$ is an output scaling function and $c_t^s$ is a skip-connection function. During the training of the online version of DSBM, $\mathbb{P}$ will be given by $\mathbb{P}^n$, where $\mathbb{P}^n = \mathbb{P}_{v^{\theta_n}}$, where the sequence $(\theta_n)_{n \in \mathbb{N}}$ is given by (9). Here, we apply the principles of Karras et al. (2022) to the case where $\mathbb{P} = (\pi_0 \otimes \pi_1)\mathbb{Q}_{|0,1}$, i.e. at initialisation of the sequence. In what follows, we assume that $\mathbb{E}_{\pi_0}[\|\mathbf{X}_0\|^2] = \mathbb{E}_{\pi_1}[\|\mathbf{X}_1\|^2] = d$. Note that our considerations can be generalised to $\mathbb{E}_{\pi_0}[\|\mathbf{X}_0\|^2] = \sigma_0^2 d$ and $\mathbb{E}_{\pi_1}[\|\mathbf{X}_1\|^2] = \sigma_1^2 d$. We also have that

$$\mathbf{X}_t = (1 - t)\mathbf{X}_0 + t\mathbf{X}_1 + \sqrt{\varepsilon t(1 - t)}\mathbf{Z}, \qquad \mathbf{Z} \sim \mathcal{N}(0, \mathrm{Id}). \tag{58}$$

Using (58), we have that for any $t \in [0, 1]$

$$\begin{aligned}\mathbb{E}_{\mathbb{P}_t}[\|\mathbf{X}_t\|^2] &= (1 - t)^2 \mathbb{E}_{\pi_0}[\|\mathbf{X}_0\|^2] + t^2 \mathbb{E}_{\pi_1}[\|\mathbf{X}_1\|^2] + \varepsilon t(1 - t)d \\ &= [(1 - t)^2 + t^2 + \varepsilon t(1 - t)]d.\end{aligned}$$

We set $c_t^i$ so that $\mathbb{E}[\|c_t^i \mathbf{X}_t\|^2] = d$ for every $t \in [0, 1]$. Hence, we have that for any $t \in [0, 1]$

$$c_t^i = 1/\sqrt{(1 - t)^2 + t^2 + \varepsilon t(1 - t)}.$$

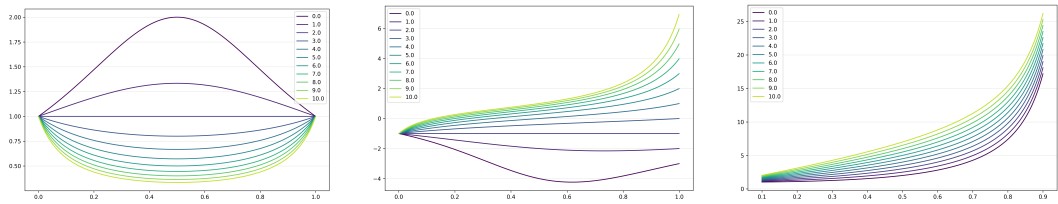

Figure 12: From left to right $((c_t^i)^2)_{t\in[0,1]}$, $(c_t^s)_{t\in[0,1]}$ and $((c_t^o)^2)_{t\in[0,1]}$ for different values of $\varepsilon \in [0,10]$.

Next, we rewrite (57). For any $t \in [0,1]$ we have that

$$
\begin{aligned}
\ell_t &= \lambda_t \mathbb{E}_{\mathbb{P}}[\|c_t^o \mathrm{nn}_t^\theta(c_t^i \mathbf{X}_t) + c_t^s \mathbf{X}_t - \tfrac{\mathbf{X}_1 - \mathbf{X}_t}{1-t}\|^2] \\
&= (c_t^o)^2 \lambda_t \mathbb{E}_{\mathbb{P}}[\|\mathrm{nn}_t^\theta(c_t^i \mathbf{X}_t) - [-c_t^s \mathbf{X}_t + \tfrac{\mathbf{X}_1 - \mathbf{X}_t}{1-t}]/c_t^o\|^2] \\
&= (c_t^o)^2 \lambda_t \mathbb{E}_{\mathbb{P}}[\|\mathrm{nn}_t^\theta(c_t^i \mathbf{X}_t) - [\tfrac{\mathbf{X}_1 - (1 + c_t^s(1-t))\mathbf{X}_t}{1-t}]/c_t^o\|^2]
\end{aligned}
$$

Hence, we get that for any $t \in [0,1]$, $\mathbf{T}_t = [\tfrac{\mathbf{X}_1 - (1 + c_t^s(1-t))\mathbf{X}_t}{1-t}]/c_t^o$ is the target of the network in the regression loss. We are going to fix $c_t^o$ and $c_t^s$ such that i) $\mathbb{E}[\|\mathbf{T}_t\|^2] = d$, ii) $c_t^o$ is as small as possible in order not to minimise the error propagation made by the neural network. Using (58), we have that for any $t \in [0,1]$

$$
\begin{aligned}
\mathbb{E}_{\mathbb{P}_{1,t}}[\|\mathbf{X}_1 - (1 + c_t^s(1-t))\mathbf{X}_t\|^2] &= (1 + c_t^s(1-t))^2 \mathbb{E}_{\mathbb{P}_t}[\|\mathbf{X}_t\|^2] + \mathbb{E}_{\pi_1}[\|\mathbf{X}_1\|^2] \\
&\quad - 2(1 + c_t^s(1-t))\mathbb{E}_{\mathbb{P}_{1,t}}[\langle \mathbf{X}_t, \mathbf{X}_1\rangle] \\
&= (1 + c_t^s(1-t))^2 \mathbb{E}_{\mathbb{P}_t}[\|\mathbf{X}_t\|^2] + d - 2(1 + c_t^s(1-t))td
\end{aligned}
$$

Hence, we get that for any $t \in [0,1]$

$$
(c_t^o)^2 = ((1 + c_t^s(1-t))^2 \mathbb{E}_{\mathbb{P}_t}[\|\mathbf{X}_t\|^2]/d + 1 - 2(1 + c_t^s(1-t))t)/(1-t)^2.
$$

We now minimise $(c_t^o)^2$ with respect to $(1 + c_t^s(1-t))$. We get that

$$
1 + c_t^s(1-t) = t/(\mathbb{E}_{\mathbb{P}_t}[\|\mathbf{X}_t\|^2]/d).
$$

Hence, we get that $c_t^s = t/[(1-t)((1-t)^2 + t^2 + \varepsilon t(1-t))] - 1/(1-t)$. With that choice, we get that for any $t \in [0,1]$

$$
(c_t^o)^2 = (1 - t^2/((1-t)^2 + t^2 + \varepsilon t(1-t)))/(1-t)^2.
$$

In Karras et al. (2022), the weighting function $\lambda_t$ is set so that the weight in front of the regression loss is equal to one for all times $t \in [0,1]$. Hence, Karras et al. (2022) suggests to set $\lambda_t = 1/(c_t^o)^2$. However, in practice we observe better results by letting $\lambda_t = 1$. This means that the effective weight is given by $1/(c_t^o)^2$. Therefore, for any $t \in [0,1]$ we have

$$
\begin{aligned}
(c_t^i)^2 &= (1 + (\varepsilon - 2)t(1-t))^{-1}, \\
c_t^s &= ((\varepsilon - 2)t - 1)/(1 + (\varepsilon - 2)t(1-t)), \\
(c_t^o)^2 &= (1 + t + (\varepsilon - 2)t(1-t))/(1-t).
\end{aligned}
$$

## K  Experimental details

In this section, we delve deeper into the specifics of each experiment, implementation details, and share additional results.

We consider two ways of parameterising the vector fields: as in DSBM, we can use two separate neural networks to approximate the forward and backward vector fields, or we can use a single neural network that is conditioned on the direction. In the latter case, we do the conditioning in a similar fashion to how DDM's neural networks, U-Nets or MLPs, are conditioned on time embeddings. After all, if we work with continuous time variables $t \in [0,1]$, then the direction signal $s \in \{0,1\}$

can be thought of as a target time. Thus, we perform the same initial transformations on $t$ and $s$, i.e. computing sinusoidal embeddings followed by a 2-layer MLP, and use the concatenated outputs in adaptive group normalisation layers (Dhariwal and Nichol, 2021; Hudson et al., 2023; Perez et al., 2018).

To optimise our networks, we use Adam (Kingma and Ba, 2015) with $\beta = (0.9, 0.999)$, and we modify the gradients to keep their global norm below 1.0. We re-initialise the optimiser's state when the finetuning phase starts.

All image samples in the paper are generated using EMA parameters as it has been known to increase the visual quality of resulting images (Song and Ermon, 2020). Sampling is also the integral part of DSBM's finetuning stage, both iterative and online. Here, we have two options: sample with EMA or non-EMA parameters. The non-EMA sampling might be easier to implement, while EMA sampling results in a more stable training and slightly better quality, e.g. see AFHQ samples in Figure 23 and Figure 24 for comparison.

For every model used in the paper, we provide hyperparameters in Table 3.

| | 2D | Gaussian | MNIST | AFHQ-64 | AFHQ-256 |
|---|---|---|---|---|---|
| Channels/hidden units | 256 | 256 | 64 | 128 | 128 |
| Depth | 3 | 3 | 2 | 4 | 4 |
| Channels multiple | n/a | n/a | 1, 2, 2 | 1,2,3,4 | 1, 1, 2, 2, 3, 4 |
| Heads | n/a | n/a | n/a | 4 | 4 |
| Heads channels | n/a | n/a | n/a | 64 | 64 |
| Attention resolution | n/a | n/a | n/a | 32, 16, 8 | 32, 16, 8 |
| Dropout | 0.0 | 0.0 | 0.1 | 0.0 | 0.0 |
| Batch size | 128 | 256 | 128 | 128 | 128 |
| Pretraining iterations | 50K | 10K | 100K | 100K | 100K |
| Finetuning iterations | 150K | 40K | 150K | 20K | 20K |
| Pretraining learning rate | 1e-4 | 1e-4 | 1e-4 | 2e-4 | 2e-4 |
| Finetuning learning rate | 1e-5 | 1e-4 | 1e-4 | 2e-4 | 2e-4 |
| Pretraining warmup steps | n/a | n/a | n/a | 5K | 5K |
| EMA decay | n/a | n/a | 0.999 | 0.999 | 0.999 |
| Parameters count | 133.4K | 371K | 8.8M | 194.4M | 226.7M |

Table 3: Hyper-parameters for each model. Note that for 2-networks models, the architectural hyper-parameters describe only one of the two identical networks. Approximate parameters counts are given for bidirectional networks, except for the Gaussian case, where we only experimented with a 2-networks model.

### K.1 2D Experiments

In addition to the experiments presented in the main text, we test our models in the simplest 2D data settings used in Tong et al. (2024a) and Shi et al. (2023). Note, that low-dimensional datasets might not be the ideal showcase for $\alpha$-DSBM given that one can successfully employ less computationally demanding techniques based on minibatch-OT methods (Tong et al., 2024b).

The results of our bidirectional model finetuned with online updates are given in Table 4. During finetuning, we generate samples using 100 Euler–Maruyama steps to solve the forward and backward SDEs. At test time, we solve the forward probability flow ODE (PF-ODE) given by:

$$\mathrm{d}\mathbf{X}_t = \frac{1}{2}\big[v_\theta(1, t, \mathbf{X}_t) - v_\theta(0, 1 - t, \mathbf{X}_t)\big]\mathrm{d}t, \qquad \mathbf{X}_0 \sim \pi_0. \tag{59}$$

To evaluate model fit, we compute 2-Wasserstein distance between the true and generated samples (generated with 20 Euler steps). Additionally, we estimate path energy as a measure of trajectory simplicity: $\mathbb{E}_{\mathbf{X}_0 \sim \pi_0}[\int_0^1 \|v_\theta(t, \mathbf{X}_t)\|^2 \mathrm{d}t]$ where $v_\theta(t, \mathbf{X}_t)$ is the drift of PF-ODE in (59), and the integral is approximated using 100 steps. We have made a deliberate effort to closely replicate the experimental setup of Shi et al. (2023) to ensure the comparability of our results. However, as

illustrated in Figure 13, 2-Wasserstein distance can be very noisy even with 10K samples in the test set. To mitigate this variance, we averaged the 2-Wasserstein distance across five random sets of 10K samples per run, and then averaged these results across multiple runs. Despite these measures, we recommend a future redesign of these 2D experiments to facilitate more robust comparisons between methods.

| Method | 2-Wasserstein | | | | Path energy | | | |
|---|---|---|---|---|---|---|---|---|
| | $\mathcal{N} \to$ moons | $\mathcal{N} \to$ scurve | $\mathcal{N} \to$ 8gaussians | moons $\to$ 8gaussians | $\mathcal{N} \to$ moons | $\mathcal{N} \to$ scurve | $\mathcal{N} \to$ 8gaussians | moons $\to$ 8gaussians |
| DSBM-IMF* | 0.144±0.024 | 0.145±0.037 | 0.338±0.091 | 0.838±0.098 | 1.580±0.036 | 2.092±0.053 | 14.81±0.255 | 41.00±1.495 |
| OT-CFM (Tong et al., 2024a)* | 0.111±0.005 | 0.102±0.013 | 0.253±0.040 | 0.716±0.187 | 1.178±0.020 | 1.577±0.036 | 15.10±0.215 | 30.50±0.626 |
| $\alpha$-DSBM | 0.168 ±0.011 | 0.213±0.031 | 0.292±0.047 | 1.374±0.286 | 1.439 ±0.024 | 2.052 ±0.025 | 15.038±0.150 | 37.626±0.590 |

Table 4: 2-Wasserstein distance and path energy for the 2D experiments. We report means ±1 standard deviations across 5 random seeds. DSBM-IMF* and OT-CFM* results are copied from Shi et al. (2023).

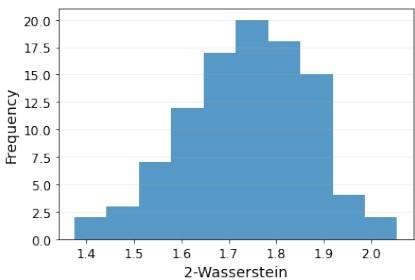

Figure 13: A histogram of 2-Wasserstein distances for the 'moons→ 8gaussians' task. These distances are calculated between 10K samples from a finetuned $\alpha$-DSBM model and 8gaussians distribution, with both sets generated using 100 different random seeds. The wide spread of scores indicates that 2-Wasserstein distance, even computed on 10K samples, may not be an ideal metric for evaluating model fit in this context.

## K.2 Gaussian data

To parameterise the forward and backward drifts, we use a 2-layer MLP network with 256 hidden units. To process time variables, we compute sinusoidal time embeddings, followed by a 2-layer MLP with 256 hidden units and 50 output units. The resulting time embeddings are then concatenated with $\mathbf{X}_t$, so the drift networks receive 100-dimensional input vectors.

For iterative DSBM finetuning, we perform 40K steps with varying number of outer iterations, i.e. when we switch between training the forward and the backward networks. Alternating every 5K steps, corresponds to 8 outer DSBM iteration. Similarly, changing the direction every 1K steps, leads to 40 outer iterations.

We do not have a cache dataloader like in the original DSBM implementation[2], thus we generate training samples on the fly by sampling either from the forward or the backward network. For this simple task, we also do not use EMA.

During training and evaluation, we use Euler–Maruyama method with 100 equidistant time steps between 0 and 1. The covariance is evaluated using 10K samples.

**Additional comparison $\alpha$-DSBM vs OT bridge-matching.** We consider the scalar Gaussian setting. We highlight the dependence of OT bridge matching (256/16/8) on the batch size, as the mini-batch OT coupling can be far from the true OT coupling as the batch size decreases. All experiments are run with similar compute and architecture/training hyperparameters, see Table 5.

**Ablation of the hyperparameter $\alpha$.** Instead of letting $\alpha$ be determined by Adam and adaptive for $\alpha$-DSBM, we explicitly set it by using Stochastic Gradient Descent (SGD) with learning parameter $\alpha$; see Figure 14. We also ran online $\alpha$-DSBM with $\alpha = 10^{-1}$ but the training diverges in this case.

---

[2]https://github.com/yuyang-shi/dsbm-pytorch

Table 5: Comparison of OT methods

| Method | Covariance |
|---|---|
| Optimal | 0.882 |
| $\alpha$-DSBM | **0.890** |
| Bridge Matching | 0.491 |
| OT Bridge Matching (256) | 0.853 |
| Bridge Matching (16) | 0.840 |
| Bridge Matching (8) | 0.824 |

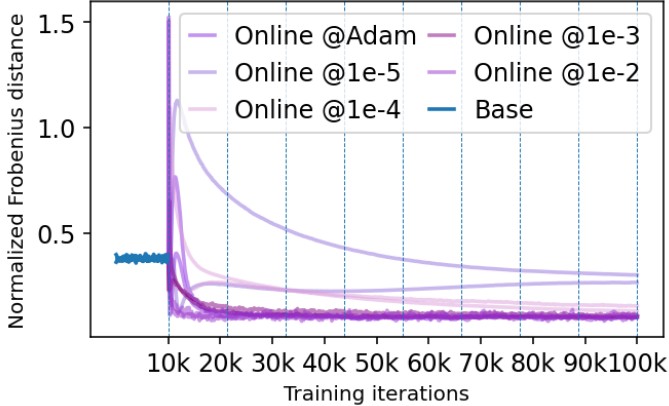

Figure 14: normFrob between $C_\star$ and its estimate for $\alpha$-DSBM with different values of $\alpha$.

## K.3  MNIST ↔ EMNIST transfer

We closely follow the setup of Shi et al. (2023) and De Bortoli et al. (2021), and train the models to transfer between 10 EMNIST letters, A-E and a-e, and 10 MNIST digits (CC BY-ND 4.0 license). We use the same U-Net architecture with hyperparameters given in Table 3.

For DSBM finetuning, we perform 30 outer iterations, i.e. alternating between training the forward and the backward networks, while at each outer iteration a network is trained for 5000 steps. We do not have a cache dataloader and generate training samples on the fly by sampling either from the forward or the backward network with EMA parameters.

During training and evaluation, we use Euler–Maruyama method with 30 equidistant time steps between 0 and 1. For evaluation, we compute FID based on the whole MNIST training set of 60000 examples and a set of 4000 samples that were initialised from each test image in the EMNIST dataset. MSD is computed between 4000 initial EMNIST test examples and their corresponding MNIST samples.

In Figures 15–18, we provide forward and backward samples, i.e. EMNIST → MNIST and MNIST → EMNIST, from models that differ in parameterisation, finetuning methods, and sampling strategy. For all the models above, we used $\varepsilon = 1$. Figure 19 illustrated the behaviour of the samples when we sweep over the $\varepsilon$ hyperparameter.

Pretraining a bidirectional model on 4 v3 TPUs takes 1 hour, while the online finetuning stage requires 4 hours on 16 v3 TPUs. The number of pretraining and finetuning steps is chosen to match the experimental setup of Shi et al. (2023).

## K.4  AFHQ: Cat ↔ Wild

We consider the problem of image translation between Cat and Wild domains of AFHQ (Choi et al. (2020); CC BY-NC 4.0 DEED licence) as introduced by Shi et al. (2023). Each domain has approximately 5000 samples in the training set, and around 500 samples in the test set. We resize the original 512 × 512 images to 64×64 or 256×256 resolutions.

(a) Initial EMNIST letters  (b) Bridge matching: FID=6.02, MSD=0.564  (c) Bridge matching + DSBM finetuning: FID=5.25, MSD=0.345  (d) Bridge matching + online finetuning: FID=4.28, MSD=0.368

Figure 15: EMNIST to MNIST transfer with a 2-networks model.

(a) Initial EMNIST letters  (b) Bridge matching: FID=6.33, MSD=0.572  (c) Bridge matching + online non-EMA finetuning: FID=4.57, MSD=0.369  (d) Bridge matching + online finetuning: FID=4.39, MSD=0.387

Figure 16: EMNIST to MNIST transfer with a bidirectional model.

(a) Initial MNIST digits  (b) Bridge matching: FID=7.50, MSD=0.553  (c) Bridge matching + DSBM finetuning: FID=3.56, MSD=0.330  (d) Bridge matching + online finetuning: FID=3.67, MSD=0.357

Figure 17: MNIST to EMNIST transfer with a 2-networks model.

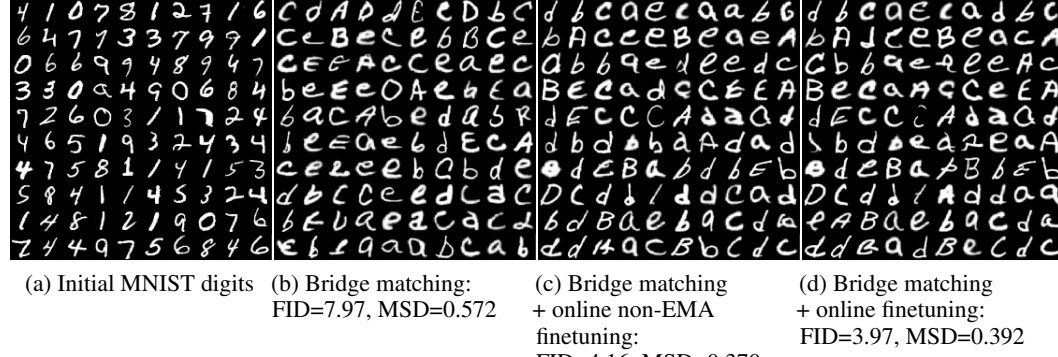

(a) Initial MNIST digits | (b) Bridge matching: FID=7.97, MSD=0.572 | (c) Bridge matching + online non-EMA finetuning: FID=4.16, MSD=0.370 | (d) Bridge matching + online finetuning: FID=3.97, MSD=0.392

Figure 18: MNIST to EMNIST transfer with a bidirectional model.

Our U-Net (Ronneberger et al., 2015) implementation is based on Ho et al. (2020) with a few improvements suggested in Dhariwal and Nichol (2021); Song et al. (2021b) such as rescaling of skip connections by $1/\sqrt{2}$, using residual blocks from BigGAN (Brock et al., 2019), and convolution-based up- and downsampling. Hyperparameters are given in Table 3. Compared to the straightforward parameterisation of the vector fields, we obtained slightly better results using EDM preconditioning Karras et al. (2022), which we derive in Appendix J for the case of bridge matching. During training, we use horizontal flips as a way to augment the data.

During training and evaluation, we use Euler–Maruyama method with 100 equidistant time steps between 0 and 1. When evaluating the quality of Cat → Wild transfer, we compute FID based on the whole training set of 4576 examples in the Wild domain and a set of 480 samples that were initialised from test images in the Cat domain. LPIPS and MSD are computed between 480 initial Cat images and Wild samples from the model. The same procedure is followed when evaluating in the reverse direction from Wild to Cat. Given that train, and especially the test sets are small, the quantitative results for AFHQ are likely unreliable (Chong and Forsyth, 2020). In Figure 22 we provide samples from the models finetuned either with an iterative or an online method. While their FID scores are different, the samples look similar between the two models.

As we discussed in the main text, hyperparameter $\varepsilon$ trades off the visual quality and alignment of the samples in the resulting transfer models. In Figure 20, we provide AFHQ $64 \times 64$ samples for pretrained and finetuned models with different values of $\varepsilon$. In addition to its relation to EOT, from a DDM perspective, $\varepsilon$ can be seen as the controlling factor of the noise schedule. As observed by Hoogeboom et al. (2023), noise schedules should be adjusted for different image sizes by shifting the noise schedule of some reference resolution where it is proven to be successful. In our case, if we find a good value of $\varepsilon$ for $64 \times 64$ images, then a shifted $\varepsilon$ for the $256 \times 256$ resolution can be computed as $\varepsilon_{256} = \varepsilon_{64} \left(\frac{256}{64}\right)^2$. Thus, if we choose $\sqrt{\varepsilon} = 0.75$ for AFHQ-64, then for AFHQ-256, we can expect $\sqrt{\varepsilon} = 3.0$ to also work well. Samples from an AFHQ-256 model trained with $\sqrt{\varepsilon} = 3.0$ are given in Figure 27.

On 16 v3 TPUs, the bidirectional base and finetuned AFHQ-64 models take 4 and 14 hours to train, respectively. For AFHQ-256, the base model trains for 15 hours, and finetuning takes an additional 37 hours. While we did not experiment with varying pretraining and fine-tuning iterations, these training times suggest that a longer pretraining stage followed by fewer fine-tuning steps may be desirable.

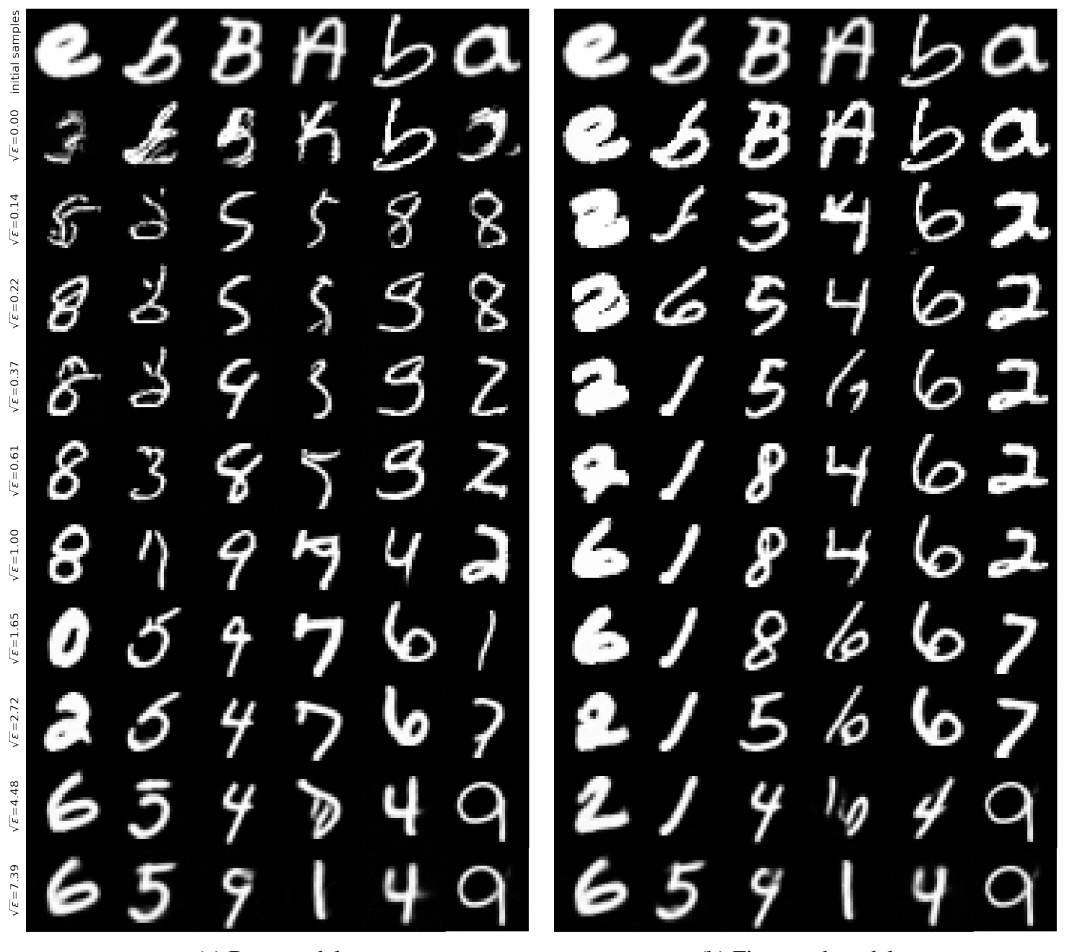

(a) Base model        (b) Finetuned model

Figure 19: MNIST samples transferred from EMNIST letter inputs (top row) using base (pretrained) and fine-tuned models for different values of $\varepsilon$. Low noise values result in poor sample quality, particularly in the base model, which finetuning cannot fully rectify. Conversely, excessively high $\varepsilon$ restricts information passing from the inputs to the outputs, leading to poor alignment. Additionally, high $\varepsilon$ increases blurriness due to increased noise levels, thus requiring more denoising steps.

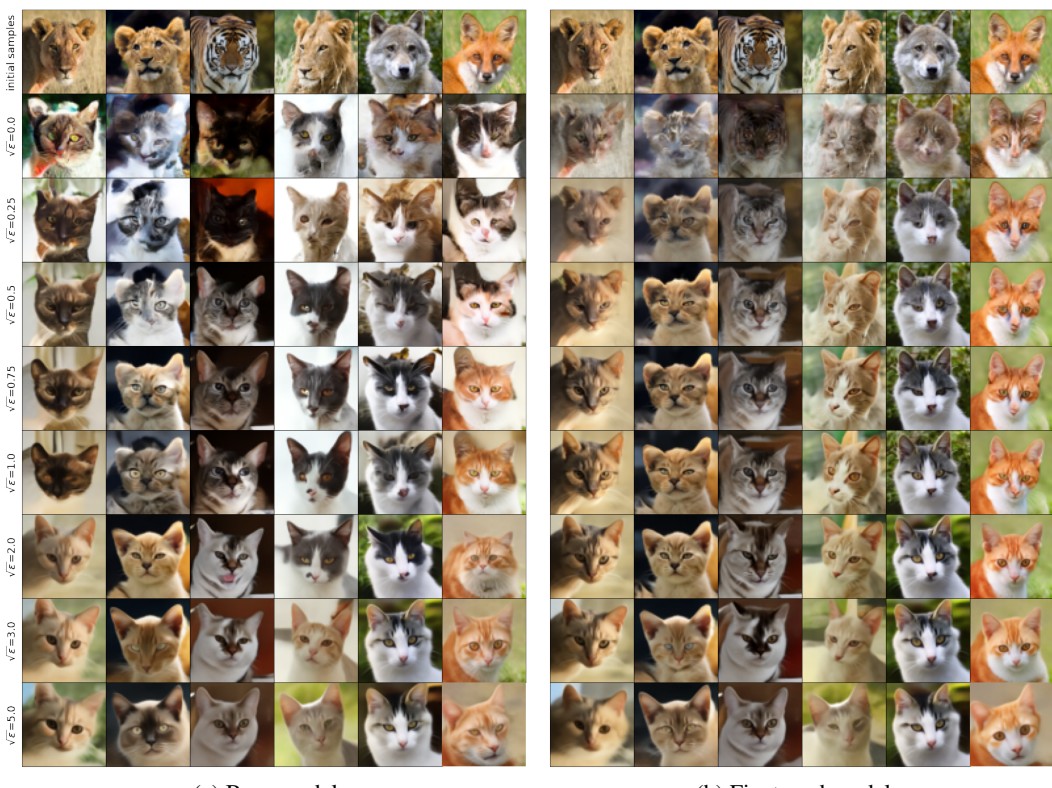

(a) Base model            (b) Finetuned model

Figure 20: AFHQ $64 \times 64$ Wild $\rightarrow$ Cat transfer results for different values of $\sqrt{\varepsilon}$ in a bidirectional model before and after online finetuning. Low values of $\varepsilon$ lead to poor sample quality in both base and finetuned models. Excessively high $\varepsilon$ values impede information passing from the inputs to the outputs, resulting in poor alignment. High values of $\varepsilon$ also increase blurriness due to noisier SDE trajectories, thus requiring more denoising steps during sampling.

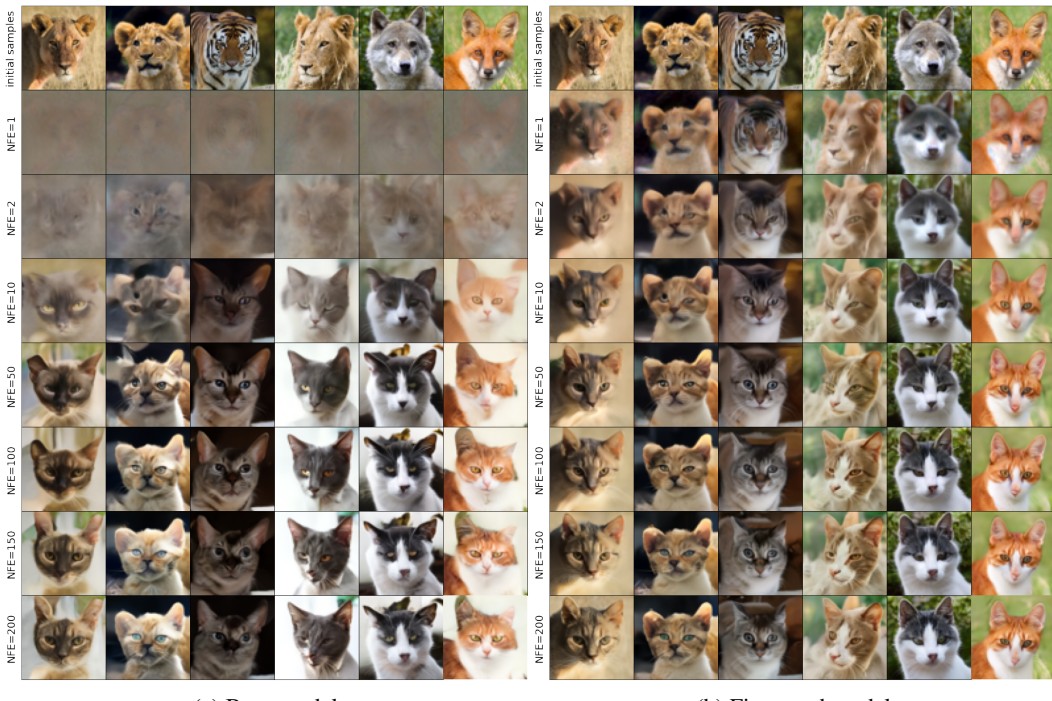

(a) Base model  (b) Finetuned model

Figure 21: AFHQ $64 \times 64$ Wild $\rightarrow$ Cat transfer results for varying number of function evaluations (equivalent to time discretisation steps in the Euler-Maruyama method) in a bidirectional model with $\sqrt{\varepsilon} = 0.75$, both before and after online finetuning. Post-finetuning, clearer images are achievable with fewer steps. This observation aligns with findings from Rectified Flows (Liu et al., 2023b).

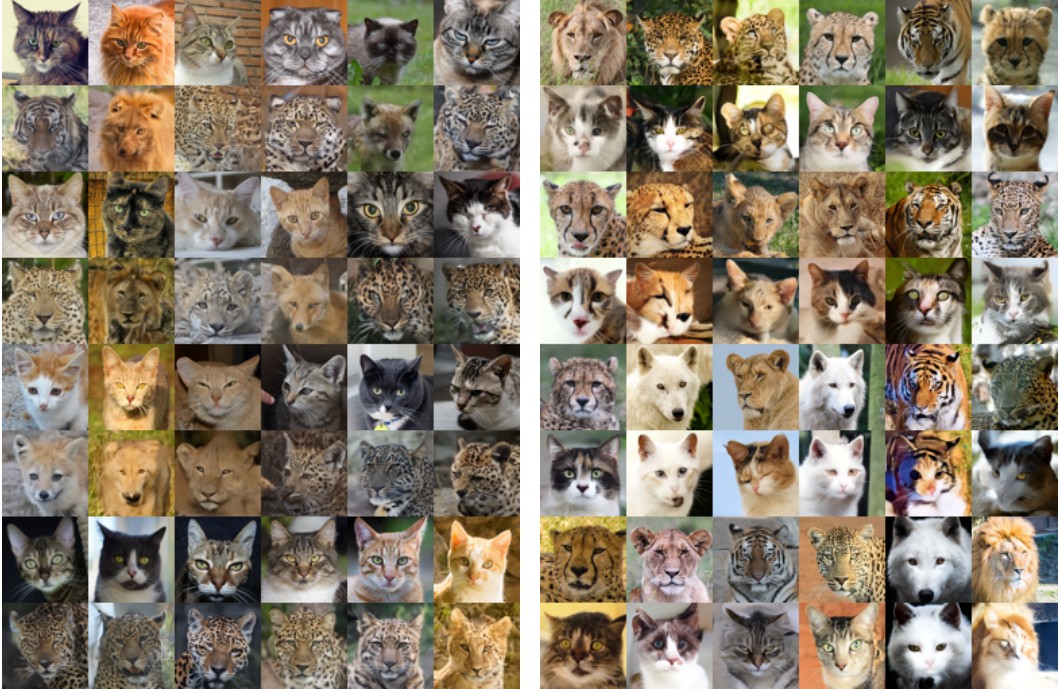

(a) Iterative finetuning: Cat → Wild.
FID=27.76, LPIPS=0.503, MSD=0.093

(b) Iterative finetuning: Wild → Cat.
FID=25.24, LPIPS=0.483, MSD=0.094

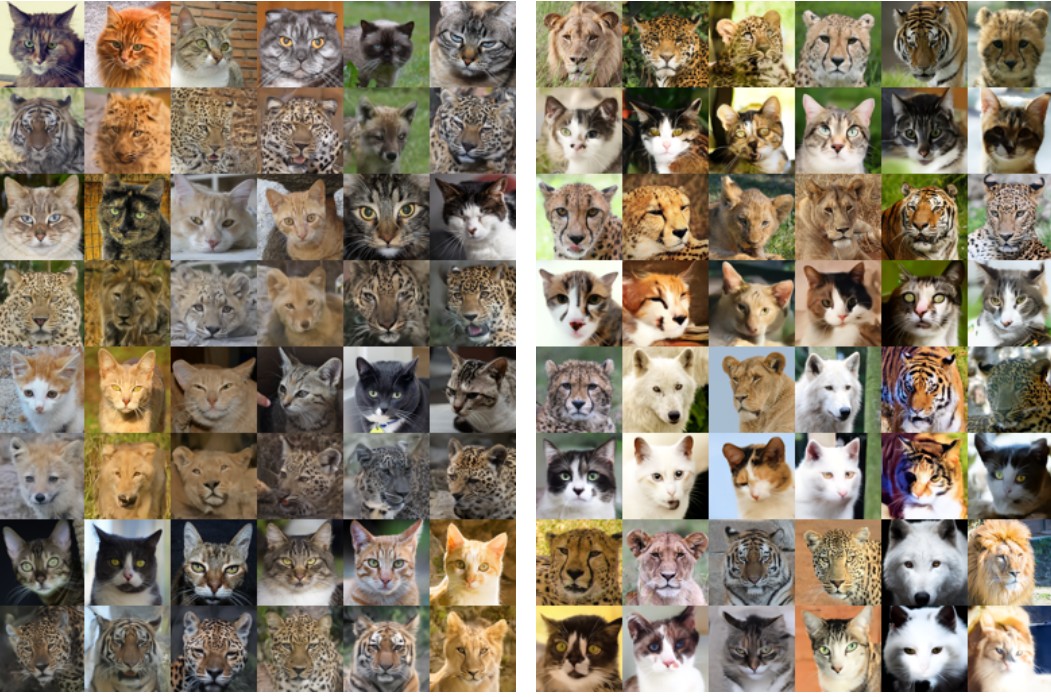

(c) Online finetuning: Cat → Wild.
FID=32.12, LPIPS=0.503, MSD=0.097

(d) Online finetuning: Wild → Cat.
FID=27.32, LPIPS=0.485, MSD=0.116

Figure 22: Samples and metrics from a 2-networks model architecture finetuned with DSBM's iterative procedure vs online finetuning. Within each two rows, initial and transferred samples are on the top and bottom respectively.

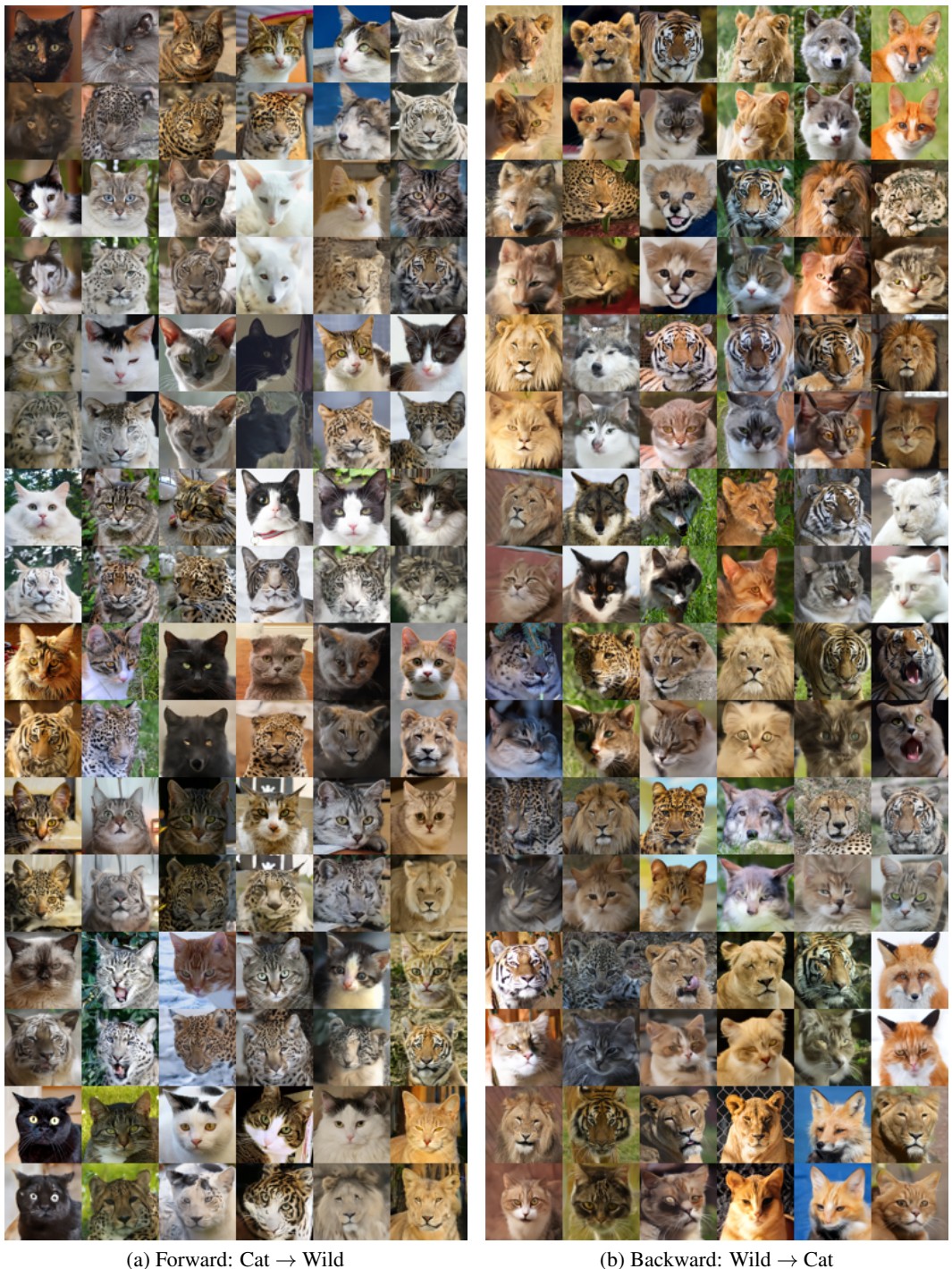

(a) Forward: Cat → Wild

(b) Backward: Wild → Cat

Figure 23: Uncurated samples for AFHQ $64 \times 64$ transfer in a bidirectional model with online finetuning with non-EMA sampling and $\sqrt{\varepsilon} = 0.75$. Within each two rows, initial and transferred samples are on the top and bottom respectively.

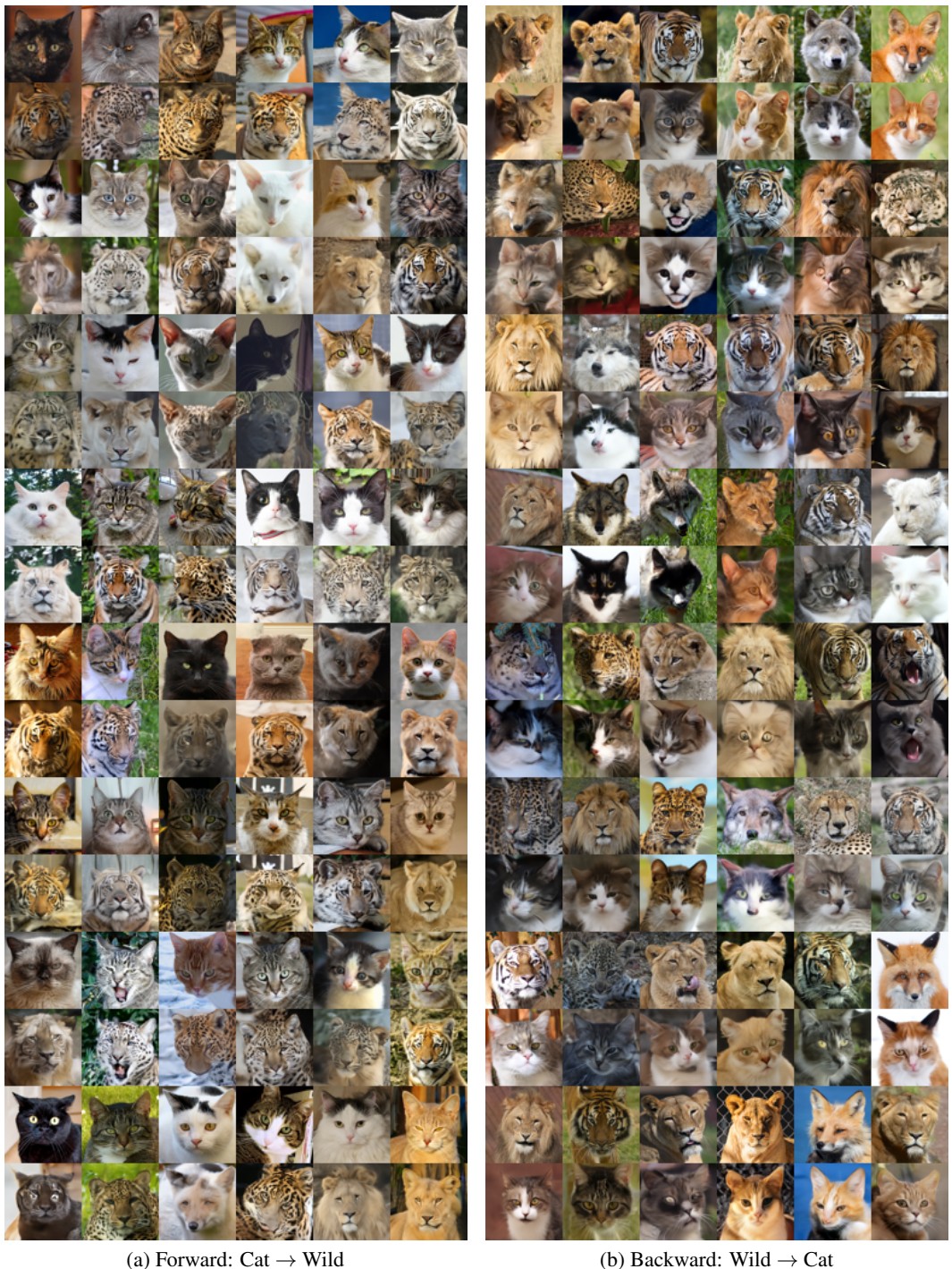

(a) Forward: Cat → Wild                    (b) Backward: Wild → Cat

Figure 24: Uncurated samples for AFHQ 64 × 64 transfer in a bidirectional model with online finetuning and $\sqrt{\varepsilon} = 0.75$. Within each two rows, initial and transferred samples are on the top and bottom respectively.

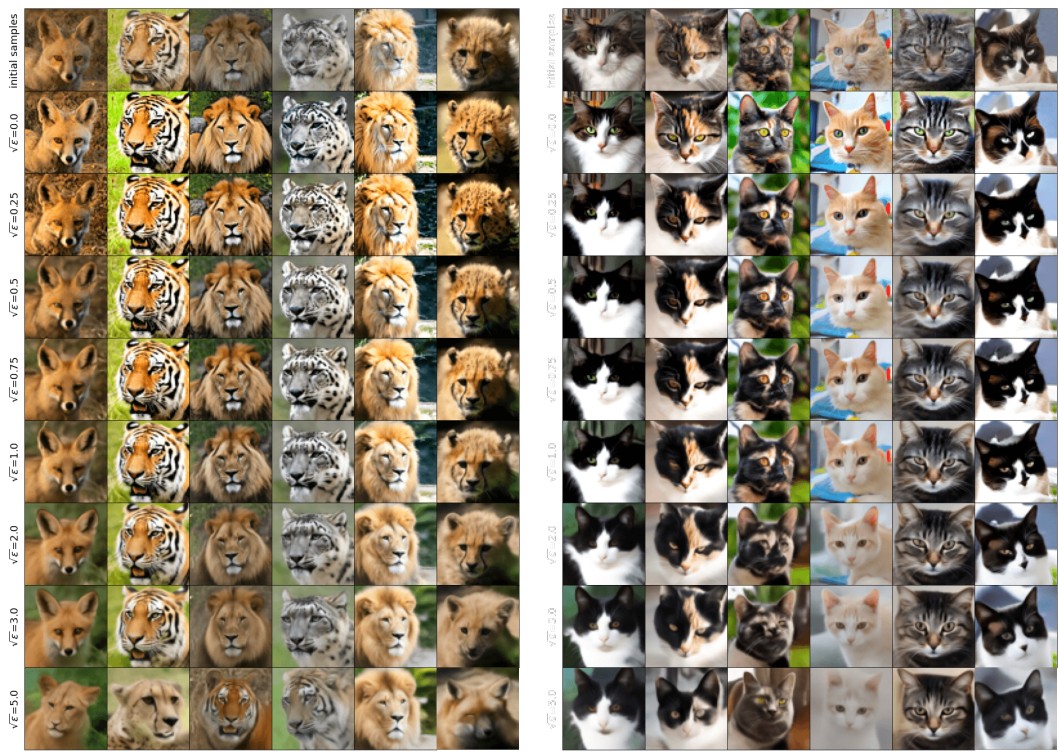

(a) Forward: Cat → Wild with inputs from Wild.

(b) Backward: Wild → Cat with inputs from Cat.

Figure 25: Samples for AFHQ $64 \times 64$ transfer in bidirectional models with online finetuning and different values of $\varepsilon$. The models are only trained on Cat and Wild domains, $\pi_0$ and $\pi_1$, respectively. Thus, in the forward direction the models expect Cat samples as inputs at $t = 0$, and transfer them to the Wild domain at $t = 1$. The reverse transfer holds in the backward direction. Here, we test the models' behaviour when inputs do not come from the same distribution as during training: we feed Wild samples in the forward direction, and Cat samples in the backward, which is the opposite from what the models expect. Ideally, the model should leave these inputs unchanged, which it does to varying degrees depending on $\varepsilon$, variance of the Gaussian noise. As we increase $\varepsilon$, less information can pass from the input to the output, thus making them less alike.

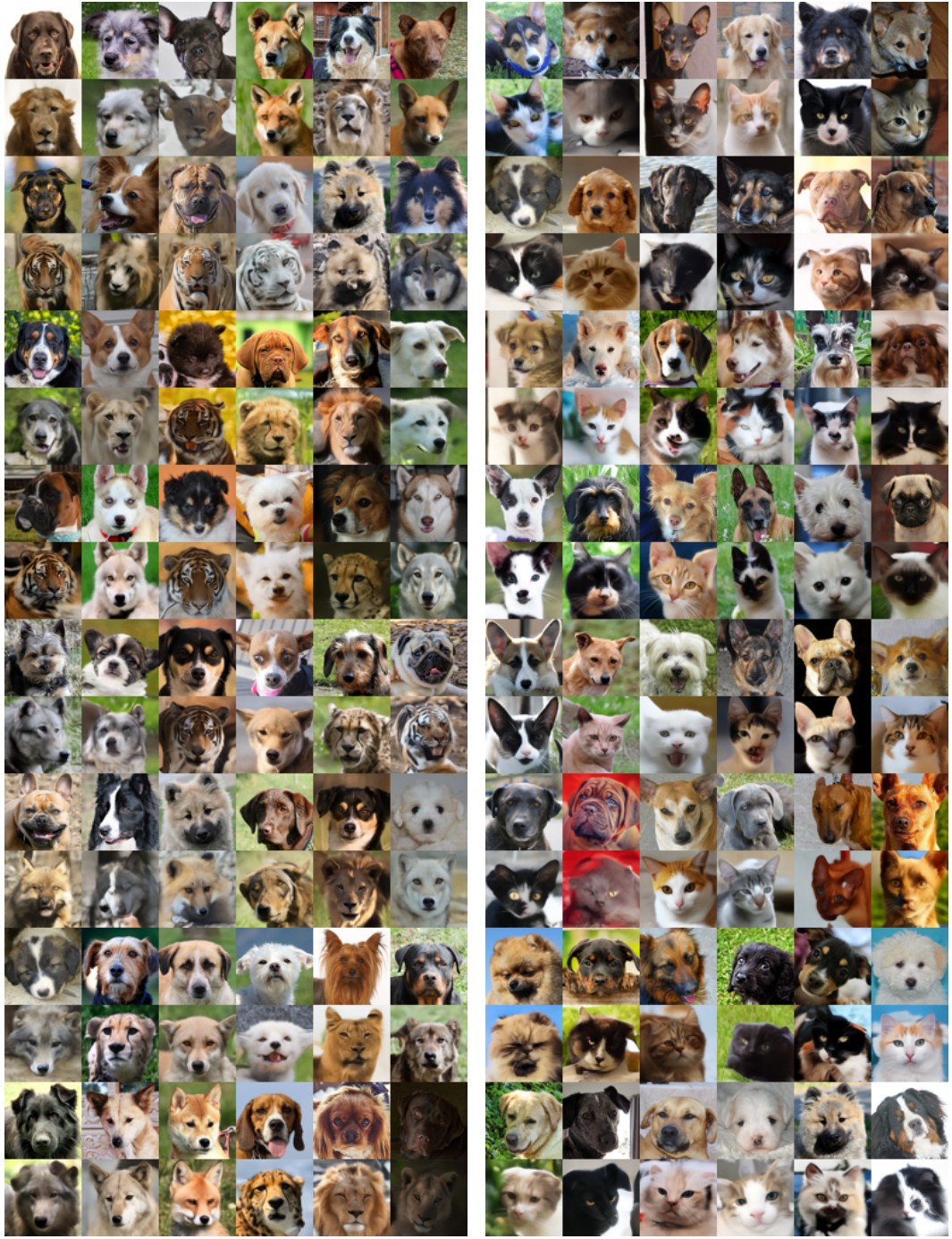

(a) Forward: Cat → Wild with inputs from Dog.    (b) Backward: Wild → Cat with inputs from Dog.

Figure 26: Samples for AFHQ $64 \times 64$ transfer in a bidirectional model with online finetuning and $\sqrt{\varepsilon} = 2.0$. The model is only trained on Cat and Wild domains, $\pi_0$ and $\pi_1$, respectively. Thus, in the forward direction the model expects Cat samples as inputs at $t = 0$, and transfers them to the Wild domain at $t = 1$. The reverse holds in the backward direction. Notably, the model generalises well to the unseen AFHQ Dog domain, often producing high-quality translations. These results come from a model with $\sqrt{\varepsilon} = 2.0$, which is higher than our chosen default value of $\sqrt{\varepsilon} = 0.75$. Higher noise allows the model to better deal with out-of-distribution inputs.

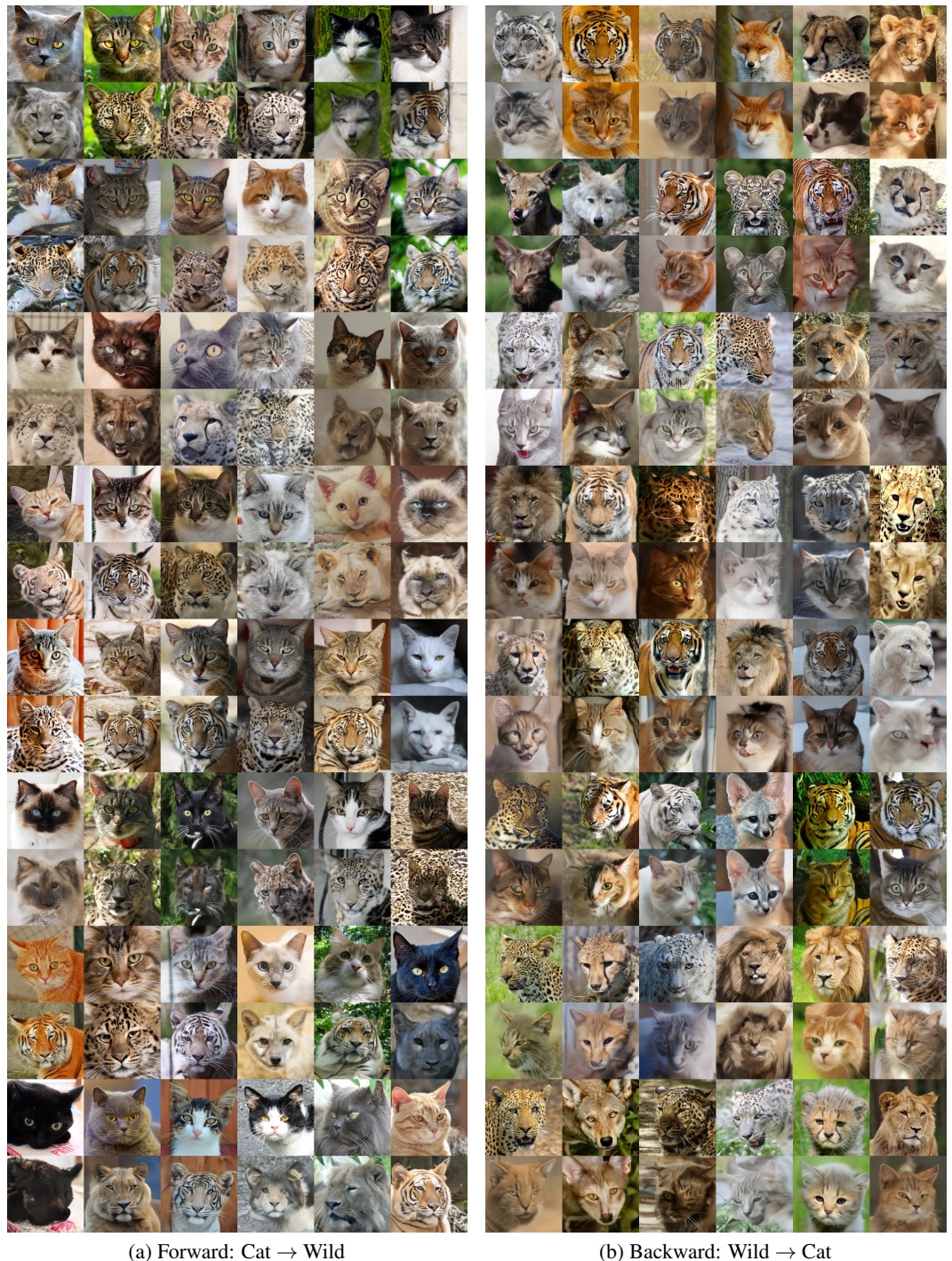

(a) Forward: Cat → Wild

(b) Backward: Wild → Cat

Figure 27: Uncurated samples for AFHQ 256 × 256 transfer in a bidirectional model with online finetuning and $\sqrt{\varepsilon} = 3$. Within each two rows, initial and transferred samples are on the top and bottom respectively.

