# OpenReview forum: "Schrodinger Bridge Flow for Unpaired Data Translation"
_NeurIPS.cc/2024/Conference — NeurIPS 2024 spotlight_

### Official Review · Reviewer_KqVg · 2024-07-12

**Soundness:** 3
**Presentation:** 2
**Contribution:** 3
**Rating:** 6
**Confidence:** 3

**Summary:**

This paper proposes $\alpha$-IMF, which is an incremental version of IMF method in DSBM paper. Specifically, this work demonstrate the convergence properties of $\alpha$-IMF and implement it through an online learning method. Here, $\alpha$ is implicitly reflected. Moreover, the functional flow interpretation provides a basis for online learning, allowing for online learning as opposed to the iterative approach of DSBM. From an engineering perspective, the authors suggest parameterizing forward and backward control simultaneously by adding a forward/backward indicator (0 or 1) as an input variable to the neural network. The method shows feasible results on toy (Gaussian) data and image-to-image translation tasks.

**Strengths:**

-	This work is theoretically fruitful. The introduction of $\alpha$-IMF leaded to a more expansive theoretical framework. This flow-based perspective enables online learning.
-	Reduced the sensitivity to hyperparameters such as the iteration number, the number of phase in DSBM.
-	Expands the interpretation of the Schrödinger bridge by demonstrating connections with reinforcement learning (RL) and Sinkhorn flow.

**Weaknesses:**

- The experiment on practical data is insufficient and the performance improvement is incremental. The paper only presents real-world I2I on Cat <-> Wild. Moreover, the performance of bidirectional online learning improves only incrementally. Totally, it needs more evaluation on other kinds of dataset, e.g. Male <-> Female / Handbags <-> Shoes.
- It would be beneficial to quantify the advantages of bidirectional learning or online learning over DSBM. For instance, comparing the actual wall-clock time or GPU memory usage would be useful.
- The current algorithm reflects $\alpha$ implicitly, and so there is no analysis provided on $\alpha$. If there were any intuition or insights into the role of $\alpha$ (step size) or if an algorithm could be developed to explicitly control $\alpha$, it would offer more valuable insights.

**Questions:**

- If we perform this algorithm with full gradient descent algorithm, is it possible to explicitly control $\alpha$?
- Can we develop exact algorithm that reflect $\alpha$ explicitly? If so, can we obtain some experimental results that compares $\alpha$?

**Limitations:**

The authors adequately addressed the limitations.

---

> ### Author Rebuttal · Authors · 2024-08-06
>
> Thank you for your comments, we appreciate your acknowledgements of the paper’s merits.
>
> > The experiment on practical data is insufficient and the performance improvement is incremental. The paper only presents real-world I2I on Cat <-> Wild. Moreover, the performance of bidirectional online learning improves only incrementally. [...] For instance, comparing the actual wall-clock time or GPU memory usage would be useful.
>
> Bidirectional online learning approximately halves the number of network parameters, thereby reducing GPU memory usage and wall-clock training time. In the case of AFHQ-64, pretraining bidirectional and 2-network models takes 4 and 9 hours, while finetuning takes 14 and 19 hours, respectively. We use the same number of gradient updates for both models. The computational cost difference comes from the use of a bidirectional versus a 2-network model. However, note that bidirectional models are incompatible with the original iterative DSBM and are suitable only for online finetuning.
> A fair comparison of DSBM and α-DSBM in terms of the convergence speed, is then possible with a 2-networks model. It is best illustrated by the Gaussian example where there is a single evaluation metric.
>
> > The current algorithm reflects 𝛼 implicitly, and so there is no analysis provided on 𝛼. If there were any intuition or insights into the role of 𝛼(step size) or if an algorithm could be developed to explicitly control 𝛼, it would offer more valuable insights.
>
> If we perform this algorithm with full gradient descent algorithm, is it possible to explicitly control 𝛼?
> Can we develop exact algorithm that reflect 𝛼 explicitly? If so, can we obtain some experimental results that compares 𝛼?
>
> We agree with the reviewer that parameter $\alpha$ merits further discussion. In our current manuscript, we state in  (l.218) “In Algorithm 1, we specify $\alpha \in (0,1]$ as a stepsize parameter. In practice, we  use Adam (Kingma and Ba, 2015) for optimization, thus the choice of $\alpha$ is implicit and adaptive throughout the training.” So, $\alpha$ is linked to the learning rate: in a Stochastic Gradient Update (SGD), $\alpha$ would correspond to the learning rate. To address this point comprehensively (and also answers reviewer Ti9G), we train a model with SGD and a sweep on the values of $\alpha$; see additional one-page rebuttal. We found out that setting explicitly the value of $\alpha$ in SGD can yield similar results as letting $\alpha$ adaptive using Adam. However, for too large values of $\alpha$ the algorithm diverges.

---

> > ### Comment · Reviewer_KqVg · 2024-08-11
> >
> > Thank you for the response. I would like to keep my score to 6.

---

### Official Review · Reviewer_Ti9G · 2024-07-12

**Soundness:** 3
**Presentation:** 3
**Contribution:** 3
**Rating:** 7
**Confidence:** 4

**Summary:**

The authors consider developing a new algorithm for a Schrödinger Bridge problem of translating between two probability distributions. Motivated by the fact that current Schrödinger Bridge (SB) approaches either use mini-batch optimal transport techniques or require training diffusion at every iteration, the authors propose discretizing the Schrödinger Bridge Flow (SB Flow) of path measures that converge to SB. They call this discretization $\alpha$-Iterative Markovian Fitting ($\alpha$-IMF) procedure. Using the fact that Markovian path measures characterizing the considered flow can be parametrized by diffusion vector field $v$, authors propose a non-parametric scheme for updating $v_n$ corresponding to $n$ discretization step of SB Flow and prove its correctness. To implement the procedure in practice, authors prove a parametric counterpart to the non-parametric update scheme that allows for optimization over a vector field $v_{\theta}$ parametrized as a neural network called $\alpha$ Diffusion Schrödinger Bridge Matching ($\alpha$-DSBM). Furthermore, the authors computationally validate the proposed algorithm on both vector and image state spaces. The authors compare their $\alpha$-DSBM algorithm to other DSBM versions [3] in the Gaussian setting. Image state space experiments were held on image-to-image translation tasks, i.e., MNIST $\leftrightarrow$ EMNIST, Wild $\leftrightarrow$ Cat on AFHQ 64/256, with quantitative evaluation and comparison with the original DSBM algorithm [3].

**Strengths:**

- The overall theory is novel and interesting. The authors prove that the proposed $\alpha$-IMF procedure with non-parametric updates converges to the solution of SB.
- The parametric counterpart to $\alpha$-IMF ($\alpha$-DSBM), which can be used with neural networks, is proposed and empirically verified.
- The possibility of online $\alpha$-DSBM finetuning seems beneficial in terms of computational resources compared to regular DSBM iterative training.
- A sufficient comparison of the proposed $\alpha$-DSBM with the previously known DSBM algorithm [3] is presented for a wide set of hyperparameter $\epsilon$.
- The Appendix presents an extensive additional theoretical review of the proposed concept and its connections to existing research.
- The study on the parameterization of diffusion in both directions by one neural network with varied conditional vectors is proposed and tested.

**Weaknesses:**

- The proposed $\alpha$-DSBM algorithm [3] is, in some sense, an advancement of the previously known DSBM algorithm [3]. It is expected to be more efficient because it removes the necessity to learn a distinct model at each iteration. However, the paper lacks a study of image quality boost given the same computational budget. Thus, it is not clear whether the authors achieved the goal of developing a more efficient algorithm.
- The convergence of parametric $\alpha$-DSBM is not proved. While it may be a hard theoretical problem to prove, at least a more extensive empirical evaluation of the setup with the known ground truth solution would be beneficial. The presented setup with a scalar covariance matrix is too simple. It would be a more solid argument to evaluate the Gaussian setup with a full covariance matrix [6] or use a mixture benchmark [7] and compare it with other neural network methods.
- The FID metric is similar for DSBM and $\alpha$-DSBM for the AFHQ experiment in Table 1 considering measured standard deviation. Thus again highlighting the lack of study with the same computational budget to justify that $\alpha$-DSBM is more efficient.
The chosen AFHQ dataset may not be the best choice considering the small test set, which may introduce bias in measuring the FID metric. In turn, the visual comparison is hard since most images produced by DSBM and $\alpha$-DSBM and given in Figure 19 are similar, and it is hard to deduce which algorithm is better. A bigger dataset such as Celeba [5] may help to solve this issue. At least a bigger dataset would give a good estimate of FID metrics.
- It seems that some hyperparameters for DSBM AFHQ experiments are missing. The number of pre-training iterations, grad updates per IMF iteration, and IMF iterations are not specified. What is the training time and computational budget?
- There is no study on *cornerstone* hyperparameter $\alpha$. What is the value of $\alpha$ used in the experiments? Is it equal to the learning rate of the diffusion model parameters?
- The absence of code may cause trouble with reproducing the presented results.

**Questions:**

- Since the AFHQ dataset is quite small, experiments on unpaired male-to-female translation on the Celeba [5] dataset could provide a more meaningful comparison (as it was in DSBM[3]).
- How do DSBM and $\alpha$-DSBM compare in terms of computational budget? Could authors provide a quantitative comparison?
- How does variation of $\alpha$ affect speed of convergence and deviation from the SB solution?
- Can authors provide experimental results of $\alpha$-DSBM on more complex Gaussian distributions or multimodal low dimensional data, i.e., gaussian mixtures?
- Is it possible to extend the proposed approach to GSBM [4], where, as far as my understanding goes, authors change the reciprocal projection step compared to DSBM [3]?
- Can this method with small $\epsilon$ can be applied to the generation problem? This could drastically reduce the computational budget of such and be applied in modern text-to-image generative models [2] similar to the work [1]. It would be great to see such experimental results.
- Is it possible to perform *parametric* updates for Sinkhorn flow in a similar to $\alpha$-DSBM way?

Overall, the paper looks very promising. I will consider increasing my score if the authors address some of the listed weaknesses and questions, especially related to the same computational budget comparison, and evaluate their method on one of the listed more complex setups with the known ground truth distributions.

[1] Liu, Xingchao, et al. "Instaflow: One step is enough for high-quality diffusion-based text-to-image generation." The Twelfth International Conference on Learning Representations. 2023.

[2] Esser, Patrick, et al. "Scaling rectified flow transformers for high-resolution image synthesis." Forty-first International Conference on Machine Learning. 2024.

[3] Shi, Yuyang, et al. "Diffusion Schrödinger bridge matching." Advances in Neural Information Processing Systems 36 (2024).

[4] Liu, Guan-Horng, et al. "Generalized Schrödinger Bridge Matching." The Twelfth International Conference on Learning Representations.

[5] Liu Z. et al. Deep learning face attributes in the wild //Proceedings of the IEEE international conference on computer vision. – 2015. – С. 3730-3738.

[6] Hicham Janati, Boris Muzellec, Gabriel Peyré, and Marco Cuturi. Entropic optimal transport between unbalanced gaussian measures has a closed form. Advances in neural information processing systems, 33:10468–10479, 2020.

[7] Gushchin N. et al. Building the bridge of schrödinger: A continuous entropic optimal transport benchmark //Advances in Neural Information Processing Systems. – 2023. – Т. 36. – С. 18932-18963.

**Limitations:**

The authors adequately addressed the limitations and potential negative societal impact of their work.

---

> ### Author Rebuttal · Authors · 2024-08-06
>
> Thank you for your comments and thoughtful questions, we are glad you enjoyed the paper.
>
> > The paper lacks a study of image quality boost given the same computational budget.
>
> > It would be a more solid argument to evaluate the Gaussian setup with a full covariance matrix [6].
>
> > A bigger dataset such as Celeba may help to solve this issue.
>
> We appreciate the reviewer's suggestions regarding our experimental setup. Concerning the image quality improvement given the same computational budget, we'd like to emphasize the inherent challenge in designing appropriate metrics for evaluating OT-based algorithms, as we are considering two distinct metrics: 1) Image quality (FID score): This roughly quantifies the alignment between the marginals $\pi_0$ and $\pi_1$ and the output distributions of the sampling process, 2) Alignment between samples from $\pi_0$ and $\pi_1$: This quantifies the minimization of the OT cost. Our objective is to minimize both the FID score and the alignment cost, necessitating the tracking of both these metrics. Nevertheless, we have conducted experiments comparing DSBM and $\alpha$-DSBM with equivalent computational costs, see the one-page supplementary.
>
> Additionally, we have now evaluated our methodology in a complex Gaussian setting using full covariance matrices, similar to the approach in [1]; see one-page additional rebuttal. The findings from this full covariance matrix setting agree with our earlier conclusions from the scalar setting.
>
> Finally, we have also evaluated our method on CelebA and refer to the one-page supplementary.
>
> [1] Janati et al., “Entropic optimal transport between unbalanced gaussian measures has a closed form”, 2020.
>
> [2] Chen et al., “Gradient flow in parameter space is equivalent to linear interpolation in output space”, 2024
>
> > The number of pre-training iterations, grad updates per IMF iteration, and IMF iterations are not specified. What is the training time and computational budget?
>
> We have added the missing hyperparameters for the training of DSBM in the case of the AFHQ experiment to the revised manuscript. The number of pretraining iterations is fixed to 100 000. The number of gradient updates per IMF iteration is 500 and the number of IMF iterations is 40.
>
> > There is no study on cornerstone hyperparameter 𝛼. What is the value of 𝛼 used in the experiments? Is it equal to the learning rate of the diffusion model parameters?
>
> We agree that we should have provided more discussion about the choice of parameter $\alpha$.  Currently, we say (l.218) “In Algorithm 1, we specify $\alpha \in (0,1]$ as a stepsize parameter. In practice, we use Adam for optimization, thus the choice of $\alpha$ is implicit and adaptive throughout the training.” So indeed, $\alpha$ is linked to the learning rate and if one were to use SGD and not Adam, $\alpha$ would be exactly the learning rate. To complement this answer and also fully answer reviewer KqVg, we train a model with SGD and a sweep on the values of $\alpha$. The results are reported in the one-page rebuttal. We found  that setting explicitly the value of $\alpha$ in SGD can yield similar results as letting $\alpha$ adaptive using Adam. However, for too large values of $\alpha$ the algorithm diverges.
>
> > The absence of code may cause trouble with reproducing the presented results.
>
> We are working on releasing the notebooks to reproduce our experiments on Gaussian data. Unfortunately, due to confidentiality agreements and intellectual property concerns, we are unable to open source the full source code. We have included a code snippet of our main training loop implementing our online methodology ($\alpha$-DSBM) in the appendix of the revised paper.
>
> > Is it possible to extend the proposed approach to GSBM
>
> We thank the reviewer for this remark. Indeed our online approach can be used to improve the computational efficiency of every method derived from DSBM. For example Generalized SBM [1] would also benefit from the techniques used in $\alpha$-DSBM. We have added a comment on this in the revised manuscript.
>
> > Can this method with small epsilon can be applied to the generation problem? [...] [2] similar to the work [1].
>
> You are correct: $\alpha$-DSBM can be used with $\varepsilon$ small (or even $\varepsilon = 0$, even though in that case, our theoretical framework does not guarantee the convergence to the solution of the OT problem). Hence, our technique can potentially enhance the performance of the 2-Reflow of [1] in a generative model context. However, we emphasize that in [2], there is no finetuning stage and only a flow-matching pretraining (i.e. this text2img model only corresponds to the pretraining of our model). As the pretraining and finetuning of such models is long and expensive, we could not conduct such experiments during the rebuttal period, however we plan to include such text2img experiments in the final version of the paper.
>
> [1] Liu et al., InstaFlow: One Step is Enough for High-Quality Diffusion-Based Text-to-Image Generation, 2024.
>
> [2] Esser et al.,  Scaling Rectified Flow Transformers for High-Resolution Image Synthesis, 2024.
>
> > Is it possible to perform parametric updates for Sinkhorn flow in a similar to $\alpha$-DSBM way?
>
> The applicability of a methodology similar to $\alpha$-DSBM to Sinkhorn-flow [1] is not straightforward. First, $\gamma$-Sinkhorn is an inherently static algorithm. In order to apply a methodology closer to $\alpha$-DSBM, we would need to develop a dynamic version of $\gamma$-Sinkhorn. This would correspond to extending DSB [2] to an online setting. It is unclear whether the objective of $\gamma$-Sinkhorn, see Lemma 1 in [1], can be modified to yield a tractable loss for an hypothetical $\gamma$-DSB version.
>
> [1] Karimi et al., Sinkhorn Flow: A Continuous-Time Framework for Understanding and Generalizing the Sinkhorn Algorithm, 2023.
>
> [2] De Bortoli et al., Diffusion Schrödinger Bridge with Applications to Score-Based Generative Modeling, 2021.

---

> > ### Comment · Reviewer_Ti9G · 2024-08-09
> >
> > I thank the authors for their reply, which resolves most of my concerns and questions. To finalize, could you please provide the metrics for the quality (e.g., FID) and similarity (e.g., L2 cost or LPIPS) for the Celeba dataset experiment? Also, could you please provide estimates on the GPU time you used to train DSBM and alpha-DSBM?
> >
> > Please incorporate Celaba, Guassian, text2img experiments, and other changes into the final revisions.

---

> > > ### Comment · Area_Chair_dKji · 2024-08-11
> > > **Reminder: Respond to Reviewer Questions by August 13, 11:59 PM AoE**
> > >
> > > Dear Authors,
> > > \
> > > \
> > > A quick reminder that the discussion phase with reviewers is open until **August 13, 11:59 PM AoE**. Please take some time to engage with Reviewer Ti9G and see if you can address the raised concerns in response to your rebuttal. Your engagement is crucial for ensuring that the reviewers have all the information they need to make informed decisions.
> > > \
> > > \
> > > Thank you for your continued efforts and contributions to the review process.
> > > \
> > > Best regards,
> > > Area Chair, NeurIPS 2024

---

> ### Author Response · Authors · 2024-08-12
>
> Thanks a lot for your comments which have improved the overall quality of the paper.
> Here are the additional metrics requested below (the hardware setup is identical to AFHQ):
> * Base model training time: 16 hours
> * Finetuning alpha-DSBM/Finetuning DSBM: 7 hours
>
> The finetuning of DBSM corresponds to two IMF iterations (one forward and one backward).
>
> Regarding the L2, LPIPS evals (lower is better):
> * DSBM: 0.159 (L2) / 0.451 (LPIPS)
> * alpha-DSBM: 0.05 (L2) / 0.376 (LPIPS)
>
> We are also reporting the Inception Score (higher is better) evals for the base models, DSBM and alpha-DSBM:
> * Base training: 2.29
> * DSBM: 2.88
> * alpha-DSBM: 3.13
>
> Hence, alpha-DSBM improves over DSBM for the same amount of compute. We see that the main reason for the underperformance of DSBM is that for that amount of compute there is little deviation of the alignement score, i.e. the model is still close to the base model. We will incorporate these numbers in the final version of the paper. We hope that these numbers resolve the concerns of the reviewer.

---

> > ### Comment · Reviewer_Ti9G · 2024-08-12
> >
> > Thank you for your reply and additional results. Since my concerns and questions were addressed, I updated my score.

---

### Official Review · Reviewer_xpT6 · 2024-07-12

**Soundness:** 3
**Presentation:** 4
**Contribution:** 3
**Rating:** 7
**Confidence:** 4

**Summary:**

This work introduces $\alpha$-DSBM, a new way of training DSBM-like models, which does not require a Markovian projection at each step and eliminates the need to train multiple models. The main advantage over previous DSBM-based approaches is that $\alpha$-DSBM only needs to train a single model with a single model, thus exhibiting a more stable training procedure.

The authors thoroughly contextualize their work within related work and provide detailed theoretical derivations to motivate their approach. Empirically $\alpha$-DSBM is validated by comparing to existing DSBM-based approaches across different unpaired image translation tasks as well as some toy data examples.

**Strengths:**

- The new $\alpha$-DSBM is introduced based on thorough theoretical motivation and derivations.
- The authors manage to present and contextualize their method very clearly within the scope of related work across the different bridge matching approaches (Appendix E) and highlight connections accordingly (e.g. connection to the reflow procedure).
- In general, the paper is well-written and, given the complexity of the topic, quite clear to follow.
- Empirical validation includes important ablations giving further insights into the hyperparameter choice of e.g. $\varepsilon$.

**Weaknesses:**

- While the experimental section of the paper thoroughly analyzes and compares DSBM and its different flavors, including the proposed method, a comparison with other competing methods could further support $\alpha$-DSBM through empirical evidence. Specifically, a comparison to bridge/flow matching with mini-batch OT similar to [1, 2], a highly optimized CycleGAN as in [3], and adversarial-based OT methods like [4] could be good candidates for further comparisons. The authors also mention these as competing methods.


[1] Alexander Tong and Kilian Fatras and Nikolay Malkin and Guillaume Huguet and Yanlei Zhang and Jarrid Rector-Brooks and Guy Wolf and Yoshua Bengio. "# Improving and generalizing flow-based generative models with minibatch optimal transport". IN TMLR 2024.

[2] Luca Eyring and Dominik Klein and Théo Uscidda and Giovanni Palla and Niki Kilbertus and Zeynep Akata and Fabian Theis. "Unbalancedness in Neural Monge Maps Improves Unpaired Domain Translation". In ICLR 2024.

[3] Dmitrii Torbunov and Yi Huang and Huan-Hsin Tseng and Haiwang Yu and Jin Huang and Shinjae Yoo and Meifeng Lin and Brett Viren and Yihui Ren. "UVCGAN v2: An Improved Cycle-Consistent GAN for Unpaired Image-to-Image Translation". In Arxiv 2023.

[4] Beomsu Kim and Gihyun Kwon and Kwanyoung Kim and Jong Chul Ye. "Unpaired Image-to-Image Translation via Neural Schrödinger Bridge". In ICLR 2024.

**Questions:**

- How does $\alpha$-DSBM perform compared to bridge matching with mini-batch OT sampling empirically? While it is clear that these methods introduce significant errors because of the mini-batch approximation, it is unclear how this affects results empirically compared to $\alpha$-DSBM. I think this is the most important additional competing work (apart from DSBM), as these methods share their overall goal.
- Does it make sense to combine $\alpha$-DSBM with mini-batch OT sampling for the initial pretraining? How would this impact $\alpha$-DSBM?

**Limitations:**

- The main limitation of $\alpha$-DSBM is that new data needs to be generated with the current model during its fine-tuning, making the procedure not simulation-free and, thus, more expensive. This is sufficiently addressed in the paper.

---

> ### Author Rebuttal · Authors · 2024-08-06
>
> Thank you for your positive comments and feedback.
>
> > While the experimental section of the paper thoroughly analyzes and compares DSBM and its different flavors, including the proposed method, a comparison with other competing methods could further support-DSBM through empirical evidence.
>
> > How does 𝛼-DSBM perform compared to bridge matching with mini-batch OT sampling empirically?
>
> We thank the reviewer for their suggestion to evaluate our approach against competing methods. While we agree that comparisons with [2,3,4] would be valuable, we have chosen, in this rebuttal, to focus on [1] which was also mentioned by reviewer ovZP and is most similar to our approach. However, we will mention [2,3,4] as possible alternatives with similar goals in our extended related work section. Our comparison with OT-bridge matching in the Gaussian setting is reported in the attached one-page rebuttal. We will include a comparison to [2,4] in the final version.
>
> [1] Tong et al., Improving and generalizing flow-based generative models with minibatch optimal transport, TMLR, 2024.
>
> [2] Eyring et al., Unbalancedness in Neural Monge Maps Improves Unpaired Domain Translation, ICLR, 2024.
>
> [3] Torbunov et al., UVCGAN v2: An Improved Cycle-Consistent GAN for Unpaired Image-to-Image Translation, arXiv:2303.16280, 2023
>
> [4] Kim et al., Unpaired Image-to-Image Translation via Neural Schrödinger Bridge, ICLR, 2024.
>
> > Does it make sense to combine 𝛼-DSBM with mini-batch OT sampling for the initial pretraining? How would this impact 𝛼-DSBM?
>
> This is indeed doable and is a great suggestion. This would require changing our pretraining strategy to include the mini-batch OT sampling during the training. Our guess is that the convergence of $\alpha$-DSBM (and even DSBM) would be faster. To support this result, we refer to the results of [1,2] which show that the convergence of the Sinkhorn algorithm depends on the Kullback-Leibler divergence between the original guess (here the output of the (E)OT-FM methodology) and the target Schr\”odinger Bridge.
>
> [1] Bernton, Schrödinger Bridge Samplers, arXiv:1912.13170, 2019.
>
> [2] L\’eger, A gradient descent perspective on Sinkhorn, Applied Mathematics & Optimization, 2021.

---

> ### Comment · Reviewer_xpT6 · 2024-08-12
>
> Thanks for the answers and added experiments. It would be very interesting to see comparisons to [1,2,4] also in the image translation settings. I think this setting would provide a more meaningful empirical comparison. Also thanks for pointing to [1,2] for the convergence of DSBM with mini-batch OT.

---

### Official Review · Reviewer_ovZP · 2024-07-13

**Soundness:** 3
**Presentation:** 3
**Contribution:** 2
**Rating:** 6
**Confidence:** 5

**Summary:**

The paper proposes a novel algorithm for mass transport problems, aiming to compute maps that transport one distribution to another. This paper introduces the Schrödinger Bridge Flow algorithm, a dynamic entropy-regularized version of OT, eliminating the need to train multiple DDM-type models. The algorithm discretizes a flow of path measures, ensuring the Schrödinger Bridge as the only stationary point. The paper demonstrates the algorithm's effectiveness on various unpaired data translation tasks, showcasing its potential to solve Entropic Optimal Transport problems with improved computational efficiency and practical implementation compared to existing methodologies.

**Strengths:**

1.This paper is eloquently written, and its ideas are easy to follow.
2.While I did not strictly follow all the proofs provided by the authors, I found them to be generally detailed, clear, and easy to understand.
3.The authors provide both quantitative and qualitative results on several datasets, along with a relatively detailed discussion and visualization. However, it is worth noting that the figures in the appendix appear to have low resolution, and the positioning of the legends is somewhat odd.
4.I understand that the convergence of the parametric methodology may require several strong assumptions. Therefore, it is reasonable that the authors have not provided further proof of convergence.

**Weaknesses:**

1.Assumptions in Theorem 3.1: Theorem 3.1 only mentions "under mild assumptions" without specifying them in the main text. It is necessary to state these assumptions clearly in the main body of the paper and provide references to justify their rationality. While a more detailed explanation can be provided in the appendix, the main text should at least include a summary of the assumptions to ensure the reader understands the conditions under which the theorem holds.
2.Experimental Results and Comparisons: The experimental results on some datasets are not particularly impressive, even though the authors claim that their method approximates OT maps more accurately than existing methods. The results in Table 1 do not show significant progress or state-of-the-art (SOTA) performance on certain datasets. Additionally, it is confusing that the authors explicitly mentioned Rectified Flow, Flow Matching, and OT-Flow Matching in the literature review, but did not include comparisons with these methods in the experiments. For instance, Rectified Flow has conducted experiments on mass transport, and other related methods, such as Denoising Diffusion Bridge Models from ICLR 2024, have tackled similar tasks. A direct comparison of the proposed method's performance with these existing methods would provide a clearer picture of its advantages and limitations.
3.Discussion of Related Work: The paper could benefit from a more comprehensive discussion of related work, including diffusion models, flow matching, and the application of Wasserstein gradient flow-based models in mass transport tasks. This would help in better contextualizing the authors' innovations and contributions. By situating their work more clearly within the existing literature, the authors could highlight the unique aspects and potential advantages of their proposed method, enhancing the readers' understanding of its significance.

**Questions:**

1.See Weaknesses
2.I am curious about the complexity and cost of the training method proposed by the authors. Could the authors provide more detailed quantitative data, such as training time comparisons with related methods?

**Limitations:**

Although the authors proposed a novel model to address mass transport problems and provided some theoretical proofs, the method is simulation-based. I believe that the training cost is higher compared to simulation-free methods, and the model performance does not achieve state-of-the-art (SOTA) results. Additionally, there is a lack of direct comparison with existing SOTA methods.

---

> ### Author Rebuttal · Authors · 2024-08-06
>
> Thank you for your comments, we appreciate your acknowledgements of the paper’s merits.
>
> > It is necessary to state these assumptions in the main body of the paper and provide references to justify their rationality.
>
> Due to space limitations, we have detailed most of the technical assumptions for our theorems in the supplementary material rather than in the main paper. However, if our submission is accepted, we will be granted an additional page, which will allow us to include the assumptions as detailed below.
>
> **Theorem 3.1**
>
> The assumptions for the result to hold are stated in Lemma D.2. We recall them here. Let $\pi_0$ and $\pi_1$ be the distributions at time 0 and 1. We require that $\pi_0$, $\pi_1$ have distributions w.r.t. Lebesgue measure, with finite entropy and finite 2nd moments. This finite 2nd moment assumption is quite weak. While the Lebesgue density hypothesis is commonly considered in the diffusion model literature, see e.g. [1,2].
>
> **Proposition 3.2**
>
> This proposition is detailed in Proposition D.1.The sole assumption in this case is that $\mathbb{P}_{v^n}$ is defined for all $n$. This means that the SDE with drift $v^n$ admits a weak solution. The existence of weak solutions of SDE is a well-known topic in the literature of SDE; see e.g. [3]. In practice this condition is satisfied since $v^n$ is replaced by a NN approximation, which satisfies the conditions for the SDE to admit a solution.
>
> **Proposition 4.1**
>
> The proof is similar to the one of Proposition 3.2 and therefore holds under the assumption that $v^{n, \rightarrow}$ and $v^{n, \leftarrow}$ define solutions of SDEs.
>
> [1] Lee et al., Convergence of score-based generative modeling for general data distributions, 2023
>
> [2] Conforti et al.,  Score diffusion models without early stopping: finite Fisher information is all you need, 2023
>
> [3] Stroock et al., Multidimensional Diffusion Processes, 2006.
>
> > The experimental results on some datasets are not particularly impressive, even though the authors claim that their method approximates OT maps more accurately than existing methods
>
> We want to emphasize that we do not claim our approach yields better results than DSBM (or any other method that recovers the EOT). There is no expectation of better results from our method compared to DSBM, as both converge to the same solution. Currently, we state “$\alpha$-DSBM is easier to implement than existing SB methodologies while exhibiting similar performance” (l.313).
>
> The main advantages of our methods are computational: 1) significantly simpler implementation, requiring only one fine-tuning stage and eliminating the need to alternate between two optimisation problems  2) utilization of only one NN (with the bidirectional implementation) 3) faster convergence, as shown in Fig. 2 (and in the Gaussian study in the one-page rebuttal). We hope this explanation clarifies the key computational contributions of our paper. We will revise our manuscript to further underline this point if it remains unclear.
>
> > It is confusing that the authors explicitly mentioned Rectified Flow (RF), Flow Matching (FM), and OT-FM in the literature review, but did not include comparisons with these methods.
>
> RF corresponds to DSBM for $\varepsilon = 0$. Our $\alpha$-DSBM algorithm thus provides an alternative online implementation of RF for $\varepsilon = 0$ and results for this online implementation are provided in Fig 3 (first column of right figure).  For unpaired data translation, RF is not competitive and adding noise improves performance. However, for generative modeling, RF outperforms DSBM and $\alpha$-DSBM using $\epsilon>0$. We conjecture that when one marginal distribution is Gaussian, adding further noise to the interpolant hurts the generative capabilities. In addition, the fixed points of RF exhibit straight paths but are not necessarily solutions to the OT problem, see [2] for a counterexample.
>
> FM [3] can be seen as the first iteration of DSBM with $\varepsilon = 0$. It consistently produces inferior results compared to RF, this is why we did not include it.
> OT-Flow Matching [4] is closer to our setting and, if the batch size has infinite size, then we recover the OT solution. In the one-page rebuttal we show that $\alpha$-DSBM outperforms this method in the Gaussian case.
>
> Finally, we also highlight that DDBM [5] do not solve the (E)OT problem but requires paired data to train the model and therefore is not comparable to our approach.
>
> [1] Liu et al., Flow Straight and Fast: Learning to Generate and Transfer Data with Rectified Flow, 2023.
>
> [2] Liu, Rectified Flow: A Marginal Preserving Approach to Optimal Transport, 2022.
>
> [3] Lipman et al., Flow Matching for Generative Modeling, 2023.
>
> [4] Tong et al., Improving and generalizing flow-based generative models with minibatch optimal transport, 2024.
>
> [5] Zhou et al., Denoising Diffusion Bridge Models, 2024
>
> > The paper could benefit from a more comprehensive discussion of related work
>
> In the revised manuscript, we have expanded the related work section in the appendix to offer a more comprehensive overview. Notably, we discuss a taxonomy graph of diffusion methods and their connection to OT, see the one-page rebuttal.
>
> > Could the authors provide more detailed quantitative data, such as training time comparisons with related methods?
>
> Bidirectional online learning approximately halves the number of network parameters, thereby reducing GPU memory usage and wall-clock training time. For AFHQ-64, pretraining bidirectional and 2-network models takes 4 and 9 hours, while fine-tuning takes 14 and 19 hours, respectively. We use the same number of gradient updates for both models. The computational cost difference comes from the use of a bidirectional versus a 2-network model. Note that bidirectional models are incompatible with the original iterative DSBM and are suitable only for online finetuning. We have included these cost and complexity considerations in the appendix.

---

> > ### Author Response · Authors · 2024-08-12
> >
> > We thank you for your review and appreciate your time reviewing our paper.
> > The end of the discussion period is close. We would be grateful if we could hear your feedback regarding our answers to the reviews. We are happy to address any remaining points during the remaining discussion period.
> > Thanks a lot in advance!

---

> ### Comment · Reviewer_ovZP · 2024-08-12
>
> Thank you for addressing many of the initial concerns in your rebuttal. However, I believe that if the main contribution is claimed to be a simpler implementation with only one neural network and faster convergence, it is crucial to provide more quantitative data to support these claims.
>
> I recommend:
>
> Including additional experiments that compare your model with other single neural network setups, highlighting computational efficiency and performance (In the rebuttal, the author provides some additional comparative experiments).
> Providing details on the reduction in neural network parameters to quantitatively demonstrate the simplicity of your approach.
> Clarifying the convergence rates in comparison with baseline methods, particularly considering the fine-tuning duration mentioned in your rebuttal experiments (I noticed that in the rebuttal experiments, the authors' fine-tuning time was much longer than the training time. Does this contradict the authors' statement?).
> If these points are addressed in the revised version and the theoretical assumptions are further clarified, I would be inclined to increase my score to a 5-6. Thank you for your efforts to improve the manuscript.

---

> > ### Author Response · Authors · 2024-08-12
> >
> > We thank the reviewer for their additional comment to which we answer below.
> >
> > > Thank you for addressing many of the initial concerns in your rebuttal. However, I believe that if the main contribution is claimed to be a simpler implementation with only one neural network and faster convergence, it is crucial to provide more quantitative data to support these claims.
> >
> > We will clarify this point in our introduction but we believe that our contribution is two-fold.
> >
> > * First, we introduce a new theoretical framework to analyze online schemes aimed at solving the Schrodinger Bridge (SB). To the best of our knowledge, this is the first time an online training algorithm is proposed to solve SB along with a theoretical framework. This is what motivates the $\alpha$-IMF scheme.
> > * Second, we introduce our practical implementation of $\alpha$-IMF, $\alpha$-DSBM. With the simplification given by $\alpha$-IMF this can be seen as a fine-tuning approach of any bridge/flow matching procedure. To further improve the scalability of our approach we only use one bidirectional network compared to existing implementations which use two networks.
> >
> > In the rebuttal we conducted additional experiments on CelebA to validate our findings. Because of the lack of space we could not include evaluation metrics in our initial rebuttal but report them below. We compare DSBM and $\alpha$-DSBM in the setting where both are evaluated with a single network (in order to answer the reviewer comment about “compare your model with other single neural network setups”). Both DSBM and $\alpha$-DSBM are run with the same number of gradient evaluations. The hardware set up and hyperparameters are similar to our AFHQ experiment. We report both alignment metrics and quality metrics.
> >
> > Regarding the L2, LPIPS evals (lower is better):
> > * DSBM: 0.159 (L2) / 0.451 (LPIPS)
> > * alpha-DSBM: 0.05 (L2) / 0.376 (LPIPS)
> >
> > We also report the Inception Score (higher is better) evaluations for the base models, DSBM and alpha-DSBM:
> > * Base training: 2.29
> > * DSBM: 2.88
> > * alpha-DSBM: 3.13
> >
> > We hope that this additional experiment resolves the reviewer’s concerns.
> >
> > > Providing details on the reduction in neural network parameters to quantitatively demonstrate the simplicity of your approach.
> >
> > We highlighted in the rebuttal that in our implementation we halve the number of parameters compared to DSBM. The results regarding the use of a bidirectional or a unidirectional network are reported in Table 1 in the original manuscript.
> >
> > > Clarifying the convergence rates in comparison with baseline methods, particularly considering the fine-tuning duration mentioned in your rebuttal experiments (I noticed that in the rebuttal experiments, the authors' fine-tuning time was much longer than the training time. Does this contradict the authors' statement?).
> >
> > We want to highlight that the finetuning is done for significantly fewer training steps than the initial base training (more precisely in the case of CelebA for instance: 100k base training and 10k finetuning). The reason the finetuning takes  longer than the base training is because our method (as well as DSB(M) and related methods) is not _sampling-free_. This is a well-known limitation of this line of work and we do not claim to solve this problem in our paper. We acknowledge this limitation in the original discussion of our paper. We agree that deriving “sampling-free” or almost sampling-free improvements of our methodology is an interesting future work.
> >
> > > the theoretical assumptions are further clarified,
> >
> > We have detailed the theoretical assumptions in our original answer to the reviewer. If any of the discussed assumptions remain unclear, we are willing to clarify them. We highlight that the detailed discussion of these theoretical assumptions will be included in the revised version of the paper as well as the discussion regarding the parameters reduction of the method and the additional comparison with DSBM and related methods.

---

> > > ### Comment · Reviewer_ovZP · 2024-08-14
> > >
> > > Thank you for your detailed reply and additional results. It addresses most of my concerns, and  I updated my score. I suggest the authors to include these discussions and the additional results in their revision.

---

### Author Rebuttal · Authors · 2024-08-06

We sincerely thank all reviewers for their valuable time and insightful feedback. We appreciate the thoughtful questions and the overall positive response.
We have provided detailed responses to each reviewer's comments. In summary, the key feedback points we have received are as follows:
* **Additional experiments**: as requested by all reviewers, additional experiments have been conducted to better illustrate the efficiency of the proposed method. In the one-page rebuttal, we have conducted additional experiments regarding the choice of $\alpha$, comparisons with OT bridge matching and experiments on CelebA.
* **Better support for the parametric case**: as suggested by reviewer TiG9 we have conducted an improved analysis of the parametric case. Following the recommendation of reviewer ovZP we have also included the assumptions our main theorems rely on in the main body of the paper.
* **Effect of the parameter alpha**: Reviewer TiG9 and reviewer KqVg have asked for more details regarding the choice of the parameter alpha. We have conducted additional experimental analysis of the effect of this parameter and report them in the one-page rebuttal.

Attached to this rebuttal is the additional one-page PDF with additional results and tables.

---

### Decision · Program_Chairs · 2024-09-25

**Decision:**

Accept (spotlight)

**Comment:**

This paper introduces a novel approach to solving Entropic Optimal Transport problems via the Schrödinger Bridge framework, demonstrating theoretical convergence and empirical improvements in image translation tasks. The reviewers generally found the work to be strong, with two reviewers (xpT6 and Ti9G) giving it an accept score of 7, showing clear support for the paper.

While Reviewer ovZP had concerns about the clarity of assumptions and comparisons with SOTA methods, these were sufficiently addressed during the discussion, leading to an increased score of 6. Reviewer KqVg also maintained a score of 6, acknowledging the incremental contributions but remaining positive overall.
Given the overall positive scores, particularly strong positive attitudes from two of the reviewers, I recommend accepting this paper.